



# Emissions Relationships in Western Forest Fire Plumes:
# I. Reducing the Effect of Mixing Errors on Emission Factors

Robert B Chatfield[1], Meinrat O Andreae[2,3], ARCTAS Science Team* and SEAC4RS Science
Team*

[1] NASA Ames Research Center, Moffett Field, CA 94035, USA
[2] Max Planck Institute for Chemistry P.O. Box 3060, D-55020 Mainz, Germany
[3] Scripps Institution of Oceanography, UCSD, La Jolla, CA 92093, USA
* A full list of authors and their affiliations appears at the end of the paper.

**Abstract**: Studies of emission factors from biomass burning using aircraft data complement the results of
lab studies and extend them to conditions of immense hot conflagrations. We illustrate and discuss emission
relationships for 422 individual samples from many forest-fire plumes in the Western US. The samples are
from two NASA investigations: ARCTAS (Arctic Research of the Composition of the Troposphere from
Aircraft and Satellites) and SEAC4RS (Studies of Emissions and Atmospheric Composition, Clouds, and
Climate Coupling by Regional Surveys). This work provides sample-by-sample enhancement ratios (EnRs)
for 23 gases and particulate properties. Many EnRs provide candidates for emission ratios (ERs, corre-
sponding to the EnR at the source) when the origin and degree of transformation is understood and appro-
priate. From these, emission factors (EFs) can be estimated when the fuel dry mass consumed is known or
can be estimated using the carbon mass budget approach. This analysis requires understanding the interplay
of mixing of the plume with surrounding air. Some initial examples emphasize that measured $C_{tot} = CO_2 +$
CO in a fire plume does not necessarily describe the emissions of the total carbon liberated in the flames,
$C_{burn}$. Rather, it represents $C_{tot} = C_{burn} + C_{bkgd}$, which includes possibly varying background concentrations
for entrained air. Consequently, we present a simple theoretical description for plume entrainment for mul-
tiple tracers from flame to hundreds of kilometers downwind and illustrate some intrinsic linear behaviors.
The analysis suggests a Mixed Effects Regression Emission Technique (MERET), which can eliminate
occasional strong biases associated with the commonly used normalized excess mixing ratio (NEMR)
method. MERET splits $C_{tot}$ to reveal $C_{burn}$ by exploiting the fact that $C_{burn}$ and all tracers respond linearly to
dilution, while each tracer has consistent EnR behavior (slope of tracer concentration with respect to $C_{burn}$).
The two effects are separable. Two or three or preferably more emission indicators are required as a mini-
mum; here we used ten. Limited variations in the EnRs for each tracer can be incorporated and the variations
and co-variations analyzed. The percentage CO yield (or the modified combustion efficiency) plays some
role. Other co-relationships involving nitrogen and organic classes are more prominent; these have strong
relationships to the $C_{burn}$ to $O_3$ emission relationship. In summary, MERET allows fine spatial resolution
(EnRs for individual observations) and comparison of similar plumes distant in time and space. Alkene
ratios provide us with an approximate photochemical timescale. This allows discrimination and definition,
by fire situation, of ERs, allowing us to estimate emission factors.



## 1. Introduction

### 1.1. Importance and Previous Work

Biomass burning has a large influence on the atmospheric burden of ozone and aerosols, and consequently also affects climate (Crutzen et al., 1979; Crutzen and Andreae; 1990; Jaffe and Wigder, 2012; Andreae, 2019). Biomass burning emission factors that are useful to drive photochemical models are most often estimated by one of two sampling techniques (Akagi et al., 2011). The first approach, measurements on the ground close to an open fire or on laboratory

fires that are controlled to approximate natural conditions, can provide the most detailed information on sources. The burning conditions can be readily assessed and fit into parameterizations of the emissions process, provided the correct mix of burn types typical of large fires can be estimated. It can, however, be difficult to mimic and safely sample truly intense flaming conflagrations. The second, measurements made from aircraft, provides a much wider sample of different

fires and emissions from different regions of a single fire. However, the estimates can be difficult to classify as simply "flaming" or "smoldering" or even as defined mixtures of just two types. Adjoining areas with fires in various stages of combustion can merge into the same plume, or remain relatively distinct. These questions of classifications related to the originating fires are addressed statistically in a succeeding paper, Chatfield and Andreae (2017).

Let us introduce our view of enhancement ratios, emission ratios, and emission factors. Under appropriately defined circumstances, the amount of fuel carbon burned that is liberated to the atmosphere is the sum of carbon added to the ambient air in the form of all fire-originated gases and particles as a result of combustion:

$$\Delta x = C_{\mathrm{burn}} = \Delta CO_2 + \Delta CO + \Delta CH_4 + \Delta(\text{particulate carbon species}) + \Delta\,(O)VOCs \qquad (1)$$

where the $\Delta$'s refer to the enhancement relative to pre-burn air.

In deriving emissions factors, i.e., how much of a species is emitted per kg of biomass burned, it is usual to obtain the amount of carbon burned from the sum of excess mixing ratios, $C_{\mathrm{burn}} = CO_2 + CO +$ other carbon-containing emissions, including aerosol particles. To an accuracy within $\gtrsim 1.5$ % (totals from the datasets we analyzed) to 3 % (Andreae and Merlet, 2001), carbon burned or $C_{\mathrm{burn}}$ is approximated by the excess ($CO_2 + CO$), as measured above a back-

ground concentration, $C_{\mathrm{bkgd}}$ (Andreae et al., 1988). With this understanding, we will assume $\Delta CO_{\mathrm{tot}} = \Delta CO_2 + \Delta CO$ in the following. In some measurement situations, frequent and accurate measurements of $CH_4$ and particulate $C$ are also available, and their inclusion could perceptibly add to the precision of estimates. Analysis proceeds similarly with these terms.

We will thus define the quantity $C_{\mathrm{burn}}$ as $C_{\mathrm{burn}} = \Delta x = \Delta CO_2 + \Delta CO$. A complication

arises from the fact that pre-burn air may have various compositions, especially when we consider various sources for low-level inflow air, and especially air that is entrained in the smoke plume by the time of sampling. This is an important topic, which has been discussed in detail by Guyon et al. (2005) and Yokelson et al. (2013) and that we will focus on below. An Enhancement Ratio (EnR) for a species or property $j$ with mixing ratio $y_j$ is then $\mathrm{EnR}_j = \Delta y_j / \Delta x$.

We will use the term "enhancement ratio," EnR, in this paper. For many species, EnRs sampled prior to substantial atmospheric transformation (e.g., chemistry or particulate processes) approximate emission ratios (ERs) and thus allow estimates of emission factors, EF. Emission factors are defined relative to the amount of fuel burned and are derived from emission ratios by accounting for the concentration of carbon in the biomass burned and adjustment of units (An-

dreae and Merlet, 2001). They can be derived from ERs by



$$EF_j = ER_j \times \frac{MW_j}{MW_c} \times C_{BM} \qquad (2)$$

where $ER_j$ is the emission ratio of species $j$, $MW_j$ and $MW_c$ are the molecular weight of species j and the atomic weight of carbon, respectively, and $C_{BM}$ is the carbon content of the dry biomass. For use of emission factors in modeling, the circumstances of emission need to be described clearly, and the ER must be a true result of emissions. This work is focused on clarifying appro-
priate methods of EnR, ER, and EF estimation.

One part of EF estimation concerns the amount of fuel consumed in fires, its carbon content, and the fraction liberated to the atmosphere (i.e., excluding char remaining on the ground); here we will focus on the other part of the question, which concerns the relationship of emitted compounds to the $C$ liberated to the atmosphere. Many of the EnRs we calculate appear good candi-
dates for EF estimates. One remaining task, making specific links of particular EFs to appropriate fire conditions for which they apply, requires individualized trajectories and fuel characterizations. This important task is beyond reasonable treatment in this publication, which focuses on the improving the understanding of airborne samples. It seems likely to us that uncertainties on the relation of area and fuel burned contribute more error to emissions estimates than those contributions of minor $C$-containing species in the plume that were described above.

There are other uses for EnRs that arise in understanding fire plumes, which revolve around the evolution of relatively fresh smoke plumes, e.g., the enhancement of ozone, peroxy acetyl nitrate, or other bound (not NO or $NO_2$) nitrogen species (Alvarado and Prinn, 2009, Alvarado et al., 2009; 2010, Jaffe and Widger, 2012). These also should have a direct relation the fuel C
burned and perhaps other properties such as burning conditions, fuel moisture, and fuel N content.

Two special sampling intensives utilizing NASA's fully instrumented DC-8 aircraft allowed us to investigate forest-burning emissions. In June, 2008, the aircraft sampled a variety of fire plumes around California (Jacob et al., 2010; Singh et al., 2010; 2012; Hornbrook et al. 2011)
during the California ARCTAS (Arctic Research of the Composition of the Troposphere from Aircraft and Satellites) intensive period. In a later part of the campaign, the DC-8 sampled in Northern Canada (Simpson et al., 2011); we excluded these plumes as representing different forest burn conditions. In 2013, the DC-8 made several samplings of forest fires in California and the Rocky Mountain West during SEAC4RS (Studies of Emissions and Atmospheric Composi-
tion, Clouds and Climate Coupling by Regional Surveys; Toon et al., 2015). We analyzed all of these fire plumes, but excluded samples east of 102 °W, which were mostly from agricultural fires. Our aim was to understand a variety of plumes, but limit variation to a single general category (temperate forest fires) as used for three-dimensional simulation models and geographical summaries.
Flight tracks for the period and locations of major fires during these periods are shown in Figure 1. Analysis of the vertical variation of fire tracers suggested that plumes below 5 km ASL included recent and informative fires in our study. We saw no unequivocal variation of composition with height, possibly due to limitations on aircraft maneuvers low and near the fires. Consequently, the aircraft samples likely cannot adequately represent ground-hugging smoke flows.

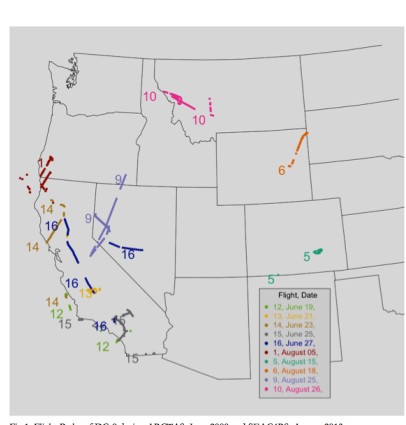

Fig 1. Flight Paths of DC-8 during ARCTAS, June 2008 and SEAC4RS, August 2013.

**Figure 1. ARCTAS and SEAC4RS flights analyzed in this work. Each flight is identified by the flight number of that series.**

We begin this work with a general overview of data analysis using conventional methods. We provide motivation for a new method, using $C_{tot}$ and combustion tracers in situations observed during the SEAC4RS mission. We then use these values to describe $C_{burn}$ as the difference between $C_{tot}$ and $C_{bkgd}$. This involves finding an "equivalent background value" of $C_{tot}$, a value which can be described as summarizing the amount of mixing-in of background $C_{tot}$ diluting a

sample. This is indicated by the agreement of all fire tracers employed. We describe the tracers we employed and the removal of urban influences. Two examples where naïve EnRs appear biased are described, one in Montana and one in Northern California. A theoretical section then gives a synthesis of tracer dilution in fire plumes and leads to a suggested empirical regression equation. Graphical examples of dilution histories emphasize the point that the regression is a

useful summary of more complex dilution processes. The "Mixed Effects Regression Emission Technique" or MERET algorithm is then described, a method that allows for species EnRs to vary from situation to situation. After describing practicalities and summarizing the method, the values of $C_{bkgd}$ and $C_{burn}$ indicated by the tracers are presented for all 422 samples. Results are shown resolving the issues seen for the Montana and California samples.

With $C_{burn}$ established, we give EnRs for the fire tracers in the many plume samples. Estimates of possible errors for each EnR are shown, as well as summaries of the estimated "true variability" of the tracer samples due to fire conditions and age. EnRs for other variables are described, especially those related to methane and to ozone. The time series suggest interesting correlations, and these are displayed and discussed. Questions regarding the extensibility of these

EnRs and their suitability for EF calculations are described and partially answered.

### 1.2. Development of EF estimation to date

EnRs and EFs for biomass burning plumes have largely been based on measurements of the $CO_2$ or CO concentrations in the plumes. Typical analyses begin with measurements of $C_{tot}$ and the concentrations of several tracers we may call $y_j$, $j = 1,…N$. Multiple instances $1, …, M$ are ob-

served, e.g., every few seconds or few minutes within a plume. An affine dependence is observed between each of the tracers and $C_{tot}$ with a $y$-intercept that depends most significantly on the local out-of-plume background values of $CO_2$, CO, and each tracer individually.


$$C_{tot} = C_{bkgd} + C_{burn} \tag{3}$$

The following analysis suggests several complexities that must be addressed in order to understand these affine relationships. Several aspects of slopes, intercepts, and deviations from linearity of the relationship of tracer $y_j$ to $C_{tot}$ plots must be examined, and so we transition to graphic terminology with $x$ representing $C_{tot}$. Later we will describe measurements of $C_{tot}$ and tracers $j$ at a given instance $i$, $x_i$ and $y_{ij}$. For a simple plume within a homogeneous mixed layer characterized by an $x$ concentration $x^E$ and y concentration $y^E$, we write

$$x = x^E + (x - x^E) \tag{4}$$

and

$$(y_j - y_j^E) = a_j(x - x^E) \tag{5}$$

with enhancement ratios, $a_j$, that can yield EFs directly. Early estimations (e.g., Greenberg et al., 1984; Andreae et al. 1988) used plots and regressions against $CO_2$ to estimate EnRs and EFs. These earliest techniques assumed fire was the main origin of $CO_2$. Very early it was recognized that other effects, e.g., variation of photosynthesis, respiration, and mixing required a more careful approach (e.g., Guyon et al., 2005). Alternatively (e.g., Andreae et al., 1988; Hobbs et al., 2003; Lefer et al., 1994), EnRs were derived with respect to CO,

$$ER\ estimate = a_{j \leftarrow CO} \cdot a_{CO \leftarrow (fire-added\ C_{burn})}$$

regression estimates of an EnR of the species with respect to CO, and a more careful estimate of the EnR of CO with respect to fire-produced $C_{burn}$. The $a_{CO \leftarrow (fire-added\ CO_2+CO)}$ was described using the Modified Combustion Efficiency,

$$MCE = \Delta CO_2/(\Delta CO_2 + \Delta CO) = 1 - \Delta CO/(\Delta CO_2 + \Delta CO) = 1 - EnR_{CO}. \tag{6}$$

The method has become known as the normalized excess mixing ratios method (NEMR; Akagi et al, 2011). Yokelson and others (2013) described the care required to make sure that the MCE was well defined; otherwise, severe difficulties ensue. They describe a situation in which $x^E$ and $y_{CO}^E$ in a diluting plume took on two distinct values, a mixed-layer value and a free-troposphere value, during plume rise and transport. More than two values may be relevant, emphasizing their call for careful sampling of pre-fire air and its dilution environment. We describe below new methods to resolve many of the difficulties with $x^E$ and to indicate unwanted effects of $y_{CO}^E$ variability. These methods could provide EnRs for many species with reasonable precision under more conditions.

This need for caution was very evident in the ARCTAS and SEAC4RS observational situations. Some Western USA data we analyzed showed variations in background $C_{tot} = CO_2 + CO$ (away from direct recent effects of respiration and photosynthesis) of 15 ppm (interquartile Range of 4 ppm), while other Western USA regions showed variations of ~8 ppm (according to the analyses in this paper that we present later in Figure 4). The contributions from fires were often comparable to this variation, ~2 − 40 ppm, mean ~6 ppm. Air flowed from the west into forest fires at low altitudes, or later diluted the smoke plume at intermediate levels. We could expect background air with a variety of histories of influence by photosynthesis (lower resultant $CO_2$) or respiration (higher $CO_2$), or urban-influenced air (higher $CO_2$). Low-level inflow air could have been mostly affected by local forests, farming, etc. Some of the most problematic situations tend to be associated with plumes sampled early in the day, when air from a nocturnal boundary layer — strongly enriched with respiration $CO_2$ — is mixed into the smoke plumes (Guyon et al., 2005). There could also have been substantial variations in $C_{tot}$ due to intercontinental transport, the composition reflecting long-previous modification due to these same processes and to latitudinal gradients. Yates et al. (2011) reported and more fully referenced atmospheric sampling of





Western air showing variations in $CO_2$ and also $CH_4$ and $O_3$. On the east side of the Pacific Anti-cyclone, the common pattern was for descent and horizontal shearing displacements, producing
substantial $C_{tot}$ variations in both horizontal and vertical directions (Barry and Chorley, 1998).

Previous analyses have been made for the ARCTAS data, by Simpson et al. (2011) for the large Canadian fires sampled and by Hornbook et al. (2011) for all fires. The Hornbook article usefully complements this paper by describing features and origins of the plumes sampled. Both groups described novel methods, but followed the traditional CO-emissions-ratio or NEMR
methodology (Andreae et al., 1988; Hobbs et al., 2003; Akagi et al., 2011). Pfister et al. (2011) cosidered the emissions and transport of CO in the California ARCTAS samples. Analyses for the SEAC4RS fires have also been reported (Liu, 2018).

We now move to motivate an alternate approach to the description of EnRs and EFs, the Mixed Effects Regression Estimation Technique, MERET; in some cases, MERET and NEMR
form complimentary supporting views of plume emissions. Whereas NEMR depends on multiple measurements in the same plume in an understood environment, MERET is typically applicable to individual measurements of similar EnR-determining fire conditions across many different plumes. It does instead require several informative fire tracer species be measured simultane-ously, not simply $CO_2$ or CO, as well as the tracer whose EnR is desired. It can also be used for
good candidate EFs when the environmental history of the plume is not well characterized. It is applicable to any plumes encountered, without need for extensive measurement of that plume history.

The MERET technique attempts to use the simultaneous variability, sample by sample, of a large set fire tracer compounds and aerosol descriptors to find a single quantification, $C_{burn}$, of
fire emissions, which it splits from $C_{bkgd}$ such that the sum is $C_{tot}$. To do so, it must also ascribe a set of EnRs to the fire tracers, and recognize that these EnRs may vary from sample to sample in a limited way. The interplay of these estimates contributing to $C_{burn}$ and EnRs for each observa-tion appears daunting. Consequently, section 3 will provide some examples of the special nature of this interplay. Section 4 will describe a theory of multiple fire emissions co-emitted from a
fire based on familiar plume concepts, and give examples. The examples show the linearity of the theory that such simple approaches with a limited number of parameter estimates yield a rea-sonable approximation to more complex behavior. Sections 5 and 6 describe a mixed-effects re-gression algorithm based on plume theory. Section 7 provides a limited number or EnR estimates and describes graphically how flight segments describing similar emissions conditions can be
identified.

## 2. Methodology: defining an indicator dataset

An initial task is the identification of appropriate variables and sampling rates. The technique we describe requires the measurement of $C_{tot} \approx CO_2 + CO$ and several concentrations of emitted species or similar, extensive, properties of emissions (e.g., dried-airstream scattering coefficients,
$b_{scat}$), which we will call *emission indicator species* or tracers. A set of indicator species was cho-sen for this publication to enable deriving relevant EnRs and to support our initial classification (e.g., flaming, smoldering, high-$N$ fuel, etc.). It is important to have as many differently behaving emission indicator species as possible, as different indicators may respond differently to different fuels and fire intensities ("fire chemistries"), and such variations are usually not known before
analysis. We favored indicator species with rapid sampling rates, so as to define $C_{burn}$ for the maximum number of instances, but certain variables like CO, $CH_4$, and $b_{scat}$ had special claims,





as they can be maximally expressed in important types of fires. It was convenient to use these same frequently measured indicator variables to define $C_{burn}$ and also for classification of fire chemistries. For classification, we added intensive variables, essentially ratios that should be physically independent of $C_{burn}$.


**Table 1. Indicator Variables**

| Concentration / property | Abbreviation | Technique | Group | Reference |
|---|---|---|---|---|
| **Extensive quantities** | *Proportional to carbon burned: define* | | | |
| Toluene | $C_6H_5CH_3$ | PTRMS | Wisthaler | Wisthaler, et all. 2013. |
| Benzene | $C_6H_6$ | PTRMS | Wisthaler | Wisthaler, et all. 2013. |
| Formaldehyde | HCHO | LAS | Fried | Fried et al., 2008. |
| Acetonitrile | $CH_3CN$ | PTRMS | Wisthaler | Wisthaler, et all. 2013. |
| Absorption Coefficient Dry, Total, 532 nm | $b_{abs}$ , Abs_5 | Nephelometry | Anderson | Wagner et al., 2015, Anderson Langley Aerosol Group, LARGE |
| Scattering Coefficient, Dry, Submicron 550 nm | $b_{acat}$ , Scat_5 | Nephelometry | Anderson | " |
| Carbon monoxide | CO | LAS, GC | Diskin, Blake | Pfister et al., 2011. |
| Methane | $CH_4$ | LAS, GC | Diskin, Blake | Pfister et al., 2011. |
| Acetaldehyde | $CH_3CHO$ | PTRMS | Wisthaler | Wisthaler, et all. 2013. |
| Methanol | $CH_3OH$ | PTRMS | Wisthaler | Wisthaler, et all. 2013. |
| **Intensive quantities** | *Not proportional to carbon burned* | | | |
| Single Scattering Albedo | SSA | Nephelometry | Anderson | Wagner et al., 2015, Anderson |
| Ångström Exponent, scattering | ÅE | Nephelometry | Anderson | " |
| **Other variables used** | $O_3$, $NO_x = NO + NO_2$, $NO_y$ | Chemilumines-cence, UV | Weinheimer (ARCTAS) Ryerson (SEAC4RS) | Weinheimer, et al. et al. 1994. Ryerson et al., 2000 |

Notes: PTRMS: proton transfer mass spectrometry, LAS: laser absorption spectrometry. $1 - \varpi$ is the single-scatter co-albedo: likewise, CO is linked to 1 – (Modified Combustion Efficiency), so that all values extend upwards from 0. $CO_2$ measurements: see text.


The emissions indicator species that satisfied these requirements for both missions are shown in Table 1, along with references to the measurement techniques and observers. Only extensive quantities (proportional to $C_{burn}$) are used in this paper. $CO_2$ was measured by Stephanie Vay (ARCTAS) and Andreas Beyersdorff (SEAC4RS) using the AVOCET instrument (Vay et al., 2011). In examining EnRs for various species, we also use the organic aerosol (OA) measure-ments of Jose Jimenez and ionic composition information from the Jack Dibb group. The ARC-

TAS and SEAC4RS data sites give full information, as instrumentation characteristics naturally vary somewhat between missions. (https://www-air.larc.nasa.gov/cgi-bin/ArcView/arctas, https://www-air.larc.nasa.gov/cgi-bin/ArcView/seac4rs?DC8=1)





Our techniques use algorithms that currently allow few missing observations among the variables. The sampling rates for emission indicators measured by PTRMS (proton-transfer ionization mass spectrometry) differed between the two aircraft missions. The SEAC4RS mission acquired suitably complete PTRMS-derived datasets at a once-per-minute rate, and this defined the data interval used for both datasets. Additionally, in SEAC4RS CO was measured only by (less frequent) can samples for the first flights prior to the Rim Fire of 26August, and $CH_4$ was sampled only by cans for all flights. These are important species: CO is the most commonly used tracer for fire plumes because of its favorable plume-to-background concentration ratio and readily available measurement instrumentation. It is also used to define the MCE in much of the biomass-burning literature (Yokelson et al., 1996; Jaffe and Widger, 2011). Consequently, SEAC4RS imposed additional difficulties and processing. However, we judged it important to include SEAC4RS in a combined analysis to broaden the fire chemistries analyzed, as the Rim Fire was exceptionally large, hot, and well sampled.

In order to provide a suitably complete dataset for SEAC4RS, we used the can samples to infer likely concentrations at one-minute intervals of key species, i.e., $CH_4$ for all flights and CO for the first few flights, using available can samples at slightly lower frequency. The **R** package for multiple imputations by chained equations (*mice()*) was employed, using the whole data period, but filling in observations with missing data. (Our assessment of the effect of imputation was informal and is reviewed again below.) It was highly desirable to include the imputed concentrations of methane, since it is commonly measured and appears to be a prominent signal of different types of "fire chemistry", i.e., enhanced emission of reduced species; methanol and acetone are often correlated with $CH_4$ and give support to this idea.

The use of imputation seemed justified by three observations: (1) Checks made when both LAS and GC data were available suggested agreement. In an early period, missing tunable-laser absorption spectrometry data for both CO and $CH_4$, some periods did not pass this test and all observations from this period were deleted. (2) The use of regression in both *mice()* and succeeding emission ratio analyses suggested that when observations were filled in, very little *information* was added, i.e., if the technique allowed missing observations, the results would be extremely similar. Specifically, $CH_4$ instances filled in using other variables do not contain any additional information. (We will describe evidence in the results later, in Section 7 and Figure 4.) (3) Comparisons of 10-second and 1-minute averages for the more detailed ARCTAS dataset (not reported here) suggested that the essential variability had been captured by 60-second data. We surmised that 30-second averages might have captured more. We are unsure how averaging affects difference-based methods.

The selection of fire plumes required some care. While $CH_3CN$ is a highly specific tracer of fires (Singh et al. 2012), detailed analysis suggests that it is not the best quantitative tracer. (Further analysis suggested that $CH_3CN$ has variable EFs, so it signals fires well, but does not quantify $C_{burn}$ adequately.) Plumes were characterized by levels of $CH_3CN$ above 0.225 ppb, over four times the assumed background of 0.054 ppb. Since some plumes are known to be quite low in gas-phase emissions, a few samples with lower $CH_3CN$ mixing ratios, but with $b_{scat} > 2 \times 10^{-2}$ were allowed in. Plots of $CH_3CN$ vs $b_{scat}$ suggested that a linear combination of the two minimal conditions clearly separated a population of forest fire plumes from other high-particulate situations.

There were forest-fire plumes for which urban sources of CO and other fire tracers made attribution and quantification problematic, and so a further test based on CO was applied to exclude urban samples, using CO vs. $CO_2$ plots for the years 2008 and 2013 separately (Figure 2).





We used a $\Delta CO/\Delta CO_2$ ratio of $<33 \times 10^{-3}$ to exclude plumes with excessive urban contamina-
tion. The figure suggests that some plumes with modest levels of urban influence remained and a
few genuinely uncertain situations were excluded where fire might still have been dominant.
Species with sources other than biomass burning and with lifetimes sufficiently long to allow re-
gional mixing can pose difficulties somewhat similar to $CO_2$ variability, with solutions suggested
in Section 8.2 and Chatfield and Andreae (2019). We noted some localized observations of per-
plexing, consistently negative $\Delta CH_4/\Delta C_{tot}$ relationships in the ARCTAS data (but not other spe-
cies) and removed these observation instances. Such relationships were found close to seaports
or oil-producing regions.

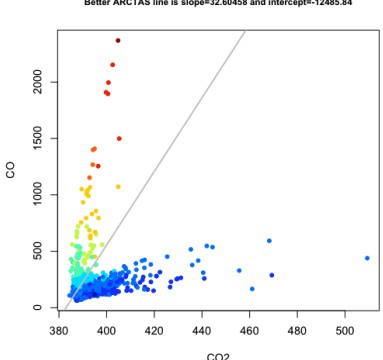

**Figure 2. Urban and forest fire plumes are separable by the ratio of CO to CO₂. Colors indicate a relative measure of**
**CH₃CN above background, from blue (lowest, ~0.1 ppb) to red (highest, ~6 ppb) values.**

## 3.   Observed behavior of $C_{tot}$ in fire plumes – Properties of tracers

Recall that in this work we wish to advance the use of multiple tracers to estimate EnRs and EFs
in more detail and also in more circumstances, e.g., where MCE is difficult to estimate. The rela-
tion of fire emissions to observed $C_{tot}$ t to $C_{burn}$, can be simple or complex, depending on how the
history of non-fire CO and $CO_2$ entrained into air parcels affects $C_{tot}$. Figure 3 describes two
flights in which plume encounters show how the interpretation of $C_{tot} = C_{burn} + C_{bkgd}$ can give ei-
ther clear or misleading descriptions of EnRs. Figure 3a shows the locations of the plume sam-
ples observed during SEAC4RS Flight 10 over Montana between 2 and 4.5 km MSL. Local to-
pography ranged from 1 to 2 km elevation. This data set included samples from the very intense
plume of the Rim Fire (discussed later), collected far downwind. Figure 3b shows ARCTAS
Flight 14, which was over the Coastal Mountains of Northern California at 0.5 to 1.5 km altitude,
with topography from 0 to 1 km.
Figure 4a gives the time series of fire indicators from Flight 10. The fire tracers CO and $b_{scat}$
appear generally well correlated with $C_{tot}$. This correlation is seen in Figure 4b-c. Colors from
blue to red give a key to sampling times. The lines connecting the adjacent plume samples sug-
gest two or perhaps three linear patterns pointing back to a no-fire background of $C_{tot} \sim 392.5$
and 394.5 ppm. Simple differences would suggest similar EnRs (slopes). A set of points near the
horizontal axis may suggest a pattern with very low EnRs. These points and those on the upper-
most suggested line (pointing to ~ 392.5) occur in the middle of sampling, just after 11 LT. Pat-
terns of variation related to $b_{scat}$ (550 nm, Figure 4c) are very similar to those of CO (Figure 4b).
Separated plumes encountered at 10:45, 11:20, and after 12:00 LT suggest very similar slopes.

Whereas the SEAC4RS data (Figure 4a-c) suggest expected behavior, the ARCTAS measurements (Figure 4d-f) show that $C_{tot}$ variations, likely due to $C_{bkgd}$ variability, can greatly complicate the attempt to estimate EnRs. The large orange dots in Figure 4d distinguish the plume

points *selected* (based on our plume tracers) from adjacent non-plume measurements made in the flight. The very first samples plotted and those after about 13:35 LT have very clean tracer levels; those between 13:03 and 13:15 LT were not selected plume points, but the tracers do indicate some fire influence. In this case, the trace of $C_{tot}$ does not reflect fire influence well at all. Both fire tracers shown in Figure 4c-d show wildly varying relationships to $C_{tot}$, but are remarka-

bly similar in that relationship.

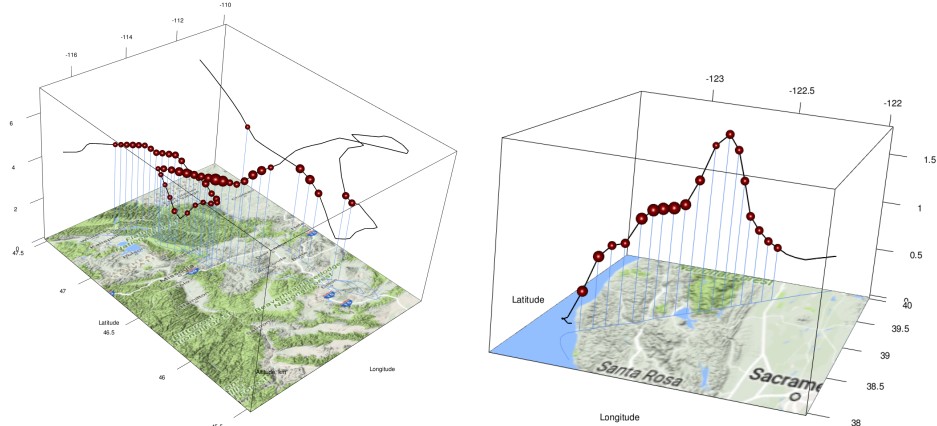

**Figure 3. Flight paths and locations of plumes for two fire samples. (a) Sampling over Montana during SEAC4RS Flight 10. (b) Sampling in a cross-mountain transect along the Northern California Coast. The size of the spheres indicates the relative amount of biomass burning contribution. Numerical values of the contribution are in a later figure, Figure 8. Some of the information was converted from GoogleMaps® using the R programing language.**

Without more analysis, either very different EnRs or different assignments of $C_{bkgd}$ during the history of the entraining fire plume for different sampling periods in the plume could explain the relationships graphed. Detailed and often sudden variations in $C_{bkgd}$ would seem to be to

blame. (The blue dashes in Figure 4a and Figure 4d attempt to suggest time traces of reasonable background levels at the times of our identified plume points. These are plotted for convenience here and are only justified later.) Note the observations of rapid changes in the successive arrows showing the $C_{tot}$-tracer relationship for both tracers, but the remaining mutual resemblance of the tracks do suggest a broadly useful approach to estimation of EnRs.





**Figure 4. (a) Variations of CO₂+CO and tracers showing complexities of fire plumes and background air. (b) Time-line of CO₂+CO and CO in SEAC4RS Flight 10 (Montana) samples. MERET analysis shown in a subsequent figure. (c) Variations of CO₂+CO and tracers in ARCTAS Flight 14 (Northern California).**





Figure 5 gives a general description of the dilution process, showing by the size of cubes how a mole of near-flame air is diluted by non-fire material as entrainment occurs. (The boxes shown suggest volumes, but lofting adiabatically changes volume. Discussion in terms of moles simplifies the discussion of mixing ratios and EnRs.) The figure is based on observations of plume size and plume dilution during rise followed by largely horizontal dilution downwind (Lareau et al., 2017; Hanna et al., 1982), which are consistent with mixing ratios measured in this dataset and near-fire $CO_2$ concentrations of $1–2.5 \times 10^4$ ppm. The sizes are meant to be suggestive, but we found that they give a valuable frame of discussion of all lofted forest fire plumes. For example, the sequence of increasingly large boxes emphasizes that the relative effect of entrained non-fire air is largest near the flames, but the absolute effect of entrained air on the composition of an observed parcel is often largest close to the parcel at its point of sampling. The following section gives a framework illustrating effects of emissions and entrainment on EnRs. (Parenthetically, the box volumes of parcels at different altitudes are similar to the mole amounts shown; to be consistent with adiabatic rise of parcels, volumes should be about ~11%-14% larger in linear dimension for most plume tops.)

## 4. Theory: expanding plume for several species

### 4.1. A general relationship

Using Figure 5 as a guide, consider a parcel originating at a time $t_1$ containing $v = v_1$ moles, that expands with an exponential relative rate $r_v = v^{-1}(dv/dt)$. (For our illustrative examples and to rationalize the MERET method, we need not start at the flame. We suggest a reasonable starting point described below.) This rate of expansion $r_v(t)$ of the molar volume varies considerably over time and fires are expected to have different magnitudes. Then molar mixing ratios will evolve with a law

$$\frac{dx}{dt} = -\frac{1}{v}\frac{dv}{dt}(x - x^E) \tag{7a}$$

$$\frac{dy_j}{dt} = -\frac{1}{v}\frac{dv}{dt}(y_j - y_j^E) \tag{7b}$$

where $x^E$ is the mixing ratio of entraining $C_{tot}$ and $y_j^E$ is the entraining background mixing ratio of fire tracer species or property $j$. The effect of volume addition is captured by $v(t)$ which varies with time and expansion. The use of the relative rate $v(t)$ does not require that the dilution is exactly exponential, but does make the algebra somewhat simpler.

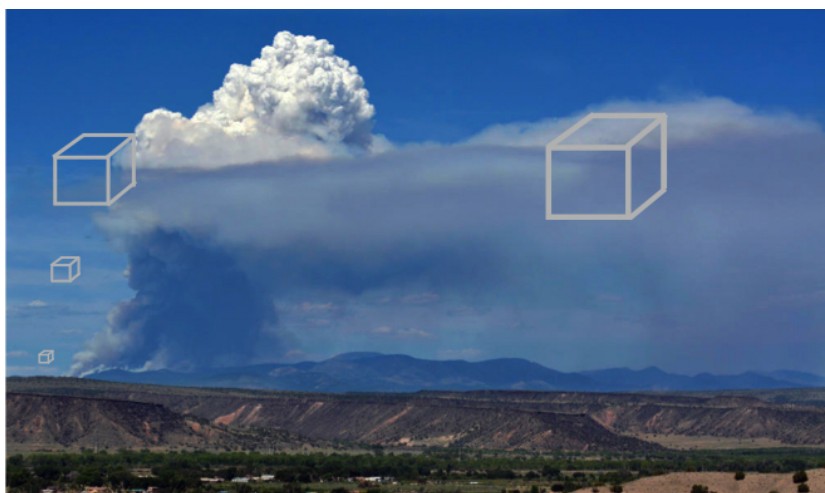

**Figure 5.** Inflow of air into an expanding fire plume; a likely near-fire aircraft sampling location would be near the cube on the upper right. Cubes are shown with three-d sizes proportional to the number of moles of entrained air. These may be considered volumes of air adjusted downward to compensate for the adiabatic expansion that rising plumes undergo. The smallest cube is taken to be near the flames, at roughly the point where fire emissions transition from mixing to entraining background air. Exact placement of this cube is not important to the analysis of entrainment, expansion, and tracer mixing ratio. Successively larger cubes have volumes roughly in the ratio of 1, 40 (partially raised), 140 (near neutral buoyancy level); sizes are consistent with a buoyant fire plume (Lareau et al., 2017). The rightmost cube has a ratio to the first of 400, consistent with horizontal Gaussian dispersion during travel downwind. See text for more details.

A general formulation of the expanding plume model consistent with the description appears to suggest formidable complexities. Using $\tau$ to describe the integration through time of an expanding parcel,

$$y_j(t_{Sample}) = \int_0^{t_{Sample}} -v(\tau)\left(y_j(\tau) - y_j^E(\tau)\right)d\tau + y_j(t=0) \qquad (8)$$

with a similar equation for $C_{tot}$, now generally referred to as $x(t_{Sample})$, involving $x(\tau)$ and $x^E(\tau)$ where $x(t=0)$ and $y_j(t=0)$. These are determined by the $C_{burn}$ from the fuel consumed and the tracer compounds released at the same time, as well as background concentrations, $C_{bkgd}$, and pre-flame backgrounds of the tracer $y_j^E$. We leave aside as a separate problem of a fire-burning model the complexities of the actual flame and its incorporation of additional air up until the point that entrainment of non-burning air becomes dominant.

Our approach is to avoid describing the complete history of $v(\tau)$ and any complex variation of $x^E(\tau)$ and $y_j^E(\tau)$. These would require a complete description of air along the parcel trajectory and the turbulent physics of entrainment. We accomplish this by concentrating on the fact that the entrainment process affects both $x(\tau)$ and all the $y_j(\tau)$ in the same proportions. This is a single-parcel description ignoring complexities of the rest of the plume. For convenience of discussion, we stipulate that the environmental air entrained has $x^E(\tau)$ and $y_j^E(\tau)$ constant over long periods. For example, it is constant in the mixed layer surrounding the fire flames and initial plume and then again in the subsequent regions often in the free troposphere. Conceptually there may be several regions which contribute; the exact history is lost. Our idea is that regression analysis allows us to infer a characteristic sum of effects which is described by a single quantity. The analysis can only be as complex as the number of our measured quantities allows.





A parenthetical note: it is natural to ask where this time/molar-expansion integration should
start, naming it as $v = v_1$. A reasonable start location is *when the fire plume parcels begin en-*
*training predominantly environmental air*, not other fresh emissions. The plume that is character-
ized by this expansion-period analysis is then that mixture over space and time of all detailed
variations in emissions before this transition. We remark that emissions from the very hottest
flaming combustion in a fire front are likely mixed with neighboring fumes from less vigorous
combustion. The hottest regions seem likely never to be directly sampled. Their relevance to all
downwind processing and effects is only as part of a mixture. In our dataset, the very hottest
burns, MCE > 0.97, were very rarely sampled. We speculate that values of $x$ and $y_{CO}$, which rep-
resent the MCE during true flaming combustion, may typically be confined to a region very close
to the fire, which is measurable in the laboratory but rarely in the field.

Returning to the differential-equation view of the simple expanding plume model suggests a
method for estimating the most important parameters. Solving each of the equations for the ex-
pansion rate and equating the expressions we obtain a form that eliminates the details of entrain-
ment and emphasizes proportionality. We recommend the reader refer back to Table 2, Table of
Symbols during the discussion of theory and then estimation details.

$$\frac{1}{(y_j - y_j^E)}\frac{dy_j}{dt} = -\frac{1}{v}\frac{dv}{dt} = \frac{1}{(x - x^E)}\frac{dx}{dt} \tag{9}$$

Since $dy_j^E/dt = 0 = dx^E/dt$, we get

$$\ln(y_j - y_j^E) = \ln(x - x^E) + C_j \tag{10}$$

$$(y_j - y_j^E) = a_j(x - x^E) \quad a_j = \exp(C_j) \tag{11}$$

Note that by our definitions, the reasonable interpretation of $C_j$ is the EnR $a_j$ for species $j$.
Consider two observations of the same plume, each made at differing degrees of dilution $v$. For
convenience, these are labeled $b$ and $a$, mnemonically "before" and "after." Temporally they
could be nearly coincident or $b$ after $a$. For these observations,

$$(y_{aj} - y_{bj}) - (y_j^{Ea} - y_j^{Eb}) = a_j(x_b - x_a) - a_j(x^{Ea} - x^{Eb}) \tag{12}$$

As a side note, in simple situations (e.g., observations in a plume with same environment but
with differing dilution, perhaps, upwind and downwind), the equation reduces to

$$(y_{aj} - y_{bj}) = a_j(x_a - x_b) \tag{13}$$

for any two observational instances, $a$ and $b$, in the same plume. *provided* we know that $x^E$ and
all the $y_j^E$ remain constant. Questions regarding constants of integration and original concentra-
tions at $a$ are handled in the regression procedure, Section 5, below. More generally we need
$|x^{Ea} - x^{Eb}| \ll |x_a - x_b|$ and $|y_j^{Ea} - y_j^{Eb}| \ll |y_{aj} - y_{bj}|$.

This formula is the basis for the NEMR technique mentioned above, with $j$ = CO playing a
particularly important role. The inequality restrictions should be evaluated for an EnR to be a
candidate for an ER and then an EF. In some cases, the background values, $x^{Ea}, y_j^{Ea}$, might be
estimated from measurements made outside the plume. It can be somewhat more difficult to esti-
mate $x^{Eb}, y_j^{Eb}$ upwind, especially for for air entraining into the fire plume at its source. A plume
may also entrain air from various backgrounds and at various times during lofting and spread.
That is, the history of entrainment may well be more complex than two conditions, "$a$" and "$b$",
and the number of situations where we may estimate EnRs and then EFs is greatly limited. This
important realization was described by Yokelson et al. (2013). The NEMR method can deal with
most differences in $x^{Eb}$ but not $y_j^{Eb}$ for most tracers, and the quantity $y_{CO}^{Eb}$ for carbon monoxide
must be well sampled and well understood.





**Table 2. Table of Symbols**

| Symbol | Signifies | Observed, Estimated, or Hypothetical |
|---|---|---|
| $C_{\text{tot}}$ | CO$_2$ + CO (+ other carbon, ignored), ppm | O |
| $C_{\text{burn}}$ | CO$_2$ + CO (+ other carbon, ignored) emitted from fire, present downwind, in plume sample, to be estimated as $(x_i - x_i^0)$) | E |
| $C_{\text{bkgd}}$ | CO$_2$ + CO **not** emitted from fire, present downwind in plume sample, thought of as a mixture of $C_{\text{tot}}$ entrained at various stages in plume expansion and rise, to be estimated as $\hat{x}_i^0$ or casually as $x_i^0$. This is not necessarily air surrounding the plume sample! | E  H* |
| $i$ | Sample sequence number, organized for convenience by time of the sample | O |
| $j$ | Tracer number for regression, here 1 to 10 or "CO" or "$b_{\text{Scat}}$", … . After regression estimation is completed, $j$ may be used similarly to specify any fire emission concentration or response, e.g. "propene" or "O$_3$" | O |
| $a$ or $b$ | Location at beginning or end of a period of idealized plume development and entrainment | H* |
| $x_i$ | $C_{\text{tot}}$ = CO$_2$ + CO at a plume sample location and time, used in algebraic development, shown on $x$ axis | O |
| $y_{ij}$ | Tracer concentration, e.g. toluene, $b_{\text{Scat}}$, at plume location and time. | O |
| $x^{\text{E}}$ or $x_A^{\text{E}}$ | Environmental air "background" $C_{\text{tot}}$ concentrations existing at location A, e.g. beginning of our integration of the plume expansion equation. B signifies condition at the end of calculation | H* |
| $y^{\text{E}}$ or $y_A^{\text{E}}$ | Environmental air "background" concentrations of fire tracers. | H |
| $x_i^0$ | *Equivalent* background concentration characterizing composition at plume sample location and time | E |
| $y_{ij}^0$ | Background concentration of tracer $j$. Typically estimated as a minimum value from observed probability density function for samples in a particular flight intensive, especially non-plume samples without signals of stratospheric air. | E |
| $a_j$ | Slope relationship of $y_{ij}$ to $x_i$ for species $j$, typically species $j$ under burning conditions for a "fire type" that is common for all species at instance (time) $i$. These slopes then transform to EnRs and ERs. | E |
| $c_j$ | Intercept relationship of $y_{ij}$ to $x_i$ | E |
| $\hat{x}_{ij}^0$ | One of several (10) estimates of $x_i^0$ based on tracer $j$ and fire type assigned by clustering for observation instance $i$. | E |
| $\hat{x}_i^0$ | The median of the $\hat{x}_{ij}^0$ over all the tracers $j$ | E |
| $(x_i - x_i^0)$ | The estimate of $C_{\text{burn}}$ for instance $i$ in theoretical development and then from regression. Regression results are properly $(x_i - \hat{x}_i^0)$ | E |

*Note* that symbols may transition from Hypothetical to Estimated as the discussion develops.

440

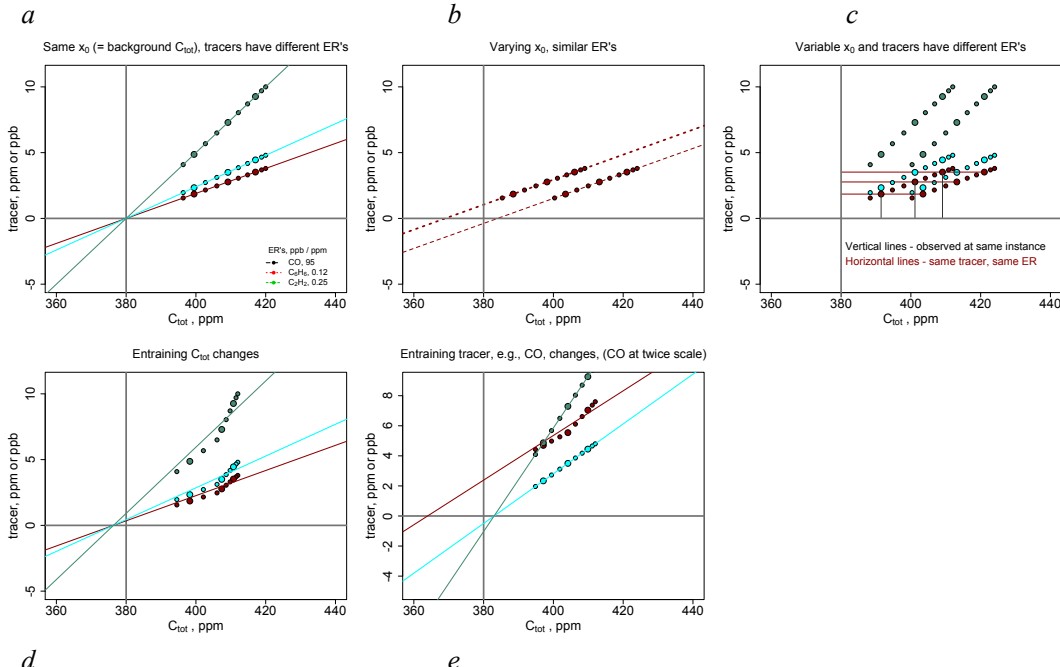

Figure 6. (a) Simulated dilution of three different fire tracers, EnRs as shown, and with environmental $x^E$ of 380 ppm. These are nominally CO, benzene, and ethylene. Background concentrations of the tracers have been subtracted. Larger dots highlight equivalent degrees of dilution. (b) Simulated dilution of one tracer, nominally CO above background, with different background $x^E$. Backgrounds are illustrated by the x intercepts. (c) Simulation of three tracers, varying EnRs and varying backgrounds (deducible from the x intercepts). Thin lines emphasize similar constant y values with different backgrounds, and constant $x^0$ values with varying EnRs. (d) Simulations like (a), but with a change of the $x^E$ entrained $x^0 = C_{burn}$ background from 395 ppm to 375 ppm at the time of the 8th dilution step. A single applicable background value ~378 ppm is an appropriate linear interpolation between 395 and 375 ppm. (e) Simulation where background $x^E$ remains constant, but background CO changes by 1.5 ppb during the time (CO drawn at twice height for visibility).

## 4.2. Examples showing robustness of computations of idealized $C_{tot}$

We used this approach to produce the following concrete examples of increasing complexity. They illustrate the origin of the features seen in Figure 6 in terms of this simple plume dilution model. They helped motivate our solution techniques and indicate methods of analysis of individual plumes. These examples indicate possible limitations, but they also indicate a comforting averaging behavior of the linear differential equations as they describe our solutions. These uniformities and deviations also showed up in the analyses that we develop below. The examples also give some quantitative feel on the effects of deviations from the simplest hypothesis, e.g., $x^E$ and $y_j^E$ remain constant through time. Figure 6 shows calculations describing behaviors of $x$ and the $y_j$ in several plausible situations. Each graph represents the development of plume mixing ratios for a period of plume doubling, similar to the analysis time chosen in Poppe et al. (1998), following their equations 7 and 8. The dots show equal increments of plume expansion. Most parameters defining the equations may be read from the graphs themselves. Each initial concentration is shown by the points to the upper right of the line, i.e., the points with maximum $x = C_{burn}$ and $y_j$ = tracer concentration for each case considered.



Figure 6 (a) illustrates a plume history for $x^E = 380$ ppm and EnRs with respect to airborne $C_{tot}$ of 95 x $10^{-3}$ ppm/ppm, 12 x $10^{-6}$ ppm/ppm, and 25 x $10^{-6}$ ppm/ppm, which are reasonable values for carbon monoxide (in ppb), benzene, and ethylene (in ppt). In the figures, focus attention on the relative behavior of the tracers. It is assumed that there are no consequential production or destruction reactions and also that there is a constant background tracer concentration,

which has been subtracted. The individual plots show situations of increasing complexity. Figure 6(a) shows the dilution behavior of the three species. A constant dilution rate is plotted; note that varying dilution rate changes the spacing of the dots but not the linear pattern. Larger dots highlight equivalent dilution of tracer and $x = C_{tot}$ as would be observed in hypothesized discrete airplane samples. Figure 6b illustrates the dilution of CO in environments with differing entrained

$x^E$. In Figure 6a, the larger dots align vertically, in Figure 6b, horizontally. Figure 6c illustrates the situation where both EnRs and backgrounds vary; the thin lines emphasize independent aspects of EnR and $x^E$. The points on the $x$ axis where (excess) tracer is zero are important to our estimation technique, more important than $x^E$. Estimation of $x^E$ utilizes data on the vertical lines, while EnRs utilize information from both the vertical and horizontal lines. Statistically

speaking, the problem of estimation of both backgrounds and EnRs illustrates simultaneous effects that are "separable". The reader may wish to extend the analysis to a large sequence of changes in entrained concentrations and note the essential linearity of this aspect of the formulation and that the solution expresses an appropriate averaging effect. We remark that the near uniqueness of the solutions obtained below (making small allowances for measurement error)

will underline the robustness of the solutions.

However, the effect of uniform variations in background tracer concentrations $y_{ij}^0$ is not completely solved in this work. $y_{ij}^0$ can be estimated by examining the lowest values $y_{ij}$ in non-plume air; it is best to exclude values that represent exotic air (e.g., stratospheric air) or possible measurement problems at very low mixing ratios (e.g. negative values). Restricting attention to larger

values of $x_i$ and $y_{ij}$ greatly ameliorates problems arising from $y_{ij}^0$; making comparisons from one variable $j$ to another in very clean parcels may also help.

## 5. Theory: A regression relationship for EnRs

Let us consider more broadly the equations that provide a basis for statistical estimation. For current purposes of explanation, we make the seemingly large assumption that points from different plumes have similar properties at the same degree of dilution and may be compared. That

is, the $a_j$ are consistent for all plumes. Effects of varying $x^E$ and $y_j^E$ between the plumes may be largely taken out by regression; that is our current concern. Later, we will describe our approach to address possible variations in the EnR relationships $a_j$ for parcels in the same or different plumes.

The basis of MERET considers the unmeasured extreme where $y_{aj} = 0$. To begin with, we consider the situation where (i) the emissions relationships $a_j$ are constant for all observations and (ii) background values of the tracers are small enough in a relative sense, i.e., $\left| y_j^{Ea} - y_j^{Eb} \right| \ll \left| y_{aj} - y_{bj} \right|$. That condition is common for many species that have loss timescales of less than a month and/or have small non-fire sources. Each of these restrictions can eventually be relaxed. In

this case

$$y_a = a_j \left( x_a - \{ x_b - x^{Eb} + x^{Ea} \} \right) + \{ y_j^{Ea} - y_j^{Eb} \} \qquad (14)$$





$$y_a = a_j \left( x_a - \{x_b - x^{Eb} + x^{Ea}\} \right), \text{ for } \left| y_j^{Ea} - y_j^{Eb} \right| \ll \left| y_{aj} - y_{bj} \right| \tag{14a}$$

notice that terms within braces can be estimated by regression as sums, varying by the situation $b$. What if the values of these terms change discretely in time, for example as a plume leaves a daytime mixed layer, or distinct upper-air plumes are encountered? Simple algebra with linear formulas suggests that estimates of the terms in braces change discretely. Gradual changes of entrained mixing ratios of course imply a continuity condition on these terms.

We return to the illustrative dilution behaviors described in Figure 6. Figure 6d describes a sudden change of background $x^E$ by 20 ppm, $x^{Eb}$ to $x^{Ea}$, midway in the expansion/dilution; at this stage of plume evolution, 20 ppm is about four times larger than typical fire contributions to $C_{\text{tot}}$. Estimates of $x_i^0$ from a few samples along these lines (without knowledge of the time of change) would be intermediate. Equation 14a suggests that the EnR estimate need not be affected. We reiterate the suggestion that if the reader makes sample calculations, it becomes clear that the estimates respond in a linear manner. Figure 6e shows a very contrasting behavior, when there is a sudden change in concentration of entraining tracer (CO) during plume dilution, a change of 1.5 ppm. In comparison, the addition of CO by burning at the start of the interval is ~3.5 ppm. We may distinguish this as $\hat{x}_{ij}^0$, where the $j = $ CO and the hat indicates an estimate. This graph also suggests that if there are more than three tracers (we use 10), then the median of all the estimates $\hat{x}_{ij}^0$ provides safety against a variable and incompletely described background. The median is not affected by undetected changes in background for one tracer species. Change in background of a tracer compared to observed change due to fire is critical in determining its usefulness in determining a useful estimate of the appropriate background $x^0$ as the well as the quality of its EnR of a tracer. Methane in particular have long atmospheric lifetime and several sources; consequently, it can exhibit variations that are more than 10% of the fire emission contribution for well dispersed plumes.

The preceding discussion suggests that we may use the specialized least-squares technique,

$$y_{ij} = a_j(x_i - x_i^0) + c_j + e_{ij} \tag{15}$$

where $x_i^0$ expresses all the terms, and any other corrections not proportional to $a_j$. If $-x^{Eb} + x^{Ea}$ remains constant in the samples considered, $x_i^0$ can still be estimated. A term $c_j$ is included; it should be zero, but a significant non-zero estimated value can alert us to inadequacies in our assumptions. We may call $x_i^0$ an "effective background". However, it is not a specific background, but actually summarizes the whole effect of changes in $C_{\text{bkgd}}$ and also the degree of dilution. This means that the regression can synthesize information from not just one well-characterized plume but rather a variety of plumes with the same $a_j$ behavior.

Figure 7 illustrates the use of regression using the ideas developed in Figure 6. Using the same formulas as above, we depict observations made of CO, $C_6H_6$, and $C_2H_2$ made at three instances (times). The three tracers determine a value $x^0$, and given that information, the three tracer enhancements and therefor tracer EnRs are determinable. This simulation assumes no error in the measurements of $CO_2$+CO or the tracers, and assumes no variation in the EnRs, so values are determined perfectly. It is useful for our following discussion to examine the uniqueness of the solutions for two tracers and two plume observation sample instances, and then two tracers and three instances. In the case of two samples, with $m$ samples and $n$ slopes (EnRs), we need to estimate $x_1, x_2, a_1, a_2,$ and $x_0$ using only $y_{11}, y_{12}, y_{21},$ and $y_{22}$, there are not enough measured variables to determine a unique solution, $N_{\text{Tracer}} + N_{\text{Instance}} + 1 > N_{\text{Tracer}} \bullet N_{\text{Instance}}$, $viz.$, $5 > 4$. However, if there are three tracers, $N_{\text{Tracer}} \bullet N_{\text{Instance}} > N_{\text{Tracer}} + N_{\text{Instance}} + 1$, and we get a solution.

Any measurement error would give conflicting solutions. Using linear regression with larger $m$
and $n$, one obtains solutions and increasingly accurate error estimates.

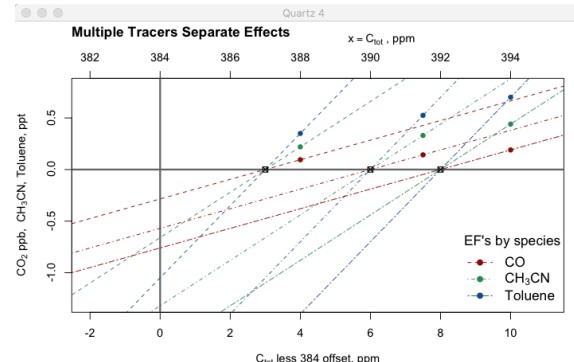

**Figure 7. Multiple tracers allow solution for an equivalent background $x^0$, illustrated by an idealized example largely replicating the conditions of fire plume sampling above an Amazonian mixed layer as described by Yokelson et al. (2013). The fact that the various colored lines associated with each x meet at the value with x-coordinate $x^0$ and $y$ coordinate 0 represents the estimation that precisely solves equation (5) above. If we had included error in observations or variation in EnRs, there would be uncertainties in the positions of $x^0$ and the slopes. See the text for details.**

## 6.  Methodology:

### 6.1. Finding the CO₂ + CO background

The use of tracers with backgrounds removed and then scaled to a common mean establishes a
well-conditioned matrix problem and easier analysis of sensitivity effects. Let us proceed with
the regression and begin to address some complications that arise. The mathematical problem we
must solve is equation 15:

$$y_{ij} - y_j^0 = a_j(x_i - x_i^0) + c_j + e_{ij} \qquad (15)$$

In the following development in this section, we will work with $y_{ij}$ above background, i.e., set
$y_j^0 \leftarrow (y_{ij} - y_j^0)$, at least until it becomes important to consider background values of the tracer
variables again.

We must attempt to fit $y_{ij}$ for each species $j$ (1 to $N_j$ ) and for each measurement instance $i$ (1
to $N_j$). The term $e_{ij}$ describes the error. The mixed-effects formulations expresses the idea that
all the fire-tracer variables $y_{ij}$ must "point back" to a zero-point, where no $C$ was added by the
burn, $C_{\text{burn}} \equiv x_i - x_i^0 = 0$ , but that each instance $i$ may have a different zero-point. A regression
formula should solve this in some way. However, it also shows why the emission factor problem
is difficult: the equation is non-linear in the multiple regression sense: i.e., we must include $a_j x_i^0$
as a regression term. Commonly, regression fits provide a $y$-intercept, i.e., the value at $x = 0$.
Here we have an $x$-intercept to estimate, i.e., the concentration of the reference species at zero
added fire emissions, and so the problem becomes non-linear in the regression sense. The prob-
lem is also a mixed-random-effects model: we must estimate the effects of two "random" varia-
bles affecting the $a_j$ and the $x_i^0$. In summary, we have a non-linear random effects model (Pin-
heiro and Bates, 2000), which requires specialized techniques.



Why not simply reverse the problem and seek $x$ as a function of $y$?

$$x_i = a_j\, y_{ij} - x_i^0 + e_{ij} \qquad (16)$$

where the $x_i^0$ are estimated with a regression model (specifically a fixed-effects model). The dif­ficulty is that the trivial solution $x_i = -x_i^0$ fits perfectly and was hard for us to avoid even when we attempted to restrict the solution with a non-linear solver. Is it not easier to convert y inter­cepts into x-intercepts? This appears more productive and should appeal to those not familiar with using a non-linear solver in this particular mode of non-linearity. We may write a regression equation with an intercept:

$$y_{ij} = a_j x_i + c_i^0 + e_{ij} \qquad (17)$$

where the $y$-intercepts, $c_i^0$, are estimated for each instance and the $e_{ij}$ are minimized by least squares. The $c_i^0$ may be thought to represent an error in $y_{ij}$ resulting from the use of $x_i$ with an incorrect baseline (the unknown $x_i^0$, an error that is constant for all species $j$. The **R** expression used for this was

```
main.lmer = lmer( y ~ x + (x - 1 | species.type) + ( 1 | id ) +
                  1
```

where `id` indicates the sequential observation number for the tracer species. The term `spe-cies.type` indicates the species number $j$, The word `type` signals a generalization described in the next section. The vertical lines indicate how factors are involved with variables, 1 indi­cates an intercept is to be described by a random effect, and (`x - 1 | species.type`) in­dicates that a slope that multiplies $x$ is to be estimated, indexed by `species.type`. "No inter­cept estimated" is signaled by –1. The regression generates a set of fitted $y$ values which we may call $\hat{y}_{ij}$. to estimate $x_i^0$. A simple way is to use the slopes estimates $\hat{a}_j$ of $a_j$ by regression to find the several estimates $\hat{x}_{ij}^0$ provided by

$$\hat{x}_{ij}^0 = (\hat{y}_{ij} - y_{ij}^0)/\hat{a}_j \qquad (18)$$

(equivalently, use $y_{ij} \leftarrow y_{ij} - y_j^0$), so that $\hat{x}_{ij}^0 = \hat{y}_{ij}/\hat{a}_j$, where we will (currently) assume that $y_{ij}^0$ has been satisfactorily estimated as $y_j^0$ as a single background value suitable for all samples, as described before. The use of a single $\hat{a}_j$ for all observation instances of the same species and NMF-assigned cluster is a fairly large constraint on the resulting estimate. The use of a larger number of cluster classes will allow more ability to follow the EnR actually characteristic of the observation, but at the cost of parsimony and sensitivity to instrumental error for the species or property. We found that the conversion of y-intercepts to x-intercepts was more accurate when the values of nominal baseline C$_{\text{tot}}$ were adjusted so that the median value was 391 ppm. Essen­tially, this use of a constant baseline makes the numerics better conditioned: the difference of small quantities, not the difference of large quantities. This adjustment made the individual $\hat{x}_{ij}^0$ symmetric around $\hat{x}_i^0$ and also more closely spaced.
We then take

$$\hat{x}_i^0 = \underset{j}{\text{median}}\ \hat{x}_{ij}^0 \qquad (19)$$

The estimation of $\hat{x}_i^0$, now allows estimates of the incremental carbon liberated to the atmos­phere, $C_{\text{burn}} = x_i - \hat{x}_i^0$. We will drop the hat from $\hat{x}_i^0$ below, writing $x_i^0$ and $C_{\text{burn}} = (x_i - x_i^0)$ except when we wish to emphasize their nature as estimates.

Emission factors for individual tracer species may be obtained directly by adding fixed and random effects on slopes for each species and each observation, $\hat{a}_j$. An enhancement ratio for any concentration or property $y_j$ with a background, measured at time $i$ in the aircraft sam­pling can be obtained using the carbon-burned estimate



$$\text{EnR for } y_{ij} = \frac{(y_{ij} - y_j^0)}{(x_i - x_i^0)} = (y_{ij} - y_j^0)/(C_{\text{burn}})_i \tag{20}$$

To repeat, the variable $y_{ij}$ now stands for any property for which we seek an EnR, for example ozone, which is not one of the ten indicator variables. $y_{ij}$ describes a non-fire-dependent background value. This ratio estimate is available for all tracers, and is preferred over a similar slope variable $\hat{a}_j$ used to estimate $x_i^0$ in the in equation 18 above.

Formulas for the statistics of ratio quantities with uncertainties in numerator and denominator
can be theoretically complex, so we simply computed error estimates by simulation using computed Bernoulli trials. One thousand samples of normal distributions were calculated each for the numerator and the denominator, using their uncertainties as $1\sigma$ values. Then the ratios of the first numerator and first denominator normal deviate sample, the second, … to the 1000th normal deviate sample for numerator divided by the 1000th normal deviate sample were calculated, and
the distribution of ratios summarized. For the numerator, the measured value and the suggested standard deviation (typically a percentage ratio) provided the parameters for the normal distribution. For the denominator, the mean was the $C_{\text{burn}}$ estimate, and the standard deviation was a value of 0.25 ppm, documented as the measurement error (precision + bias) of $CO_2$. (See Table 1). Uncertainties in the calculation of $\hat{x}_i^0$ were considered small and did not add to the dispersion
of the denominator, especially since it is clear that any additive biases contributing to the quoted uncertainty of $(CO_2+CO)$ cancel out. Sample calculations in the supplementary material suggest errors typically of magnitude 0.03 ppm due to variations in technique, and usually < 0.1 ppm. Additive errors should also cancel out for the numerator, since a background is subtracted. Indeed, some tracers like ethene appeared to have a negative background as determined from plots
and simple regression calculations of $y_{ij}$ on $(C_{\text{burn}})_i$. This is not unexpected, since these compounds are sampled into cans, where a small but self-limiting coating of the measured species on the can surfaces might cause such a negative offset, and yet the integrity of the can sample at larger values might be little affected.

### 6.2. Practicalities: variable EnRs

Equations 17–19 provide the basis of the MERET technique. There are however some details that increase its relevance and accuracy. First there is normalization. Common practice is to normalize all the tracer species $j$ with respect to the mean of all observations of species $j$, after subtracting a baseline. This allows each tracer to influence $y_{ij}$ equally. Assigning weights accomplishes the same purpose, but scaling allows better diagnostic graphs. In fact, the literature refer-
enced above emphasizes how informative $j = CO$ is, despite its relatively small variation in EnR or slope. Consequently, we give CO twice the weight of all the other species.

Secondly, we allow for a certain amount of true variation in the EnRs, expecting this to make equation 18 perform better. This is done by imagining that virtual species can be associated with "fire types" for example "flaming CO" or "smoldering CO" or "high-nitrogen-fuel $CH_3CN$". A
"fire-type" is a value for each observation that applies to all tracer species at that instance. It expresses commonalities between different mixes of burning emissions, commonalities that may be more frequently or less frequently expressed in any given plume, e.g., "smoldering-CO firetype". We might speculate on the nature of the fire-type, e.g. "smoldering" or "derived from nitrogen-rich fuel". However, we let the define these types, and so apply basic clustering tech-
niques. We used non-negative matrix factorization (NMF), but Mahalanobis clustering or other techniques seem to do as well. NMF is more fully described in a companion paper describing





patterns linking EnRs for several compounds. We used the **R** routine *nmf ()* with $k = 5$ components and the Brunet approach (Gaujoux, 2014). Since all fire tracers are correlated, such clustering characterizations are much better defined if based on a rough normalization to the fuel burned. We used a consensus variable, essentially a first principal component of the tracers, to act as a normalizing agent, playing the role logically played by $C_{burn} = (x_i - x_i^0)$. (This estimate cannot resolve situations in which the EnR for a tracer varies; also, we found it problematic to use regression to give it an accurate value of $C_{burn}$ in ppm.) Exact quantitative calibration of $C_{burn}$ in ppm is not required, just a relative scale. (Note: A certain degree of characterization of fire-types can be found in Chatfield and Andreae [2019], which explores striking relationships among the EnRs. In that work, we cluster the fire-types using a completed description of the estimated $(C_{burn})_i$ values, the estimates described below in Figure 8)

In a separate analysis effort, we sought timescales that could be inferred from the data, which could distinguish the relative age of burning emissions. At greater distances from the fire, there is both aerosol transformation and photochemical loss/production of species. Photochemical processing appeared easier to diagnose. We followed the ideas of Roberts et al. (1984), McKeen and Liu (1984), Parrish et al. (2007), and Warneke et al. (2013). The Parrish et al. presentation was most directly relevant. For considerations of these plume samples, a single origin strongly controlling mixing ratios made analysis simpler. Following Parrish's Equation 3, and using the symbols $E$ and $Y$ are the mixing ratios of ethEne, and ethYne, respectively,

$$\tau_{age}(OH) = -\frac{1}{k_E - k_Y}\left\{ln\frac{y_Y}{y_E} - ln\frac{ER_Y}{ER_E}\right\} \tag{21}$$

In view of this we constructed estimates for each instance $i$ of $\log_{10}(y_{i\,Y}/y_{i\,E})$ − Constant. The Constant can be estimated with similar results (a) so that the shortest times are about +15 minutes, or (b) from the highest observed values of $\log_{10}(y_{i\,Y}/y_{i\,E})$. The values of longer times are determined by the assumed value of [OH]. The references cited describe the fact that most $\tau_{age}(OH)$ observations have a contribution from mixing as well as photochemistry, but this has little affect the relative ages. In view of the uncertainty of the history of [OH] during transport, we simply graph the log of the ratios. Data analysis suggested that the assumed background mixing ratios of the species of ethyne and ethene were small. The supplementary material provides some more details and one estimate of the associated times.

### 6.3. Summary of the MERET method

Here is a summary of the MERET method as we currently propose it. It contains many steps, but we note that there are many processes that affect EnR estimation.

1. Select a dataset of plumes with sufficient mixing ratios of the tracer, $CH_3CN$, above 0.225 ppb and expectably minimum interfering signals from urban sources: $(CO–CO^{Backg})/(CO_2–CO_2^{Backg})$ ratio of $> 33 \times 10^{-3}$. Exact background levels are not critical, but should correspond to the year, season, and location sampled (examine plume and non-plume dataset). Plumes from agricultural burning and with detectable urban influence are thus excluded.

2. Select $N_{Trace}$ fire tracer datasets that can provide sufficient measurement instances (we suggest $N_{Instance} > 50$) for each of the tracers, preferably hundreds of points. Tracers must be measurable sufficiently frequently to populate the dataset.

3. Estimate background levels $y_j^E$ typically entrained into fire plumes. Plume values of tracers should be at least four times higher. For numerical reasons, select an offset $C_{baseline}$ for each plume (each day). This is a value fixed for each plume $I$ that is 2 ppm $CO_2+CO$ smaller





than the minimum plume value observed for that day (with enough samples, for that plume of > 4 points). Subtract this from the $x_i$ values but record it so as to add it to quantities like the estimated $x_i^0$ later.

4. Normalize the tracers above estimated backgrounds. This simplifies the regression estimation and its interpretation. All tracers tend to contribute equally. Weighting inversely proportional to the quantities could be used instead.

5. Roughly cluster plumes into $N_{\text{Types}}$ clusters by estimated EnRs (this can be applied recursively, after other steps). The value of $N_{\text{Types}}$ is observed to make little difference. Too small a value $N_{\text{Types}}$ can however lead to negative $\hat{y}_{ij}$. Here, five types were used. $N_{\text{Types}}$ should be at least one smaller than the number of fire tracers.

6. Make many estimates of $\hat{x}_{ij}^0$ and $\hat{a}_{j_j}$ using a mixed effects regression like

`main.lmer`, allowing random effects corresponding to species (or species and type of fire) and by instance. Calculate $\hat{x}_{ij}^0 = (\hat{y}_{ij} - y_{ij}^0) / \hat{a}_j$ from the fitted values and thus $(x_i - \hat{x}_{ij}^0)$. Although a single $x_i^0$ is sought, each tracer will give an estimate $\hat{x}_{ij}^0$ at each instance. Take the robust estimate of $x_i^0$ as the median of these $\hat{x}_{ij}^0$ at each instance $i$.

7. Use of an offset in calculations: We subtracted a baseline, $C_{\text{baseline}}$, a value determined as a constant for each flight and yielding a ~1 ppm offset. We found that this minimized *skewness* and *variance* in the $\hat{x}_{ij}^0$ estimates for each observation instance $i$. It is comforting that the effect of differing offsets on the values of the *median*, $x_i^0$ is small, < 1 ppm. One must not forget to add the offset back in the $C_{\text{baseline}}$ when reporting $x_i^0$!

8. Sharp positive and negative excursions of $x_i^0$ are seen near dramatic spikes in $x_i$. However, $(x_i - x_i^0)$ and consequently the EnRs are little affected. We can only speculate that small differences in the time averaging of $CO_2$ and the tracers due to the instruments may explain these.

Note also that the number of parameters $N_{\text{Trace}} \cdot N_{\text{Types}} + N_{\text{Instance}}$ for the mixed effects regression is $\ll N_{\text{Trace}} \cdot N_{\text{Instance}}$ so that the mixed effects regression is very over-determined.

It is possible to use this method recursively, making presumably better classifications of fire types. In our experience, while it is possible to make convergent, recursive characterizations of the appropriate $C_{\text{burn}}$ quantity, more appropriate clustering, and mixed-model *lmer()* estimates, the quantities $x_i^0$ and $(x_i - x_i^0)$ were almost unaffected by such care. If we had available fewer than 10 tracers, such recursion might be important. We will incrementally update documentation

of the code on https://github.com/RobertBChatfield/FireEmissionsEstimate/ .

### 6.4. Remark: Number of independent samples

A natural broader question is: "How well do these mean EnRs for a species represent the EnRs that might be measured in a large suite of significant forest fires in the Western US?" Clearly, this question can only be asked in the context of the sample provided by the two campaigns. In-

stances when the aircraft continued to sample smoke for many minutes could contain several types of plumes, as we will see illustrated for the Rim Fire Plume of August 2013. The use of 10-sec averages (if available) would not provide six times as much information about fire plumes as 60-sec averages over the same measurement run. We tried a simple, approximate quantification of "independent instances" available to us using a frequently used formulation by Trenberth

1984). This can also be seen as providing one answer to the question "How many effectively independent samples of $C_{\text{burn}}$ are there contributing to a mean, standard deviation, etc., of $C_{\text{burn}}, \ldots$



or of CO tracer?" That could be useful if instruments appeared to give imprecise measurements that required averaging. Trenberth assessed the correlation of successive observations by estimating an autoregressive AR(1) model with parameter $\phi$ and random error $\varepsilon_i$, i.e.,

estimate of $\xi_{i+1} = \phi\, \xi_i + \varepsilon_i$ . We applied this for $x_i - x^0$ $(= C_{bum})$ and several fire tracers $y_{ij}$, like toluene. Most contiguous sampling periods suggested around 0.6; this suggested Trenberth's "effective time between independent observations" as $\approx (1 + \phi)/(1 - \phi)$, about 4 minutes. Four minutes corresponds to about 15 km at lower-tropospheric airspeeds for the DC-8. The effect of this on the formal standard errors as described by a normal distribution was to increase

them by a factor of ~2. Roughly similar effects are expected for the empirical descriptions of EnR variability described below. Undoubtedly, for plumes within minutes of the source, the number of degrees of freedom corresponds more closely to the number of 1-min observations, but the number of such samples is low.

      Not surprisingly, residuals in regressions of CO against $C_{burn}$ are very little correlated. We

surmise that such low correlation gives confidence in the mathematical determination of the mean regression slope. However, it does not provide help in answering the larger question, that of relevance in new situations. The sequential samples of plumes may have features like non-stationarity and selection bias; we hope that these ideas suggest more sophisticated analyses of relevance, left to future work.

## 7.   Results: Estimation of $x_i^0$ and $C_{burn}$

The important results of the mixed model are the background $\hat{x}_i^0$, and even more importantly the incremental carbon liberated to the atmosphere, $C_{burn} = x_i - \hat{x}_i^0$. The background estimates of $\hat{x}_i^0$ for all samples and the contributing individual estimates $\hat{x}_{ij}^0$ are shown in Figure 9. The median $\hat{x}_i^0$ is shown as a thin black line. The colored circles in the legend identify how the tracer

species $j$ contribute an individual $\hat{x}_{ij}^0$. determining the median $\hat{x}_{ij}^0$.

      What are the uncertainties in the estimates of $\hat{x}_i^0$ and $C_{burn}$ we have made? The uncertainty in estimated carbon burned $(x_i - \hat{x}_i^0)$ plays an important role in the ultimate estimates, the emission factors. In this section, we will confine our exposition to this uncertainty for now. The graphs of $\hat{x}_i^0$ and $(x_i - \hat{x}_i^0)$ shown in Figure 8 provide a practical understanding of the uncertainty. Note

the continuity in $\hat{x}_i^0$; this important observation is described below.

      Traditional estimation of uncertainties for $(x_i - \hat{x}_i^0)$ is complex due to the several steps involved and the use of median estimates. The advisability of using the median estimator and its statistical properties have long been recognized (Laplace, 1774; Lawrence, 2013). This variety of uncertainty estimation may be useful as the MERET technique is refined. However, we expect

that the study of uncertainty depends more on evaluating sources of true variability in the EnRs and also on the conservation of tracer concentrations from the flames to the sampling point than on the mathematics of median estimation. Consequently, the following paragraphs explore these questions related to the number and choice of tracers.

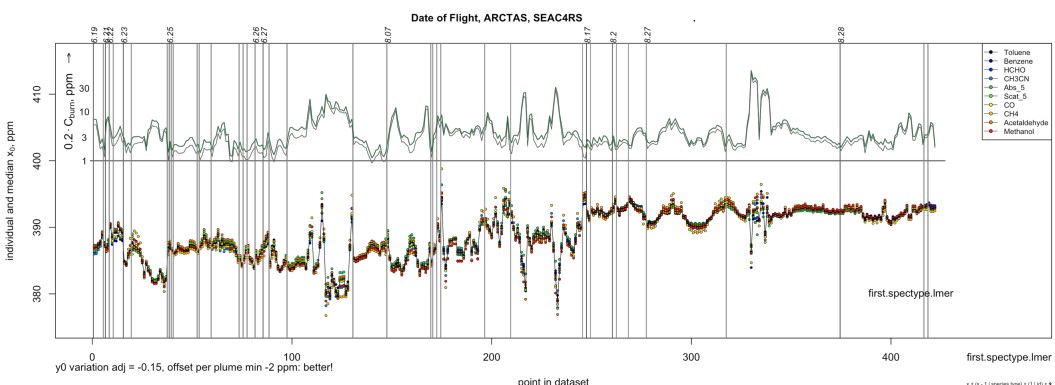

**Figure 8. (Lower panel). Estimates of the 422 background $\hat{x}_i^0$ = CO$_2$+CO concentrations implied based on the 10 fire tracers indicated in the legend. Individual $\hat{x}_{ij}^0$ are shown by overlapping colored bars (–), with the median estimate indicated by a black bar. (Upper panel) Estimates of $C_{\text{burn}} = x_i - \hat{x}_i^0$ indicators of fuel carbon burned, in green line. A preliminary estimate of $C_{\text{burn}}$ based on the consensus of tracer deviations (without variable EnR estimates) is also shown. Flight days are indicated by the days marked on the top axes, and individual plumes, separated by non-plume concentrations of longer than 10 minutes, are shown as vertical separator lines.**

*How does the number of tracers affect results?* What are the effects of using alternate or simpler sets of tracers? How many tracers are required for stable estimates? We began to address these questions by examining estimates made with fewer tracers in the intercept-determining set: the selection of the set of $j$'s. The supplementary material gives two examples of subsets. Here is a summary of that material. The two sets chosen are those that are the most unambiguous indica-

tors of $\hat{x}_i^0$ based on their mutual agreement with $\hat{x}_i^0$ from the full set of 10 tracers. They are Set 1 (CO, Scat_5, and HCHO) and Set 2 (CO, Scat_5, HCHO, acetaldehyde, and toluene). These are indicated by a careful examination of Figure S8 in the Supplementary Material. (A note: methane and Abs_5 contributed $\hat{x}_{ij}^0$ most varying from $\hat{x}_{ij}^0$, and each with separate patterns of disagreement among the samples.) Set 2 gave variations around 0.02 ppm, the smaller set, Set 1, gave

very similar variation except for the flights of June 22 and 25, where many observation instances varied by around 0.1 ppm, but with 11 points out of 422 differing by 0.3 ppm. This level of agreement surprised us. More significantly to our aims, the relative error in $C_{\text{burn}}$ was only about 2%. When sets containing the less correlated tracers were used, deviations ranged up to 0.2–0.4 ppm, which appeared still remarkably small.

Observation-to-observation consistency in $\hat{x}_i^0$-estimates, seen for most plumes observed in Fig. 9, is the strongest argument for the precision of the $C_{\text{burn}}$ estimates. Note that the successive estimates are essentially independent of each other. There is of course the dependency due to each observation's contribution to the estimate as one component of the entire dataset. This continuity is maintained even though the magnitudes of CO$_2$+CO and estimated $C_{\text{burn}}$ can change

dramatically as the sampling aircraft enters and leaves each plume. Smooth excursions seen early in the flight marked 8.27 are explicable in terms of large changes in sampling altitude and location around the Rim Fire on that day. There are variations in $\hat{x}_i^0$ from plume to plume and day to day.

In contrast to this typical continuity of $\hat{x}_i^0$-estimates, there are 15 to 20 brief and large excur-

sions which deserve some attention. Of course, these may be disregarded in getting a general picture of EnRs. All the tracers suggest these excursions of the median, although there is a larger



variation between the individual tracer-based estimates $\hat{x}^0_{ij}$. These excursions are always associ-
ated with large changes in $CO_2+CO$ and $C_{burn}$, but often they occur one minute later. We exam-
ined these excursions in detail. They do not seem to relate to changes in the EnRs $\hat{a}_j$ (as qualified
by fire-type) estimated simultaneously. The observations $y_{ij}$ and the fitted $\hat{y}_{ij}$ agree well, as well
as for non-excursion points. Note however, that we may only use a consistent set of $\hat{a}_j$'s and $\hat{y}_{ij}$
to make the $\hat{x}^0_{ij}$ estimates, including the effect of fire-type classes on $\hat{a}_j$'s, and that this enforces
considerable homogeneity in the $\hat{x}^0_{ij}$ estimates.

The results for $C_{burn}$ and tracer EnRs suggested to us that one likely source of uncertainty is
that $C_{burn}$, $\hat{x}^0_i$, and the tracers may change very rapidly in comparison to our one-minute sampling
intervals. Looking into this, we found that many of the $C_{burn}$ estimates are of small magnitude, 12
of the 422 samples yielded $C_{burn} < 1.5$ ppm. Even large jumps from sample to sample in esti-
mated $\hat{x}^0_i$ were not particularly associated with anomalous estimates of $C_{burn}$. The remaining, ap-
pealing possibility is occasional imprecise time alignment of all measurements, particularly of
the $CO_2$ measurements. Such imprecise alignment could happen at any stage, from sampling line
delays to interpolation to one-minute time intervals. Such variations in $CO_2$ would affect the $\hat{x}^0_{ij}$
found for all tracers in a coordinated way, just as was observed. Note that estimates of $C_{burn}$ were
little affected, since significant $\hat{x}^0_{ij}$ excursions were associated with large $CO_2+CO$ values.

More technical observations are these: one implication of this tendency for homogeneity in
the $\hat{x}^0_{ij}$ estimates is that the MERET technique works best with a large set of alternative fire-type
classes. MERET should work better with more tracers. Adding additional classes (clusters) also
tends to add only minor variations in the slopes $\hat{a}_j$, and harmful effects seen in over-fitting of re-
gression models are avoided by a requirement that the $\hat{a}_j$'s be positive.

## 8. Estimates of emissions ratios: Two MERET examples

### 8.1. MERET results for our two examples

The usefulness of our estimates of $\hat{x}^0_i$ and $C_{burn} = x_i - \hat{x}^0_i$ is seen in the MERET analysis (Fig. 9)
of the two case studies analyzed above using the NEMR approach, where portions of Flights 10
and 14 were shown in Fig. 4. The tracers CO and $b_{scat}$ appear much better correlated with the
$C_{burn}$ from MERET, especially in Flight 14. The plots for both CO and scattering imply linear re-
lationships with an implied intercept near 0, i.e., background values of $y_i$ have been satisfactorily
removed. For CO, the $x$ and $y$ intercepts for each segment joining observations are shown in a
light red color near the origin. Difficulties with a variable background not evident. The slopes of
all the lines do not all agree, however. The observation points are marked by colors following the
fire-type class we obtained using all tracers in the rough categorization of correlated variations in
slopes. (Refer back to Section 6.2 and Section 6.3, step 5 – "Roughly cluster".) The colors are
correlated from one observation to the next, suggesting that the somewhat differing plumes char-
acterized are resolved with our sampling frequency. Variation of slope is more evident for $b_{scat}$
than CO. Generally, the slopes for CO tend to be more uniform in all flights analyzed.

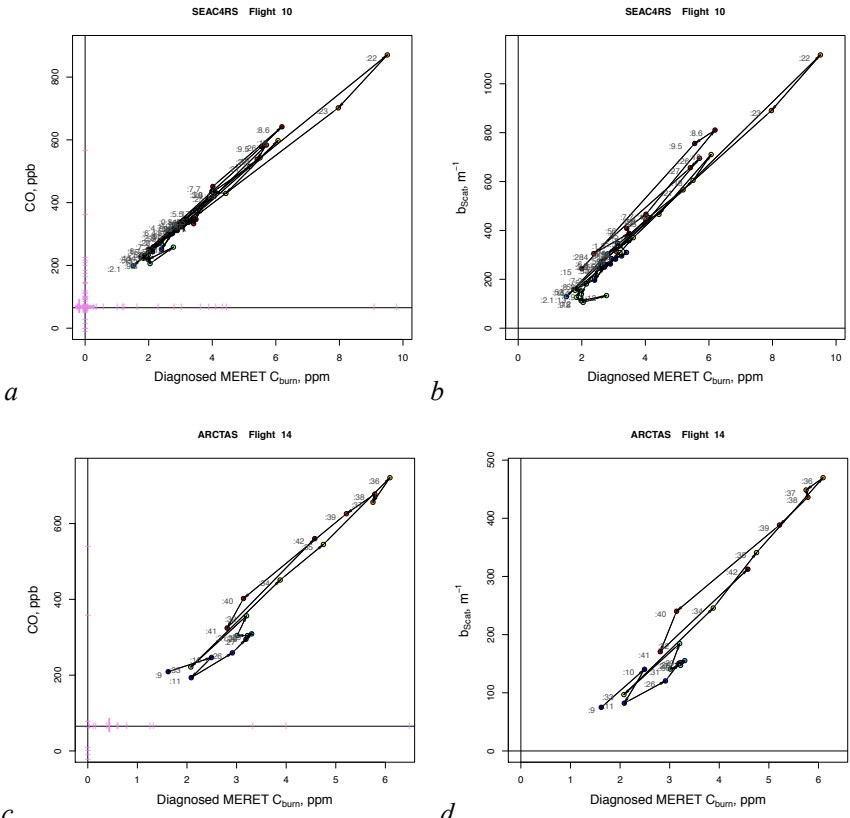

**Figure 9. Analyzed relation of tracers to carbon burned using the MERET technique for portions of SEAC4RS Flight 10 and ARCTAS Flight 14. Compare (a) through (d) with Figure 4, (b), (c), (e), and (f). Lightly colored marks on the axes show intercepts associated with the analysis for each instance.**

### 8.1 Table of several significant emissions

Table 3 provides a summary of the EnR relationships for some of the most significant gaseous emissions and particulate properties. In many cases, these EnRs can be converted to ERs and emission factors when the relationship of airborne $C_{burn}$ to surface fuel consumed can be established. For the most highly reactive species, these EnRs will tend to be underestimates. An interpretive paper (Chatfield and Andreae, 2019) will give additional information on the photochemi-

cal age of the observation in many cases. Ozone and peroxy acetyl nitrate (PAN) are not emissions, but produced in the plumes. The relationships to fuel burned, and their variations are nevertheless interesting. Descriptions of variation are given as the 16[th] and 84[th] percentiles of all the estimates. These are similar to error estimates if nothing more is known about the origin and age of the particular samples, a matter more fully discussed in Chatfield and Andreae (2019).






**Table 3. EnR Estimates for Fire Emissions Considered**

| Fire Emission | EnR estimate | Percentile 16 | Percentile 84 | Unit | Conversion factor to EF |
|---|---|---|---|---|---|
| CO | 74 | 62 | 85 | ppb ppm$^{-1}$ | 1.17 |
| CH$_4$ | 8.6 | 2.3 | 13.0 | ppb ppm$^{-1}$ | 0.67 |
| Ethyne | 0.26 | 0.205 | 0.31 | ppb ppm$^{-1}$ | 1.08 |
| Ethene | 0.88 | 0.65 | 1.07 | ppb ppm$^{-1}$ | 1.17 |
| Ethane | 0.70 | 0.57 | 0.80 | ppb ppm$^{-1}$ | 1.25 |
| Propene | 0.056 | 0.005 | 0.100 | ppb ppm$^{-1}$ | 1.75 |
| Propane | 0.16 | 0.12 | 0.19 | ppb ppm$^{-1}$ | 1.83 |
| n-Butane | 0.028 | 0.019 | 0.037 | ppb ppm$^{-1}$ | 2.17 |
| Benzene | 0.094 | 0.073 | 0.134 | ppb ppm$^{-1}$ | 3.25 |
| Toluene | 0.054 | 0.023 | 0.067 | ppb ppm$^{-1}$ | 3.88 |
| Methanol | 2.1 | 1.7 | 3.1 | ppb ppm$^{-1}$ | 1.33 |
| HCHO | 1.15 | 0.81 | 1.62 | ppb ppm$^{-1}$ | *1.25* |
| Acetaldehyde | 0.56 | 0.24 | 0.71 | ppb ppm$^{-1}$ | 1.83 |
| Acetone | 0.74 | 0.54 | 1.14 | ppb ppm$^{-1}$ | 2.42 |
| CH$_3$CN | 0.13 | 0.11 | 0.16 | ppb ppm$^{-1}$ | 1.25 |
| NO$_x$ (as N) | 0.051 | 0.024 | 0.131 | ppb ppm$^{-1}$ | 0.63 |
| O$_3$ | 14.8 | 8.5 | 25.1 | ppb ppm$^{-1}$ | *(2.0)* |
| PAN | 0.26 | 0.17 | 0.38 | ppb ppm$^{-1}$ | *(3.17)* |
| Scat_5, $b_{Scat}$ | 79 | 50 | 100 | m$^{-1}$ ppm$^{-1}$ | 0.042 |
| Abs_5, $b_{Abs}$ | 3.2 | 2.2 | 4.4 | m$^{-1}$ ppm$^{-1}$ | 0.042 |
| Ammonium | 0.32 | 0.19 | 0.47 | µg m$^{-3}$ ppm$^{-1}$ | 0.032 |
| Nitrate | 0.28 | 0.11 | 0.60 | µg m$^{-3}$ ppm$^{-1}$ | 0.107 |
| Sulfate | 0.156 | 0.063 | 0.290 | µg m$^{-3}$ ppm$^{-1}$ | 0.164 |

Notes: Conversions assume a C to dry biomass ratio of 0.5. Conversions to µg m$^{-3}$ assume 25 ºC and 1013 hPa. O$_3$ and PAN are not directly produced by fires. HCHO is produced but often decreases rapidly. Under appropriate conditions indicated in Chatfield and Andreae (2019), the EnR estimates can be used as ERs. For tracers that are rapidly removed or transformed, these tend to be the higher values.

## 9. Conclusions

A major problem with the estimation of fire enhancement ratios and emission factors is inherent in their character: flames promote mixing in their plumes. Total carbon liberated to the atmosphere (approximately $C_{burn} = CO_2+CO$) is mixed with background air at different points in the plume's evolution, and removal of that mixing effect has been a difficulty. The NEMR technique often uses CO as unique tracer, but the EnR of CO is variable, adding uncertainty to the estimation of the EFs. Given the variability of CO due to combustion efficiency (MCE) and environmental variability, it has been emphasized that the NEMR technique can only be confidently applied in situations in which conditions affecting the ratio of CO to (CO$_2$+CO) can be well determined, ideally from source to sampling (Yokelson et al., 2013). The method also tends to emphasize the use of samples of CO and tracer collected over many minutes, so that the regression method for EnRs of tracer relative to CO, defining , $a_{CO \leftarrow (fire-added\ c_{burn})}$ becomes stable, and a conversion to fuel carbon burned becomes possible.



840  We sought to decompose $C_{tot}$ into $C_{bkgd}$ and $C_{burn}$. However, meteorology and mixing allow significant variations in $C_{bkgd}$ due to other powerful processes, e.g., $CO_2$ from respiration/photosynthesis in mixed-layer air. Once lofted, $C_{bkgd}$ varies little unless the plume enters layers of free tropospheric air from long-range transport with different $C_{bkgd}$, which further dilute the plume (Yokelson et al., 2013). We noted such problems using NEMR in analyzing a significant number of plumes for enhancement ratios studied in the Western US during two campaigns, ARCTAS-

845  California and SEAC4RS, with 422 one-minute samples in all.

 The problem of deriving an accurate $C_{bkgd}$ is solved by noticing that there are two different kinds of information provided by multiple observational instances of a tracer and multiple tracers at a single instance. Information about the various EnRs and $C_{bkgd}$ are mixed, but not inextricably. There is a solution based on mixed-effects (also called random effects) regression modeling.

850  We propose a Mixed Effects Regression Emissions Technique (MERET) to replace or at least to check on NEMR, for which we used the **R** routine *lmer()*.

 The MERET technique is related to traditional entraining-plume models for parcels. We presented a synthesis describing multiple tracers from fire to sampling location. Sample calculations with the model suggest that it deals appropriately (linearly) with several varied histories for

855  plume mixing. This motivates a regression equation for an "equivalent background" $x_i^0$ for each observation that is related to entraining concentrations $x^E(t)$ along the trajectory, and is appropriate for each tracer species (Figure 8). The theory then allows this $x_i^0$ to be used to define $C_{burn}$ and thus to define the EnR for any appropriate fire-derived variable. This technique makes EnRs appear appropriate even in difficult situations, and allows estimates of EnRs for individual sam-

860  ples.

 EnRs are appropriate for the estimation of emission factors when the plume age is short compared to the transformation timescale of the measured fire tracer, and we provide an approximate diagnostic for this age for most samples. Formaldehyde, acetaldehyde, the alkenes, benzene, $NO_x$, $b_{Scat}$, and $b_{Abs}$ particularly require such attention.

865  Analysis of the entire dataset from the Rim Fire gave relatively good comparisons of NEMR and MERET. However, NEMR had to be applied for a very long time period (> 30 min); shorter periods gave puzzling results. Variability of the MCE and the $C_{bkgd}$ each seemed to have a role. Our MERET analysis suggested various fire conditions and ages for the airborne sample during this period. The time series of EnRs suggested appropriate sub-periods where the NEMR could

870  be more skillfully used.

 Carbon monoxide is usually best single tracer that correlates with fire emissions ($C_{burn}$), supporting the use of the NEMR technique. Our analysis suggests other tracers had EnR variations that collectively helped to distinguish $C_{burn}$ from CO in regression. The NMER methodology depends on a full analysis of the history of CO influences on a sample to obtain a reliable MCE.

875  MERET allows estimates of MCE as well as $C_{burn}$ for each sample. Thus it demarcates sampling periods with nearly homogeneous MCE. However, possible large variations in the entraining background of CO should still be considered carefully in dilute plumes with $C_{burn}$ <2 ppm.

### 9.1. Questions for future research

 We conclude with some questions for future research; these also review the suggested conclu-

880  sions of this paper and acknowledge the limitations of a single publication.



(1) How well can the use of one or a few tracers, e.g., CO, $b_{scat}$, HCHO, actually constrain EnRs and EFs when only a few instruments may be used? How many variables need to be measured or how fresh should the plumes be to allow CO to be used both as a fire tracer and to allow useful estimates of MCE?

(2) Can MERET be used to identify time periods of relatively homogeneous MCE, and can that MCE value be used with NEMR to create suitable EnRs? Since NEMR uses differences sample by sample (in time), no minimum value of another tracer need be estimated. (Consider that MERET does allow some evaluation of the minimum value estimate to be assessed and a better minimum assigned.)

(3) Can the MERET/NEMR and better near-fire non-plume sampling help us to prevent mis-attribution of fire emissions? These would include observations for fire intake air, air likely to be entrained in ascent, and air surrounding a plume and likely to be entrained as a plume spreads downwind. Can simulations of entraining plumes aid this effort?

(4) What do "fire types" represent and which species or properties tend to correlate in their
EnRs (Chatfield and Andreae, 2019).

## 10. Team List

**ARCTAS Science Team:**
James Crawford (NASA Langley Research Center), Henry Fuelberg (Florida State University), Chris Hostetler (NASA Langley Research Center), Daniel Jacob (Harvard University), Hal Mar-
ing (NASA Headquarters), Philip B. Russell (NASA Ames Research Center), Kent Shiffer (NASA Ames Research Center), Hanwant Singh (NASA Ames Research Center), Kathy Thompson (Science Systems and Applications, Inc.), Bruce E. Anderson (NASA Langley Research Center), Eric Apel (National Center for Atmospheric Research), Donald R. Blake (University of California Irvine), William H. Brune (Pennsyvania State University), Christopher A. Cantrell
(National Center for Atmospheric Research), Ronald C. Cohen (University of California Berkeley), Jack Dibb (University of New Hampshire), Glenn S. Diskin (NASA Langley Research Center), Alan Fried (University of Colorado), Johnathan W. Hair (NASA Langley Research Center), L. Greg Huey (Georgia Tech), Jose-Luis Jimenez (CIRES, University of Colorado), Yutaka Kondo (University of Tokyo), Richard E. Shetter (University of North Dakota), Robert Talbot
(University of New Hampshire), Stephanie A. Vay (NASA Langley Research Center), Rodney Weber (Georgia Tech), Andrew Weinheimer (National Center for Atmospheric Research), Paul Wennberg (Cal Tech), Armin Wisthaler (Universitãt Innsbruck and Universitatet i Oslo), John Barrick (NASA Langley Research Center), Anthony Bucholtz (Naval Research Laboratory), Antony Clarke (University of Hawaii), Lawrence C. Freudinger (NASA Armstrong Flight Center),
Charles Gatebe (NASA Goddard Space Flight Center), Athanasios Nenes (Georgia Tech), James Podolske (NASA Ames Research Center), Jens Redemann (University of Oklahoma), Sebastian Schmidt (CU Boulder), Anthony Strawa (NASA Ames Research Center), Brian Cairns (NASA Goddard Institute for Space Science), Richard Ferrare (NASA Langley Research Center), Samuel Oltmans (NOAA ESRL, CU Boulder), Glenn Shaw (University of Alaska Fairbanks), Anne
M. Thompson (NASA Goddard Space Flight Center), Bob Curry (NASA Armstrong Flight Center), David Easmunt (NASA Goddard Space Flight Center), Michael S. Wusk (NASA Langley Research Center).





**SEAC4RS Science Team:**

Hal Maring (NASA Headquarters), Ken Jucks (NASA Headquarters), Jassim Al-Saadi (NASA Headquarters), Richard Eckman (NASA Headquarters), Alex Pszenny (NASA Headquarters), Kent Shiffer (NASA Ames Research Center), Jhony Zavaleta (NASA Ames Research Center), Michael Craig (NASA Ames Research Center), David Jordan (NASA Ames Research Center), Quincy Allison (NASA Ames Research Center), Dan Chirica (NASA Ames Research Center),

Susan Tolley (NASA Ames Research Center), Erin Czech (NASA Ames Research Center), Erin Justice (NASA Ames Research Center), Katja Drdla (NASA Ames Research Center), Brian Toon (University of Colorado), Eric Jensen (NASA Ames Research Center), Rich Ferrare (NASA Langley Research Center), David Diner (NASA Jet Propulsion Laboratory), Anthony Bucholtz (Naval Research Laboratory), Lance Christensen (NASA Jet Propulsion Laboratory),

Matt McGill (NASA Goddard Space Flight Center), Steve Platnick (NASA Goddard Space Flight Center), Brian Cairns (NASA Goddard Space Flight Center), Jim Anderson (Harvard University), Rushan Gao (NOAA ESRL), Bob Herman (NASA Jet Propulsion Laboratory), MJ Mahoney (NASA Jet Propulsion Laboratory), Rushan Gao (NOAA ESRL), Steve Wofsy (Harvard University), Sebastian Schmidt (University of Colorado), Elliot Atlas (University of Miami),

Paul Bui (NASA Ames Research Center), Hanwant Singh (NASA Ames Research Center), Jack Dibb (University of New Hampshire), Phillip Russell (NASA Ames Research Center), Jose-Luis Jimenez (University of Colorado), Charles Brock (NOAA ESRL), Simone Tanelli (NASA Jet Propulsion Laboratory), Andreas Beyersdorf (NASA Langley Research Center), Anthony Bucholtz (Naval Research Laboratory), Samual Hall (National Center for Atmospheric Research),

Paul Wennberg (Cai Tech), Thomas Ryerson (NOAA ESRL), Glenn Diskin (NASA Langley Research Center), Armin Sorooshian (University of Arizona), Alan Fried (National Center for Atmospheric Research), John Hair (NASA Langley Research Center), Glenn Diskin (NASA Langley Research Center), Greg Huey (Georgia Tech), Rushan Gao (NOAA ESRL), Thomas Hanisco (NASA Goddard Space Flight Center), Bruce Anderson (NASA Langley Research Center), Paul

Bui (NASA Ames Research Center), Karl Froyd (NOAA ESRL), Greg Huey (Georgia Tech), Armin Wisthaler (Universität Innsbruck), Jack Dibb (University of New Hampshire), Paul Lawson (SPEC Incorporated), Sebastian Schmidt (University of Colorado), Ronald Cohen (University of California Berkeley), Donald Blake (University of California Irvine), Joshua Schwarz (NOAA ESRL), J. Vanderlei Martins (University of Maryland Baltimore County), Lenny Pfister

(NASA Ames Research Center), Henry Fuelberg (NASA Langley Research Center), Rennie Selkirk (Universities Space Research  Association), Daniel Jacob (Harvard University), Arlindo Da Silva (NASA Goddard Space Flight Center), Paul Konopka (Georgia Tech), Ed Hyer (Naval Research Laboratory), Greg Charmichael (University of Iowa), Jennifer Olson (NASA Langley Research Center), Louisa Emmons (National Center for Atmospheric Research), Gao Chen (NASA

Langley Research Center), Chip Trepte (NASA Langley Research Center), Jay Mace (University of Utah), Pat Minnis (NASA Langley Research Center), Eric Ray (NOAA ESRL), Karen Rosenlof (NOAA ESRL), Daniel Jacob (Harvard University), Robert Yokelson (University of Montana), Jeffery Reid (Naval Research Laboratory), Jens Redemann (NASA Ames Research Center), David Starr (NASA Goddard Space Flight Center), Johnny Luo (City College of New

York), Jay Mace (University of Utah), Laura Pan (National Center for Atmospheric Research), David Starr (NASA Goddard Space Flight Center), Jay Mace (University of  Utah), Johnny Luo (City College of New York), James Crawford (NASA Langley Research Center), Frank Culter (NASA Armstrong Flight Center), Patrick Lloyd (NASA Armstrong Flight Center), Rick Shetter



(National Suborbital Education and Research Center), Adam Webster (National Suborbital Education and Research Center), Eric Buzay (National Suborbital Education and Research Center), David VanGilst (National Suborbital Education and Research Center), Jane Peterson (National Suborbital Education and Research Center), Tim Moes (NASA Armstrong Flight Center), Chris Miller (NASA Armstrong Flight Center), Mike Kapitzke (NASA Armstrong Flight Center), Denis Steele (NASA Armstrong Flight Center), Stuart Broce (NASA Armstrong Flight Center),
Jan Nystrom (NASA Armstrong Flight Center), Kevin Kraft (NASA Armstrong Flight Center).

Some of the team have changed affiliations since their work was documented. Two members of NASA's Earth Science Projects Office (ESPO) may direct you on how to make enquiries regarding either team, marilyn.vasques@nasa.gov and bernadette.luna@nasa.gov. Also, a leader at
NASA Headquarters, has some knowledge of both teams' members, barry.lefer@nasa.gov.

## 11. Acknowledgements

Particular acknowledgment goes to the instrumental groups in Table 1 and Section 4 and the many contributing efforts of the ARCTAS and SEAC4RS Science Teams and staff. Chatfield pursued this research under NASA grant 13-ACCDAM13-0110, and the effort grew out of re-
search funded by California Air Resources Board and San Joaquin Valley air regional authority, where difficulties with previous methodology became apparent. We appreciate reviews by S. Palacios, M. Segal-Rozenhaimer, E. Yates, and especially R. Koppmann. M. O. Andreae acknowledges extensive support by the German Max Planck Society and facilitation of recent research by the Scripps Institute of Oceanography. Robert F. Esswein helped with certain data reformatting.
We acknowledge the great value of the **R** programming language and the lme4 and NMF packages.

## 12. Author contributions

MOA contributed motivating ideas and approach following on Yokelson et al. (2013). RBC brought data into suitable format and evaluated suitability of tracers. He contributed the theory of
parallel plume rise expressions for fire tracers and the mixed-effects modeling approach. RBC wrote the publication with considerable contributions of MOA.





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
