# Peer review of "Emissions Relationships in Western Forest Fire Plumes: I. Reducing the Effect of Mixing Errors on Emission Factors"

_Atmospheric Measurement Techniques, 2019_

## Referee Comment (RC1) · Anonymous Referee #1 · 4 Oct 2019

This manuscript is one of a set of publications, of which the second appears to be in preparation (Chatfield and Andreae, 2019). The authors take over 400 samples from the recent ARCTAS and SEAC4RS aircraft campaigns and develop a complex methodology for estimating Enhancement Ratios (EnRs) and their relationship to emission ratios (ERs) and emission factors (EFs), as well as a means for separating the background and burned carbon content in measurements of carbon emissions from biomass burning plumes. The primary work here is in the development of the Mixed Effects Regression Emission Technique (MERET), which can be used to disentangle total carbon levels in a plume from the carbon existing in background air and carbon from the biomass burning itself. The manuscript starts with a review of previous work quantifying EnRs, ERs, and EFs including a thorough description of the history of EF estimation efforts. It then moves to a description of the methodology in which data are selected and a description of their properties. Then follows two separate theory sections (one on the theory of plume expansion and one on EnR regression relationships), and then another methodology section on determining the background carbon levels and the description of the MERET method. Next, results are presented, and two examples are given. The manuscript closes with conclusions and questions for future research.

This is a very long manuscript that alternates between analysis and theory. Both parts have significant scientific value, but the manuscript itself is difficult to follow due to its length, the complexity of both the theory and analysis sections, and the way in which the manuscript is presented. There is no clear outline presented in the manuscript, so I had to create my own (copied below) in order to grasp the manuscript completely. The manuscript includes two separate methodology sections (Section 2 and 6) and two separate theory sections (Section 4 and 5), and their arrangement and transitions left me frequently confused. Additionally, there are many instances of parenthetical asides, notes, and comments (e.g., L362-365, L399-409, L422-428, all of Section 6.4) that interrupt the flow of the manuscript and greatly impede its overall understandability. The conversational tone of this manuscript additionally introduces confusion. For instance, L203 stats "We now move to..." and it's unclear if this means in the following paragraphs or in the next section. In L312 the phrase "Recall that..." is unclear. Also, the included figures are very difficult to understand, in part because their text, captions, and legends are frequently too small to read (esp. Figures 4, 8, and 9) and because full explanations of what are in the figures are found both within the figure captions themselves and within various portions of the manuscript body. Overall, these makes the manuscript difficult to follow and the presented scientific concepts and results difficult to understand.

1. Intro
    1.1. Importance of Previous Work
    1.2. Development of EF Estimation to Date
2. Methodology: defining an indicator dataset
3. Observed behavior of $C_{tot}$ in fire plumes – Properties of tracers
4. Theory: expanding plume for several species
    4.1. A general relationship
    4.2. Examples showing robustness of computations of idealized $C_{tot}$
5. Theory: A regression relationship for EnRs

I feel that there are two different manuscripts here, or at least one manuscript with a large appendix or supplement that includes the majority of the theory (Sections 4, 5, pages 12 – 19). The forthcoming paper (Chatfield and Andreae (2019) appears to be a useful companion to this manuscript, and it is referenced several times (e.g., L669-672), but it is unclear if the two papers are meant to be considered together or if they are stand-alone manuscripts.

While I believe that this manuscript has significant scientific value and falls within the scope of AMT, and that the work described and methodology proposed (the MERET method) has substantial value, the current structure and length imposes a significant impediment on its understandability and impact. There were many times in which I was confused or lost, and so while I feel like I understand much of what was presented, I am not confident that the manuscript has successfully communicated all that the authors intended. As such, I feel that significant reorganization and clarification is needed before this can be recommended for publication.

Independent of these issues, the following is a review of the content of the manuscript itself.

The scientific value of understanding forest-fire plume properties, and in particular of quantifying the enhancement ratios (EnRs) for properties of interest via the MERET method, is very high and this manuscript is a significant contribution to the field. The descriptions of the relationships between EnRs, ERs, and EFs in Section 1 is informative, although it would be particularly valuable if additional descriptions of how EnRs "approximate emission ratios (ERs)" (L77) if they are sampled before atmospheric transformations can occur. What is the relation after transformations? This needs to be made clear in the introduction.

The interpretation of Figures 4b,c,e,f in Section 3 is extremely valuable, but I largely struggled with understanding what was being represented until the description of the different examples later in the manuscript (esp. Sections 4.2 and 5). Only on a second read-through was I able to follow the text and more completely understand what is presented in Figure 4.

SPECIFIC COMMENTS

There are many places where there are typos and undefined variables being used in the manuscript:

- L54: "Chatfield and Andreae (2017)" should be "Chatfield and Andreae (2019, in preparation)"
- L66: "$\Delta CO_{tot}$" should be "$\Delta C_{tot}$".
- Table 1: The line labeled "Proportional to carbon burned: define" is confusing. What does define mean here? Is this a typo?
- Figure 2 refers to a slope of 32.60458 while the text (L299) refers to a slope of $33 \times 10^{-3}$. This inconsistency is confusing.
- The variable $C_j$ used in L417-418 and other lines does not appear in the Table of Symbols (Table 2) and is only described on L418
- L425: "...the same plume. *provided* we..." is confusing
- Figure 6 has an x-axis label of $C_{tot}$ while the text (L469) refers to $C_{burn}$
- L659: "However, we let the define the types..." seems to be missing a word.
- I believe "Figure 9" on Line 733 should be "Figure 8"

The phrase "affine dependence" is used several times (e.g., L145) and is unfamiliar to me.

In Section 1.2, there are many places where I get lost. For instance, the equation on L168 lacks a sufficient description and I'm unsure what the "$a_j$ <-- CO" and "$a_{CO}$ <-- (fire – added CO2 + CO)" terms mean. I feel a more complete explanation is needed.

The use of the variable x for $C_{tot}$ in Section 1.2 and other places is confusing, especially when $C_{tot}$ and x are used together (e.g., L153-158).

L522-523: The suggestion that the reader should make their own calculations in order to understand the linear responses is unhelpful.

L528-529: I do not understand what is meant by "provides safety against a variable and incompletely described background" or "The median is not affected by undetected changes in background..."

Figure 4 is extremely difficult to understand as there is almost no description in the caption itself; the descriptions and explanations are found within the text body. Specifically:

- The text and images are very small
  - The label "$b_{scat}$" in Figure 4a,c is too small
  - The number labels in Figure 4b,c,e,f are too small
- There are many individual components that are confusing

- It is nearly impossible to see the arrows in 4b,c,e,f
- It is unclear without locating the matching description in the text what:
  - the color lines are on the top of Figure 4a
  - What the difference is between the blow circles, red circles, and orange dots are in Figures 4a,d
  - What the blue dashes are in Figures 4a,d
  - What the colors indicate in Figures 4b,c,e,f
  - Which of the three plotted variable are on which line in Figures 4a,d

The text and color labels in Figures 8 and 9 are similarly difficult to see and understand.

---

## Referee Comment (RC2) · Anonymous Referee #2 · 4 Feb 2020

This is an interesting manuscript and the authors deserve credit for putting a great deal of effort into it. It presents some of the challenges in interpreting atmospheric observations of biomass burning emissions. The writing is not straightforward, but I believe the paper presents the following: To understand the impact of fires at local, regional and global scales, the emissions of various important atmospherically relevant species need to be known at the source. Emission ratios (ERs) provide a means for deriving emission factors. Often it is not possible to obtain ERs because the mixing ratios measured in aircraft studies are often not sampled near t=0 so enhancement ratios (EnRs) obtained at some time from t=0 can be useful for estimating ERs if the physical and chemical transformational history since emission is known. The MERET method is

put forward and described as a means to extract more information from chemical data downwind of a fire with the ultimate goal of estimation of emission factors.

In my opinion, the manuscript needs a better organizational structure as it is quite difficult to follow. I found the conversational style of writing also distracting, and it requires a great deal of effort from the reader. The manuscript is very long and there is no clear payoff at the end. The authors should work on providing a more concise presentation of their work. For me, even after some effort I found it difficult to make sense of a lot of it. I am sure that there is a good deal of scientific merit embedded within the manuscript and I believe with some reformulation there is a forum somewhere for these concepts to be presented. I recommend rejecting this version of the manuscript.

---

## Author Comment (AC1) · 14 Apr 2020

There is no clear outline presented in the manuscript, so I had to create my own (copied below) in order to grasp the manuscript completely. The manuscript includes two separate methodology sections (Section 2 and 6) and two separate theory sections (Section 4 and 5), and their arrangement and transitions left me frequently confused. We have included "An Outline of This Paper" immediately after the first paragraph, following the style suggested by R1. We will reformat this as AMT will ultimately decide. We believe that leads us to omit paragraphs Section 1.1, L120–L140, which are over-detailed superfluous. Additionally, there are many instances of parenthetical asides, notes,

and comments (e.g., L362-365, L399-409, L422- 428, all of Section 6.4) that interrupt the flow of the manuscript and greatly impede its overall understandability. These instances and one other have been replaced by a named section of the Supplementary Material, e.g. "See also SM for a Note on Initial Point." at L362. I have attempted to make all such references minimally disruptive to the flow of the paper. Following AMT guidelines, they are not fundamental to advancing the arguments of the paper. The conversational tone of this manuscript additionally introduces confusion. For instance, L203 stats "We now move to..." and it's unclear if this means in the following paragraphs or in the next section. In L312 the phrase "Recall that..." is unclear. These are restated.: L203: The next section provides motivation for and understanding of an alternate approach ... L312: With this section, we illustrate tracer relationships that define our approach to EnRs and EFs in more detail and also in more difficult circumstances, e.g., where the MCE is difficult to estimate, for example because its range of applicability during continued flight sampling is not clear. Also, the included figures are very difficult to understand, in part because their text, captions, and legends are frequently too small to read (esp. Figures 4, 8, and 9) and because full explanations of what are in the figures are found both within the figure captions themselves and within various portions of the manuscript body. Overall, these makes the manuscript difficult to follow and the presented scientific concepts and results difficult to understand. The figures have been largely redrafted to have larger text. Figures 4 and 9 have been redrafted to show labels more clearly. (An remaining error on some time markings will be corrected.) Explanations of Figure 4 are expanded: Figure 1. (a) Timeline of sampling, for the period shown in Figure 3a, Montana, of CO2+CO (blue, left axis) and the fire tracers CO and bscat (red and green points, right axis). Orange-filled points were identified as clear plume points. Unfilled points were not, but might have some fire influence, especially near plume points. (b) scatter diagram of CO vs CO2+CO with arrows showing the time progression of aircraft sampling of identified plume points. Colors provide a key to times shown in (a). (c) a similar diagram of bscat vs CO2+CO. Similar shapes of figures are noted in the text. (d) Timeline of sampling

for the period shown in Figure 3b, Coastal Transect. (e) scatter diagram of CO vs CO2+CO during the transect, like (b). (f) a similar diagram of bscat vs CO2+CO for the Coastal Transect. The black bars graphed in (a) and (d) are estimates of non-fire influenced Cbkgd, see text. They and the non-plume points suggest air-mass changes in CO2+CO. Figure10 has been made larger, and a large display in the published paper is recommended. When points representing different tracers overlap in the figure, this truly signals something about the excellent precision of the individual measurements, and we do not attempt to distinguish them. The figure caption has been expanded: Figure 10. (Lower panel). Estimates of the 422 background x $\grave{I}\acute{C}\_i\hat{}0$ = CO2+CO concentrations implied based on the 10 fire tracers indicated in the legend. Individual x $\grave{I}\acute{C}\_ij\hat{}0$ are shown by overlapping colored bars (–), with the median estimate indicated by a black bar. (Upper panel) Estimates of C_"burn " = x_i-x $\grave{I}\acute{C}\_i\hat{}0$ indicators of fuel carbon burned, in green line. A preliminary estimate of C_"burn " based on the consensus of tracer deviations (without variable EnR estimates) is also shown. Flight days are indicated by the days marked on the top axes, and individual plumes, separated by non-plume concentrations of longer than 10 minutes, are shown as vertical separator lines. A set of horizontal lines at ∼400 ppm indicates selected intervals for optimizing numerics (see text, Section 6.3, item 7).. I feel that there are two different manuscripts here, or at least one manuscript with a large appendix or supplement that includes the majority of the theory (Sections 4, 5, pages 12 – 19). The forthcoming paper (Chatfield and Andreae (2019) appears to be a useful companion to this manuscript, and it is referenced several times (e.g., L669-672), but it is unclear if the two papers are meant to be considered together or if they are stand-alone manuscripts. While I believe that this manuscript has significant scientific value and falls within the scope of AMT, and that the work described and methodology proposed (the MERET method) has substantial value, the current structure and length imposes a significant impediment on its understandability and impact. There were many times in which I was confused or lost, and so while I feel like I understand much of what was presented, I am not confident that the manuscript has successfully communicated all that the authors intended. As such, I

feel that significant reorganization and clarification is needed before this can be recommended for publication. This is well-considered, but the authors find few other options. We have put much more into the Supplementary Material. Consequently, (a) Material not strictly necessary has been moved to the material. (b) A table of contents has been included, following the reviewer's first comment and suggestion above. (c) The fact that the paper contains a development of plume theory is more prominent in the abstract: A new theoretical development of plume theory for multiple tracers is developed after examining the aircraft samples If the editors of AMT allow, we could change the title to: Theory and Estimation of Emissions Relationships in Forest Firea Plumes: 1: Reducing Effect of Mixing Errors on Emission Factors (d) The authors do not think that the theory could stand alone without showing that it leads to apparently good statistical estimates. and are unwilling to begin the whole AMT review process again if we suggest a division. The scientific value of understanding forest-fire plume properties, and in particular of quantifying the enhancement ratios (EnRs) for properties of interest via the MERET method, is very high and this manuscript is a significant contribution to the field. The descriptions of the relationships between EnRs, ERs, and EFs in Section 1 is informative, although it would be particularly valuable if additional descriptions of how EnRs "approximate emission ratios (ERs)" (L77) if they are sampled before atmospheric transformations can occur. What is the relation after transformations? This needs to be made clear in the introduction. Besides rewriting the paragraph, we have added a note to the Supplementary Material which clarifies this:' More on the relationships of EnRs, ERs, and EFs is found in the Supplementary Material (SM), "Note on EnRs and ERs". ' The reviewer appears to want more information about when ERs can be larger or smaller than EnRs. This seemed appropriate for a note. A helpful suggestion! The interpretation of Figures 4b,c,e,f in Section 3 is extremely valuable, but I largely struggled with understanding what was being represented until the description of the different examples later in the manuscript (esp. Sections 4.2 and 5). Only on a second read-through was I able to follow the text and more completely understand what is presented in Figure 4. We thank the reviewer for this observation. We have

rewritten the introductory paragraph: This section provides some examples of Ctot and fire tracers. It illustrates the limitations of changes in Ctot along a sampling path as an indicator of fire influence, Cburn, for emissions estimation and the much greater similarities of the such changes of tracers that possess shorter transformation time-scales. These define our approach to EnRs and EFs. The relation of fire emissions to observed Ctot to Cburn, can be apparently simple or complex, depending on how the history of non-fire CO and CO2 entrained into fire plume air parcels affects Ctot. We show this commonality of relationships will to motivate the theory of expanding plumes in Section 4. That theory will suggest a method worked out in Sections 5 and 6 to find the key variable, Cbkgd, that then provides Cburn and thus EnRs. We have also edited several places succeeding paragraph, not described here.. L54: "Chatfield and Andreae (2017)" should be "Chatfield and Andreae (2019, in preparation)" L66: "DCOtot" should be "DCtot". Table 1: The line labeled "Proportional to carbon burned: define" is confusing. What does define mean here? Is this a typo? Figure 2 refers to a slope of 32.60458 while the text (L299) refers to a slope of 33x10-3. This inconsistency is confusing. The variable Cj used in L417-418 and other lines does not appear in the Table of Symbols (Table 2) and is only described on L418 L425: "...the same plume. provided we..." is confusing

Figure 6 has an x-axis label of Ctot while the text (L469) refers to Cburn L659: "However, we let the define the types..." seems to be missing a word. I believe "Figure 9" on Line 733 should be "Figure 8" √ ïĄĎCtot = ïĄĎCO2 + ïĄĎCO Yes, a typo. Now Proportional to total burned material, as measured by Cburn Chose ppb/ppm rather than ppm/ppm Included.

Changed. Remarks placed in the Supplement. After the equation (12) we now have: For periods of expansion in which the entrained concentrations are constant. See also SM: Note on Varying Entrainment Changed. However, we let the statistical technique define these types, Changed The variable Cj used in L417-418 and other lines does not appear in the Table of Symbols (Table 2) and is only described on L418 L425: "...the

same plume. provided we..." is confusing

Figure 6 has an x-axis label of Ctot while the text (L469) refers to Cburn

L659: "However, we let the define the types..." seems to be missing a word.

I believe "Figure 9" on Line 733 should be "Figure 8" This has been added. The variable is the constant of integration and is generally replaced by a_j=expâĄą(C_j )

Now in Supplementary Material, This now uses alpha and beta for different possible positions, values of i, and re-worded "$\alpha$ and $\beta$, in the same plume. These are supposed chosen so that we know that x^E and all the y_j^Eremain coanstant. Both Ctot and Cburn are used. The x axis has Ctot , units, while the increment beyond the vertical axis at 380, shows Cburn. This is now indicated. "However, we let the statistical technique define these types, and so apply basic clustering techniques." We also added a sentence soon afterward: "NMF and k-means clustering are shown to be equivalent in cases appropriate to our work (Ding et al., 2005)." Yes, Figure 8, thank you! The phrase "affine dependence" is used several times L522-523: The suggestion that the reader should make their own calculations in order to understand the linear responses is unhelpful. True. "Linear" has several meaning in English. See Wikipedia. So we have now "An affine dependence (linear polynomial relationship including an intercept). Linear Transformations in linear algebra must omit the intercept, hence the unusual phrasing "linear polynomial." Chatfield has considered the ramifications of this dependence NMF linear transformations considerably. The reader is relieved of calculations now: "Some similar calculations make it clear that the estimates respond in an appropriate averaging manner under varied assumptions." We simply emphasize the linearity of the analysis. (e.g., L145) and is unfamiliar to me. In Section 1.2, there are many places where I get lost. For instance, the equation on L168 lacks a sufficient description and I'm unsure what the "aj <– CO" and "aCO <– (fire – added CO2 + CO)" terms mean. I feel a more complete explanation is needed. The use of the variable x for Ctot in Section 1.2 and other places is confusing, especially when Ctot and x are

used together (e.g., L153-158). We have added an explanatory phrase: "the slope $a_{(j \leftarrow CO)}$ of the regression estimates of an EnR of the species with respect to CO, multiplied by an attempted very careful estimate of the slope $a_{(CO \leftarrow (fire-added}$ $C\_burn))$ EnR of CO with respect to fire-produced Cburn . The $a_{(CO \leftarrow (fire-added}$ $CO_2+CO)}$ was described using the Modified Combustion Efficiency,..."

L528-529: I do not understand what is meant by "provides safety against a variable and incompletely described background" or "The median is not affected by undetected changes in background..." A good observation. We also needed to explain why we were concerned about this. I have changed this to "This graph also suggests that if there are more than three tracers (we use 8), then the median of all the estimates, median $(x \grave{I}\acute{C}\_ij\hat{~}0)$, is robust against errors resulting if a tracer j has a variable or poorly described background resulting in $x \grave{I}\acute{C}\_ij\hat{~}0$ at falling distinctly higher or lower than the others. We must be concerned about this since tracers can have occasionally important non-fire sources." Figure 4 is extremely difficult to understand as there is almost no description in the caption itself; the descriptions and explanations are found within the text body. Specifically: The text and images are very small o The label "bscat" in Figure 4a,c is too small o The number labels in Figure 4b,c,e,f are too small There are many individual components that are confusing We have put a lot of time to address this remark. All figures have been redrawn with larger lettering. See above for the wording of the section introduction and the expanded figure caption.

Please also note the supplement to this comment:
https://www.atmos-meas-tech-discuss.net/amt-2019-235/amt-2019-235-AC1-supplement.pdf
* * *

---

## Author Comment (AC2) · 14 Apr 2020

It is difficult to understand how this review came to be written. We the authors sympathize with the reviewer's difficulties, as expressed. The paper has analysis, theory, statistical technique, and some resultant Enrichment Ratios. However, bearing in mind the seven months since the paper's submission, and the 4-5 months since the first review came in, the reviewer is requested to spend perhaps 5%, maybe 1%, of the number of hours expended we expended on this very, very full response. Apparently the reviewer has not even read at the first reviewers courteous and very full and helpful review, and has not attempted wording to counter the positive aspects of that review.

Perhaps the time allotted to get in "just some review" seemed now very short. True, we are all very distracted for the last months. Shouldn't our sheltering time now give this reviewer even more opportunity to attempt some helpful and constructive comments. Following is a text-only version of the response, taken from a table. Figures attached show the very full responses as the original table. The attachment provides this tabular view as well as revised supplementary material which address the matter of creating a direct main paper.

There is no clear outline presented in the manuscript, so I had to create my own (copied below) in order to grasp the manuscript completely. The manuscript includes two separate methodology sections (Section 2 and 6) and two separate theory sections (Section 4 and 5), and their arrangement and transitions left me frequently confused. We have included "An Outline of This Paper" immediately after the first paragraph, following the style suggested by R1. We will reformat this as AMT will ultimately decide. We believe that leads us to omit paragraphs Section 1.1, L120–L140, which are over-detailed superfluous. Additionally, there are many instances of parenthetical asides, notes, and comments (e.g., L362-365, L399-409, L422- 428, all of Section 6.4) that interrupt the flow of the manuscript and greatly impede its overall understandability. These instances and one other have been replaced by a named section of the Supplementary Material, e.g. "See also SM for a Note on Initial Point." at L362. I have attempted to make all such references minimally disruptive to the flow of the paper. Following AMT guidelines, they are not fundamental to advancing the arguments of the paper. The conversational tone of this manuscript additionally introduces confusion. For instance, L203 stats "We now move to..." and it's unclear if this means in the following paragraphs or in the next section. In L312 the phrase "Recall that..." is unclear. These are restated.: L203: The next section provides motivation for and understanding of an alternate approach ... L312: With this section, we illustrate tracer relationships that define our approach to EnRs and EFs in more detail and also in more difficult circumstances, e.g., where the MCE is difficult to estimate, for example because its range of applicability during continued flight sampling is not clear. Also, the included figures

are very difficult to understand, in part because their text, captions, and legends are frequently too small to read (esp. Figures 4, 8, and 9) and because full explanations of what are in the figures are found both within the figure captions themselves and within various portions of the manuscript body. Overall, these makes the manuscript difficult to follow and the presented scientific concepts and results difficult to understand. The figures have been largely redrafted to have larger text. Figures 4 and 9 have been redrafted to show labels more clearly. (An remaining error on some time markings will be corrected.) Explanations of Figure 4 are expanded: Figure 1. (a) Timeline of sampling, for the period shown in Figure 3a, Montana, of CO2+CO (blue, left axis) and the fire tracers CO and bscat (red and green points, right axis). Orange-filled points were identified as clear plume points. Unfilled points were not, but might have some fire influence, especially near plume points. (b) scatter diagram of CO vs CO2+CO with arrows showing the time progression of aircraft sampling of identified plume points. Colors provide a key to times shown in (a). (c) a similar diagram of bscat vs CO2+CO. Similar shapes of figures are noted in the text. (d) Timeline of sampling for the period shown in Figure 3b, Coastal Transect. (e) scatter diagram of CO vs CO2+CO during the transect, like (b). (f) a similar diagram of bscat vs CO2+CO for the Coastal Transect. The black bars graphed in (a) and (d) are estimates of non-fire influenced Cbkgd, see text. They and the non-plume points suggest air-mass changes in CO2+CO. Figure10 has been made larger, and a large display in the published paper is recommended. When points representing different tracers overlap in the figure, this truly signals something about the excellent precision of the individual measurements, and we do not attempt to distinguish them. The figure caption has been expanded: Figure 10. (Lower panel). Estimates of the 422 background x $\grave{I}\acute{C}\_i\hat{}0$ = CO2+CO concentrations implied based on the 10 fire tracers indicated in the legend. Individual x $\grave{I}\acute{C}\_{ij}\hat{}0$ are shown by overlapping colored bars (–), with the median estimate indicated by a black bar. (Upper panel) Estimates of $C\_"burn\ "$ = $x\_i$-x $\grave{I}\acute{C}\_i\hat{}0$ indicators of fuel carbon burned, in green line. A preliminary estimate of $C\_"burn\ "$ based on the consensus of tracer deviations (without variable EnR estimates) is also shown. Flight days

are indicated by the days marked on the top axes, and individual plumes, separated by non-plume concentrations of longer than 10 minutes, are shown as vertical separator lines. A set of horizontal lines at ~400 ppm indicates selected intervals for optimizing numerics (see text, Section 6.3, item 7).. I feel that there are two different manuscripts here, or at least one manuscript with a large appendix or supplement that includes the majority of the theory (Sections 4, 5, pages 12 – 19). The forthcoming paper (Chatfield and Andreae (2019) appears to be a useful companion to this manuscript, and it is referenced several times (e.g., L669-672), but it is unclear if the two papers are meant to be considered together or if they are stand-alone manuscripts. While I believe that this manuscript has significant scientific value and falls within the scope of AMT, and that the work described and methodology proposed (the MERET method) has substantial value, the current structure and length imposes a significant impediment on its understandability and impact. There were many times in which I was confused or lost, and so while I feel like I understand much of what was presented, I am not confident that the manuscript has successfully communicated all that the authors intended. As such, I feel that significant reorganization and clarification is needed before this can be recommended for publication. This is well-considered, but the authors find few other options. We have put much more into the Supplementary Material. Consequently, (a) Material not strictly necessary has been moved to the material. (b) A table of contents has been included, following the reviewer's first comment and suggestion above. (c) The fact that the paper contains a development of plume theory is more prominent in the abstract: A new theoretical development of plume theory for multiple tracers is developed after examining the aircraft samples If the editors of AMT allow, we could change the title to: Theory and Estimation of Emissions Relationships in Forest Firea Plumes: 1: Reducing Effect of Mixing Errors on Emission Factors (d) The authors do not think that the theory could stand alone without showing that it leads to apparently good statistical estimates. and are unwilling to begin the whole AMT review process again if we suggest a division. The scientific value of understanding forest-fire plume properties, and in particular of quantifying the enhancement ratios (EnRs) for properties of interest via

the MERET method, is very high and this manuscript is a significant contribution to the field. The descriptions of the relationships between EnRs, ERs, and EFs in Section 1 is informative, although it would be particularly valuable if additional descriptions of how EnRs "approximate emission ratios (ERs)" (L77) if they are sampled before atmospheric transformations can occur. What is the relation after transformations? This needs to be made clear in the introduction. Besides rewriting the paragraph, we have added a note to the Supplementary Material which clarifies this:' More on the relationships of EnRs, ERs, and EFs is found in the Supplementary Material (SM), "Note on EnRs and ERs". ' The reviewer appears to want more information about when ERs can be larger or smaller than EnRs. This seemed appropriate for a note. A helpful suggestion! The interpretation of Figures 4b,c,e,f in Section 3 is extremely valuable, but I largely struggled with understanding what was being represented until the description of the different examples later in the manuscript (esp. Sections 4.2 and 5). Only on a second read-through was I able to follow the text and more completely understand what is presented in Figure 4. We thank the reviewer for this observation. We have rewritten the introductory paragraph: This section provides some examples of Ctot and fire tracers. It illustrates the limitations of changes in Ctot along a sampling path as an indicator of fire influence, Cburn, for emissions estimation and the much greater similarities of the such changes of tracers that possess shorter transformation timescales. These define our approach to EnRs and EFs. The relation of fire emissions to observed Ctot to Cburn, can be apparently simple or complex, depending on how the history of non-fire CO and CO2 entrained into fire plume air parcels affects Ctot. We show this commonality of relationships will to motivate the theory of expanding plumes in Section 4. That theory will suggest a method worked out in Sections 5 and 6 to find the key variable, Cbkgd, that then provides Cburn and thus EnRs. We have also edited several places succeeding paragraph, not described here.. L54: "Chatfield and Andreae (2017)" should be "Chatfield and Andreae (2019, in preparation)" L66: "DCOtot" should be "DCtot". Table 1: The line labeled "Proportional to carbon burned: define" is confusing. What does define mean here? Is this a typo? Figure 2 refers to a slope

of 32.60458 while the text (L299) refers to a slope of 33x10-3. This inconsistency is confusing. The variable Cj used in L417-418 and other lines does not appear in the Table of Symbols (Table 2) and is only described on L418 L425: "...the same plume. provided we..." is confusing

Figure 6 has an x-axis label of Ctot while the text (L469) refers to Cburn L659: "However, we let the define the types..." seems to be missing a word. I believe "Figure 9" on Line 733 should be "Figure 8" $\sqrt{}$ ïĄĎCtot = ïĄĎCO2 + ïĄĎCO Yes, a typo. Now Proportional to total burned material, as measured by Cburn Chose ppb/ppm rather than ppm/ppm Included.

Changed. Remarks placed in the Supplement. After the equation (12) we now have: For periods of expansion in which the entrained concentrations are constant. See also SM: Note on Varying Entrainment Changed. However, we let the statistical technique define these types, Changed The variable Cj used in L417-418 and other lines does not appear in the Table of Symbols (Table 2) and is only described on L418 L425: "...the same plume. provided we..." is confusing

Figure 6 has an x-axis label of Ctot while the text (L469) refers to Cburn

L659: "However, we let the define the types..." seems to be missing a word.

I believe "Figure 9" on Line 733 should be "Figure 8" This has been added. The variable is the constant of integration and is generally replaced by a_j=expâĄą(C_j )

Now in Supplementary Material, This now uses alpha and beta for different possible positions, values of i, and re-worded "$\alpha$ and $\beta$, in the same plume. These are supposed chosen so that we know that xˆE and all the y_jˆEremain coanstant. Both Ctot and Cburn are used. The x axis has Ctot , units, while the increment beyond the vertical axis at 380, shows Cburn. This is now indicated. "However, we let the statistical technique define these types, and so apply basic clustering techniques." We also added a sentence soon afterward: "NMF and k-means clustering are shown to be

equivalent in cases appropriate to our work (Ding et al., 2005)." Yes, Figure 8, thank you! The phrase "affine dependence" is used several times L522-523: The suggestion that the reader should make their own calculations in order to understand the linear responses is unhelpful. True. "Linear" has several meaning in English. See Wikipedia. So we have now "An affine dependence (linear polynomial relationship including an intercept). Linear Transformations in linear algebra must omit the intercept, hence the unusual phrasing "linear polynomial." Chatfield has considered the ramifications of this dependence NMF linear transformations considerably. The reader is relieved of calculations now: "Some similar calculations make it clear that the estimates respond in an appropriate averaging manner under varied assumptions." We simply emphasize the linearity of the analysis. (e.g., L145) and is unfamiliar to me. In Section 1.2, there are many places where I get lost. For instance, the equation on L168 lacks a sufficient description and I'm unsure what the "aj <− CO" and "aCO <− (fire − added CO2 + CO)" terms mean. I feel a more complete explanation is needed. The use of the variable x for Ctot in Section 1.2 and other places is confusing, especially when Ctot and x are used together (e.g., L153-158). We have added an explanatory phrase: "the slope $a_{(j \leftarrow CO)}$ of the regression estimates of an EnR of the species with respect to CO, multiplied by an attempted very careful estimate of the slope $a_{(CO \leftarrow (fire\text{-}added C\_burn))}$ EnR of CO with respect to fire-produced Cburn . The $a_{(CO \leftarrow (fire\text{-}added CO_2+CO))}$ was described using the Modified Combustion Efficiency,..."

L528-529: I do not understand what is meant by "provides safety against a variable and incompletely described background" or "The median is not affected by undetected changes in background..." A good observation. We also needed to explain why we were concerned about this. I have changed this to "This graph also suggests that if there are more than three tracers (we use 8), then the median of all the estimates, median (x ÌĆ_ij^0 ), is robust against errors resulting if a tracer j has a variable or poorly described background resulting in x ÌĆ_ij^0 at falling distinctly higher or lower than the others. We must be concerned about this since tracers can have occasionally important non-fire sources." Figure 4 is extremely difficult to understand as there is

almost no description in the caption itself; the descriptions and explanations are found within the text body. Specifically: The text and images are very small o The label "bscat" in Figure 4a,c is too small o The number labels in Figure 4b,c,e,f are too small There are many individual components that are confusing We have put a lot of time to address this remark. All figures have been redrawn with larger lettering. See above for the wording of the section introduction and the expanded figure caption.

[Figure]

| | |
|---|---|
| There is no clear outline presented in the manuscript, so I had to create my own (copied below) in order to grasp the manuscript completely. The manuscript includes two separate methodology sections (Section 2 and 6) and two separate theory sections (Section 4 and 5), and their arrangement and transitions left me frequently confused. | *We have included "An Outline of This Paper" immediately after the first paragraph, following the style suggested by R1. We will reformat this as AMT will ultimately decide. We believe that leads us to omit paragraphs Section 1.1, L120–L140, which are over-detailed superfluous.* |
| Additionally, there are many instances of parenthetical asides, notes, and comments (e.g., L362-365, L399-409, L422- 428, all of Section 6.4) that interrupt the flow of the manuscript and greatly impede its overall understandability. | *These instances and one other have been replaced by a named section of the Supplementary Material, e.g. "See also SM for a Note on Initial Point." at L362. I have attempted to make all such references minimally disruptive to the flow of the paper. Following AMT guidelines, they are not fundamental to advancing the arguments of the paper.* |
| The conversational tone of this manuscript additionally introduces confusion. For instance, L203 stats "We now move to..." and it's unclear if this means in the following paragraphs or in the next section. In L312 the phrase "Recall that..." is unclear. | *These are restated.:*
 *L203:* The next section provides motivation for and understanding of an alternate approach …
 *L312:* With this section, we illustrate tracer relationships that define our approach to EnRs and EFs in more detail and also in more difficult circumstances, e.g., where the MCE is difficult to estimate, for example because its range of applicability during continued flight sampling is not clear. |
| Also, the included figures are very difficult to understand, in part because their text, captions, and legends are frequently too small to read (esp. Figures 4, 8, and 9) and because full explanations of what are in the figures are found both within the figure captions themselves and within various portions of the manuscript body.
 Overall, these makes the manuscript difficult to follow and the presented scientific concepts and results difficult to understand. | *The figures have been largely redrafted to have larger text. Figures 4 and 9 have been redrafted to show labels more clearly. (An remaining error on some time markings will be corrected.) Explanations of Figure 4 are expanded:*

 **Figure 1.** (a) Timeline of sampling, for the period shown in Figure 3a, Montana, of $CO_2+CO$ (blue, left axis) and the fire tracers CO and $b_{scat}$ (red and green points, right axis). Orange-filled points were identified as clear plume points. Unfilled points were not, but might have some fire influence, especially near plume points. (b) scatter diagram of CO vs $CO_2+CO$ with |

**Fig. 1.**

| | arrows showing the time progression of aircraft sampling of identified plume points. Colors provide a key to times shown in (a). (c) a similar diagram of $b_{scat}$ vs $CO_2$+CO. Similar shapes of figures are noted in the text. (d) Timeline of sampling for the period shown in Figure 3b, Coastal Transect. (e) scatter diagram of CO vs $CO_2$+CO during the transect, like (b). (f) a similar diagram of $b_{scat}$ vs $CO_2$+CO for the Coastal Transect. The black bars graphed in (a) and (d) are estimates of non-fire influenced $C_{bkgd}$, see text. They and the non-plume points suggest air-mass changes in $CO_2$+CO. |
|---|---|
| | *Figure 10 has been made larger, and a large display in the published paper is recommended. When points representing different tracers overlap in the figure, this truly signals something about the excellent precision of the individual measurements, and we do not attempt to distinguish them. The figure caption has been expanded:* **Figure 10.** (Lower panel). Estimates of the 422 background $\hat{x}_i^0 = CO_2$+CO concentrations implied based on the 10 fire tracers indicated in the legend. Individual $\hat{x}_{ij}^0$ are shown by overlapping colored bars (–), with the median estimate indicated by a black bar. (Upper panel) Estimates of $C_{burn} = x_i - \hat{x}_i^0$ indicators of fuel carbon burned, in green line. A preliminary estimate of $C_{burn}$ based on the consensus of tracer deviations (without variable EnR estimates) is also shown. Flight days are indicated by the days marked on the top axes, and individual plumes, separated by non-plume concentrations of longer than 10 minutes, are shown as vertical separator lines. A set of horizontal lines at ~400 ppm indicates selected intervals for optimizing numerics (see text, Section 6.3, item 7).. |
| I feel that there are two different manuscripts here, or at least one manuscript with a large appendix or supplement that includes the majority of the theory (Sections 4, 5, pages 12 – | *This is well-considered, but the authors find few other options. We have put much more into the Supplementary Material.* |

**Fig. 2.**

| | |
|---|---|
| 19). The forthcoming paper (Chatfield and Andreae (2019) appears to be a useful companion to this manuscript, and it is referenced several times (e.g., L669-672), but it is unclear if the two papers are meant to be considered together or if they are stand-alone manuscripts.

While I believe that this manuscript has significant scientific value and falls within the scope of AMT, and that the work described and methodology proposed (the MERET method) has substantial value, the current structure and length imposes a significant impediment on its understandability and impact. There were many times in which I was confused or lost, and so while I feel like I understand much of what was presented, I am not confident that the manuscript has successfully communicated all that the authors intended. As such, I feel that significant reorganization and clarification is needed before this can be recommended for publication. | *Consequently,*

*(a) Material not strictly necessary has been moved to the material.*

*(b) A table of contents has been included, following the reviewer's first comment and suggestion above.*

*(c) The fact that the paper contains a development of plume theory is more prominent in the abstract:* A new theoretical development of plume theory for multiple tracers is developed after examining the aircraft samples

*If the editors of AMT allow, we could change the title to:* Theory and Estimation of Emissions Relationships in Forest Firea Plumes: 1: Reducing Effect of Mixing Errors on Emission Factors

*(d) The authors do not think that the theory could stand alone without showing that it leads to apparently good statistical estimates. and are unwilling to begin the whole AMT review process again if we suggest a division.* |
| The scientific value of understanding forest-fire plume properties, and in particular of quantifying the enhancement ratios (EnRs) for properties of interest via the MERET method, is very high and this manuscript is a significant contribution to the field. The descriptions of the relationships between EnRs, ERs, and EFs in Section 1 is informative, although it would be particularly valuable if additional descriptions of how EnRs "approximate emission ratios (ERs)" (L77) if they are sampled before atmospheric transformations can occur. What is the relation after transformations? This needs to be made clear in the introduction. | *Besides rewriting the paragraph, we have added a note to the Supplementary Material which clarifies this:* 'More on the relationships of EnRs, ERs, and EFs is found in the Supplementary Material (SM), "Note on EnRs and ERs". ' *The reviewer appears to want more information about when ERs can be larger or smaller than EnRs. This seemed appropriate for a note. A helpful suggestion!* |

**Fig. 3.**

[Figure]

| The interpretation of Figures 4b,c,e,f in Section 3 is extremely valuable, but I largely struggled with understanding what was being represented until the description of the different examples later in the manuscript (esp. Sections 4.2 and 5). Only on a second read-through was I able to follow the text and more completely understand what is presented in Figure 4. | *We thank the reviewer for this observation. We have rewritten the introductory paragraph:*

 This section provides some examples of $C_{tot}$ and fire tracers. It illustrates the limitations of changes in $C_{tot}$ along a sampling path as an indicator of fire influence, $C_{burn}$, for emissions estimation and the much greater similarities of the such changes of tracers that possess shorter transformation time-scales. These define our approach to EnRs and EFs. The relation of fire emissions to observed $C_{tot}$ to $C_{burn}$, can be apparently simple or complex, depending on how the history of non-fire CO and $CO_2$ entrained into fire plume air parcels affects $C_{tot}$. We show this commonality of relationships will to motivate the theory of expanding plumes in Section 4. That theory will suggest a method worked out in Sections 5 and 6 to find the key variable, $C_{bkgd}$, that then provides $C_{burn}$ and thus EnRs.

 *We have also edited several places succeeding paragraph, not described here..* |
|---|---|
| <li>L54: "Chatfield and Andreae (2017)" should be "Chatfield and Andreae (2019, in preparation)"</li><li>L66: "DCO$_{tot}$" should be "DC$_{tot}$".</li><li>Table 1: The line labeled "Proportional to carbon burned: define" is confusing. What does define mean here? Is this a typo?</li><li>Figure 2 refers to a slope of 32.60458 while the text (L299) refers to a slope of 33x10$^{-3}$. This inconsistency is confusing.</li><li>The variable C$_j$ used in L417-418 and other lines does not appear in the Table of Symbols (Table 2) and is only described on L418</li><li>L425: "...the same plume. *provided* we..." is confusing</li> | √

 $\Delta C_{tot} = \Delta CO_2 + \Delta CO$

 *Yes, a typo. Now*
 *Proportional to total burned material, as measured by $C_{burn}$*

 *Chose ppb/ppm rather than ppm/ppm*

 *Included.*

 *Changed. Remarks placed in the Supplement. After the equation (12) we now have:* For periods of expansion in which the entrained |

**Fig. 4.**

| | |
|---|---|
| <li></li><li></li><li>Figure 6 has an x-axis label of $C_{tot}$ while the text (L469) refers to $C_{burn}$</li><li>L659: "However, we let the define the types..." seems to be missing a word.</li><li>I believe "Figure 9" on Line 733 should be "Figure 8"</li> | concentrations are constant. See also SM: Note on Varying Entrainment

*Changed.*

However, we let the statistical technique define these types,

*Changed* |
| <li>The variable $C_j$ used in L417-418 and other lines does not appear in the Table of Symbols</li>
(Table 2) and is only described on L418

<li>L425: "...the same plume. *provided* we..." is confusing</li><li></li>

<li></li><li>Figure 6 has an x-axis label of $C_{tot}$ while the text (L469) refers to $C_{burn}$</li><li></li><li>L659: "However, we let the define the types..." seems to be missing a word.</li>

<li>I believe "Figure 9" on Line 733 should be "Figure 8"</li><li></li> | *This has been added. The variable is the constant of integration and is generally replaced by* $a_j = \exp(C_j)$

*Now in Supplementary Material, This now uses alpha and beta for different possible positions, values of i, and re-worded "α and β, in the same plume. These are supposed chosen so that we know that $x^E$ and all the $y_j^E$ remain coanstant.*

*Both Ctot and Cburn are used. The x axis has Ctot , units, while the increment beyond the vertical axis at 380, shows Cburn. This is now indicated.*

"However, we let the statistical technique define these types, and so apply basic clustering techniques." We also added a sentence soon afterward: "NMF and *k-means* clustering are shown to be equivalent in cases appropriate to our work (Ding et al., 2005)."

*Yes, Figure 8, thank you!* |

**Fig. 5.**

| | |
|---|---|
| • The phrase "affine dependence" is used several times L522-523: The suggestion that the reader should make their own calculations in order to understand the linear responses is unhelpful.

• | *True. "Linear" has several meaning in English. See Wikipedia. So we have now* "An affine dependence (linear polynomial relationship including an intercept). *Linear Transformations in linear algebra must omit the intercept, hence the unusual phrasing "linear polynomial." Chatfield has considered the ramifications of this dependence NMF linear transformations considerably.*

*The reader is relieved of calculations now:*

"Some similar calculations make it clear that the estimates respond in an appropriate averaging manner under varied assumptions." We simply emphasize the linearity of the analysis. |
| • (e.g., L145) and is unfamiliar to me.
• In Section 1.2, there are many places where I get lost. For instance, the equation on L168 lacks a sufficient description and I'm unsure what the "aj <-- CO" and "aCO <-- (fire – added CO2 + CO)" terms mean. I feel a more complete explanation is needed.
• The use of the variable x for Ctot in Section 1.2 and other places is confusing, especially when Ctot and x are used together (e.g., L153-158).

• | *We have added an explanatory phrase:* "the slope $a_{j \leftarrow CO}$ of the regression estimates of an EnR of the species with respect to CO, multiplied by an attempted very careful estimate of the slope $a_{CO \leftarrow (fire-added \, c_{burn})}$ EnR of CO with respect to fire-produced $C_{burn}$. The $a_{CO \leftarrow (fire-added \, CO_2 + CO)}$ was described using the Modified Combustion Efficiency,…" |
| L528-529: I do not understand what is meant by "provides safety against a variable and incompletely described background" or "The median is not affected by undetected changes in background…"

• | *A good observation. We also needed to explain why we were concerned about this. I have changed this to* "This graph also suggests that if there are more than three tracers (we use 8), then the median of all the estimates, median ($\hat{x}_{ij}^0$), is robust against errors resulting if a tracer $j$ has a variable or poorly described background resulting in $\hat{x}_{ij}^0$ at falling distinctly higher or lower than the others. We must be concerned about this since tracers can have occasionally important non-fire sources." |

**Fig. 6.**

[Figure]

| Figure 4 is extremely difficult to understand as there is almost no description in the caption itself; the descriptions and explanations are found within the text body. Specifically:

• The text and images are very small
    ○ The label "$b_{scat}$" in Figure 4a,c is too small
    ○ The number labels in Figure 4b,c,e,f are too small
• There are many individual components that are confusing | *We have put a lot of time to address this remark.*

*All figures have been redrawn with larger lettering. See above for the wording of the section introduction and the expanded figure caption.* |
| --- | --- |
| | |

**Fig. 7.**

---

## Author Response (AR1)

| Reviewer #1 Comments, advice and requests | Authors' full compliance, response, and thanks |
|---|---|
| *Since submitting responses to the reviewers, the authors have made some further changes in the spirit of the requests. A couple of Figure reference numbers have been corrected. They are recurrent features of Microsoft Word idiosyncrasies (sorry).*
 *Format should be more similar to Copernicus standards.* | |
| There is no clear outline presented in the manuscript, so I had to create my own (copied below) in order to grasp the manuscript completely. The manuscript includes two separate methodology sections (Section 2 and 6) and two separate theory sections (Section 4 and 5), and their arrangement and transitions left me frequently confused. | *We have included "An Outline of This Paper" immediately after the first paragraph, following the style suggested by R1. We will reformat this as AMT will ultimately decide. We believe that leads us to omit paragraphs Section 1.1, L120–L140, which are over-detailed superfluous.* |
| Additionally, there are many instances of parenthetical asides, notes, and comments (e.g., L362-365, L399-409, L422- 428, all of Section 6.4) that interrupt the flow of the manuscript and greatly impede its overall understandability. | *These instances and one other have been replaced by a named section of the Supplementary Material, e.g. "See also SM for a Note on Initial Point." at L362. I have attempted to make all such references minimally disruptive to the flow of the paper. Following AMT guidelines, they are not fundamental to advancing the arguments of the paper.* |
| The conversational tone of this manuscript additionally introduces confusion. For instance, L203 stats "We now move to…" and it's unclear if this means in the following paragraphs or in the next section. In L312 the phrase "Recall that…" is unclear. | *These are restated.:*
 *L203:* The next section provides motivation for and understanding of an alternate approach …
 *L312:* With this section, we illustrate tracer relationships that define our approach to EnRs and EFs in more detail and also in more difficult circumstances, e.g., where the MCE is difficult to estimate, for example because its range of applicability during continued flight sampling is not clear. |
| Also, the included figures are very difficult to understand, in part because their text, captions, and legends are frequently too small to read (esp. Figures 4, 8, and 9) and because full explanations of what are in the figures are found both within the figure captions themselves and within various portions of the manuscript body. | *The figures have been largely redrafted to have larger text.*
 *Figures 4 and 9 have been redrafted to show labels more clearly. (An remaining error on some time markings will be corrected.)*
 *Explanations of Figure 4 are expanded:*

 **Figure 4.** (a) Timeline of sampling, for the period shown in Figure 3(a), Montana, of $CO_2$+CO (blue, left axis) and the fire |

Overall, these makes the manuscript difficult to follow and the presented scientific concepts and results difficult to understand.

tracers CO and $b_{scat}$ (red and green points, right axis). Orange-filled points were identified as clear plume points. Unfilled points were not, but might have some fire influence, especially near plume points. (b) scatter diagram of CO vs $CO_2$+CO with arrows showing the time progression of aircraft sampling of identified plume points. Colors provide a key to times shown in (a). Light gray numerals give observation times in minutes. (c) a similar diagram of $b_{scat}$ vs $CO_2$+CO. Similar shapes of figures are noted in the text. (d) Timeline of sampling for the period shown in Figure 3(b), Coastal Transect. (e) scatter diagram of CO vs $CO_2$+CO during the transect, like (b). (f) a similar diagram of $b_{scat}$ vs $CO_2$+CO for the Coastal Transect. The black bars graphed in (a) and (d) are estimates of non-fire influenced $C_{bkgd}$, see text. They and the non-plume points suggest air-mass changes in $CO_2$+CO.

*Figure 9 has similar changes and changes to the caption*
**Figure 9**. Analyzed relation of tracers to carbon burned using the MERET technique for portions of SEAC4RS Flight 10 and ARCTAS Flight 14. Compare (a) through (d) with Figure 4, (b), (c), (e), and (f). Colors key the observations to times shown in the timelines, Figures 4(a) and 4(c). Light gray numerals give observation times in minutes.

*Figure 8 has been made larger, and a large display in the published paper is recommended. When points representing different tracers overlap in the figure, this truly signals something about the excellent precision of the individual measurements, and we do not attempt to distinguish them. The figure caption has been expanded:*
**Figure 8.** (Lower panel). Estimates of the 422 background $\hat{x}_i^0 = CO_2$+CO concentrations implied based on the 8 fire tracers indicated in the legend. Contributing individual estimates $\hat{x}_{ij}^0$ are shown by overlapping colored points, with the median estimate

| | |
|---|---|
| | $\widehat{x}_i^0$ indicated by a black bar. Usually the colored points overlap closely, this indicates strong agreement (Upper panel) Estimates of $C_{\text{burn}} = x_i - \widehat{x}_i^0$ indicators of fuel carbon burned, in the heavy green line. The preliminary estimate of $C_{\text{burn}}$ based on the consensus of tracer deviations (without variable EnR estimates) is also shown in a light line. A scale factor, maximizing overlap with the heavy line, was necessarily estimated by regression. Flight days are indicated by the days marked on the top axes, and individual plumes, separated by non-plume concentrations of longer than 10 minutes, are shown as vertical separator lines. A set of horizontal lines at ~400 ppm indicates selected intervals for optimizing numerics (see text, Section 6.3, item 7). |
| I feel that there are two different manuscripts here, or at least one manuscript with a large appendix or supplement that includes the majority of the theory (Sections 4, 5, pages 12 – 19). The forthcoming paper (Chatfield and Andreae (2019) appears to be a useful companion to this manuscript, and it is referenced several times (e.g., L669-672), but it is unclear if the two papers are meant to be considered together or if they are stand-alone manuscripts.

While I believe that this manuscript has significant scientific value and falls within the scope of AMT, and that the work described and methodology proposed (the MERET method) has substantial value, the current structure and length imposes a significant impediment on its understandability and impact. There were many times in which I was confused or lost, and so while I feel like I understand much of what was presented, I am not confident that the manuscript has successfully communicated all that the authors intended. As such, I feel that | *This is well-considered, but the authors find few other options. We have put much more into the Supplementary Material.*

*Consequently,*

*(a) Material not strictly necessary has been moved to the material.*

*(b) A table of contents has been included, following the reviewer's first comment and suggestion above.*

*(c) The fact that the paper contains a development of plume theory is more prominent in the abstract:* A new theoretical development of plume theory for multiple tracers is developed after examining the aircraft samples

*If the editors of AMT allow, we could change the title to:* Theory and Estimation of Emissions Relationships in Forest Fire Plumes: 1: Reducing Effect of Mixing Errors on Emission Factors |

| | |
|---|---|
| significant reorganization and clarification is needed before this can be recommended for publication. | *(d) The authors do not think that the theory could stand alone without showing that it leads to apparently good statistical estimates. and are unwilling to begin the whole AMT review process again if we suggest a division.* |
| The scientific value of understanding forest-fire plume properties, and in particular of quantifying the enhancement ratios (EnRs) for properties of interest via the MERET method, is very high and this manuscript is a significant contribution to the field. The descriptions of the relationships between EnRs, ERs, and EFs in Section 1 is informative, although it would be particularly valuable if additional descriptions of how EnRs "approximate emission ratios (ERs)" (L77) if they are sampled before atmospheric transformations can occur. What is the relation after transformations? This needs to be made clear in the introduction. | *Besides rewriting the paragraph, we have added a note to the Supplementary Material which clarifies this:'More on the relationships of EnRs, ERs, and EFs is found in the Supplementary Material (SM), "Note on EnRs and ERs". ' The reviewer appears to want more information about when ERs can be larger or smaller than EnRs. This seemed appropriate for a note. A helpful suggestion!* |
| The interpretation of Figures 4b,c,e,f in Section 3 is extremely valuable, but I largely struggled with understanding what was being represented until the description of the different examples later in the manuscript (esp. Sections 4.2 and 5). Only on a second read-through was I able to follow the text and more completely understand what is presented in Figure 4. | *We thank the reviewer for this observation. We have rewritten the introductory paragraph:*

This section provides some examples of $C_{tot}$ and fire tracers. It illustrates the limitations of changes in $C_{tot}$ along a sampling path as an indicator of fire influence, $C_{burn}$, for emissions estimation and the much greater similarities of the such changes of tracers that possess shorter transformation time-scales. These define our approach to EnRs and EFs. The relation of fire emissions to observed $C_{tot}$ to $C_{burn}$, can be apparently simple or complex, depending on how the history of non-fire CO and $CO_2$ entrained into fire plume air parcels affects $C_{tot}$. We show this commonality of relationships will to motivate the theory of expanding plumes in Section 4. That theory will suggest a method worked out in Sections 5 and 6 to find the key variable, $C_{bkgd}$, that then provides $C_{burn}$ and thus EnRs. |

| | |
|---|---|
| | *We have also edited several places succeeding paragraph, not described here..* |
| • L54: "Chatfield and Andreae (2017)" should be "Chatfield and Andreae (2019, in preparation)"
 • L66: "DCO$_{tot}$" should be "DC$_{tot}$".
 • Table 1: The line labeled "Proportional to carbon burned: define" is confusing. What does define mean here? Is this a typo?
 • Figure 2 refers to a slope of 32.60458 while the text (L299) refers to a slope of 33x10$^{-3}$. This inconsistency is confusing.
 • The variable C$_j$ used in L417-418 and other lines does not appear in the Table of Symbols (Table 2) and is only described on L418
 • L425: "...the same plume. *provided* we..." is confusing
 •
 •
 • Figure 6 has an x-axis label of C$_{tot}$ while the text (L469) refers to C$_{burn}$
 • L659: "However, we let the define the types..." seems to be missing a word.
 • I believe "Figure 9" on Line 733 should be "Figure 8" | √

 $\Delta C_{tot} = \Delta CO_2 + \Delta CO$

 *Yes, a typo. Now*
 *Proportional to total burned material, as measured by C$_{burn}$*

 *Chose ppb/ppm rather than ppm/ppm*

 *Included.*

 *Changed. Remarks placed in the Supplement. After the equation (12) we now have:* For periods of expansion in which the entrained concentrations are constant. See also SM: Note on Varying Entrainment

 *Changed.*

 However, we let the statistical technique define these types,

 *Changed* |
| • The variable C$_j$ used in L417-418 and other lines does not appear in the Table of Symbols

 (Table 2) and is only described on L418

 • L425: "...the same plume. *provided* we..." is confusing | *This has been added.  The variable is the constant of integration and is generally replaced by* $a_j = \exp(C_j)$

 *Now in Supplementary Material,   This now uses alpha and beta for different possible positions, values of i, and re-worded "α and β, in the same plume.* |

| | |
|---|---|
| • | *These are supposed chosen so that we know that $x^E$ and all the $y_j^E$ remain coanstant.* |
| • | *Both Ctot and Cburn are used. The x axis has Ctot , units, while the increment beyond the vertical axis at 380, shows Cburn. This is now indicated.* |
| • Figure 6 has an x-axis label of $C_{tot}$ while the text (L469) refers to $C_{burn}$

•

• L659: "However, we let the define the types..." seems to be missing a word.

• I believe "Figure 9" on Line 733 should be "Figure 8"

• | "However, we let the statistical technique define these types, and so apply basic clustering techniques."  We also added a sentence soon afterward: "NMF and *k-means* clustering are shown to be equivalent in cases appropriate to our work (Ding et al., 2005)."

*Yes, Figure 8, thank you!* |
| • The phrase "affine dependence" is used several times L522-523: The suggestion that the reader should make their own calculations in order to understand the linear responses is unhelpful.

• | *True. "Linear" has several meaning in English.  See Wikipedia. So we have now* "An affine dependence (linear polynomial relationship including an intercept). *Linear Transformations in linear algebra must omit the intercept, hence the unusual phrasing "linear polynomial." Chatfield has considered the ramifications of this dependence NMF linear transformations considerably.*

*The reader is relieved of calculations now:*

"Some similar calculations make it clear that the estimates respond in an appropriate averaging manner under varied assumptions." We simply emphasize the linearity of the analysis. |
| • (e.g., L145) and is unfamiliar to me.
• In Section 1.2, there are many places where I get lost. For instance, the equation on L168 lacks a sufficient description and I'm unsure what the "aj <-- CO" and | *We have added an explanatory phrase: "*the slope $a_{j \leftarrow CO}$ of the regression estimates of an EnR of the species with respect to CO, multiplied by an attempted very careful estimate of the slope $a_{CO \leftarrow (fire-added\ c_{burn})}$ EnR of CO with respect to fire- |

| | |
|---|---|
| "aCO <-- (fire – added CO2 + CO)" terms mean. I feel a more complete explanation is needed.

• The use of the variable x for Ctot in Section 1.2 and other places is confusing, especially when Ctot and x are used together (e.g., L153-158).

• | produced $C_{burn}$ . The $a_{CO \leftarrow (fire-added\ CO_2+CO)}$ was described using the Modified Combustion Efficiency,…" |
| L528-529: I do not understand what is meant by "provides safety against a variable and incompletely described background" or "The median is not affected by undetected changes in background…"

• | *A good observation. We also needed to explain why we were concerned about this. I have changed this to* "This graph also suggests that if there are more than three tracers (we use 8), then the median of all the estimates, median $(\hat{x}_{ij}^0)$, is robust against errors resulting if a tracer $j$ has a variable or poorly described background resulting in $\hat{x}_{ij}^0$ at falling distinctly higher or lower than the others. We must be concerned about this since tracers can have occasionally important non-fire sources." |
| Figure 4 is extremely difficult to understand as there is almost no description in the caption itself; the descriptions and explanations are found within the text body. Specifically:

• The text and images are very small
  o The label "b$_{scat}$" in Figure 4a,c is too small
  o The number labels in Figure 4b,c,e,f are too small
• There are many individual components that are confusing | *We have put a lot of time to address this remark.*

*All figures have been redrawn with larger lettering. The sequence of times of observations should be much clearer for Figure 4 and Figure 9. See above for the wording of the section introduction and the expanded figure caption.* |
| *Since submitting responses to the reviewers, the authors have made some further changes in the spirit of the requests.* ||

[revised manuscript text omitted]

Underline color: Auto

| Page 5: [2] Formatted | RChatfield | 4/19/20 4:29:00 PM |
|---|---|---|

Underline color: Auto

| Page 5: [2] Formatted | RChatfield | 4/19/20 4:29:00 PM |
|---|---|---|

Underline color: Auto

| Page 5: [2] Formatted | RChatfield | 4/19/20 4:29:00 PM |
|---|---|---|

Underline color: Auto

| Page 5: [2] Formatted | RChatfield | 4/19/20 4:29:00 PM |
|---|---|---|

Underline color: Auto

| Page 5: [2] Formatted | RChatfield | 4/19/20 4:29:00 PM |
|---|---|---|

Underline color: Auto

| Page 5: [2] Formatted | RChatfield | 4/19/20 4:29:00 PM |
|---|---|---|

Underline color: Auto

| Page 5: [2] Formatted | RChatfield | 4/19/20 4:29:00 PM |
|---|---|---|

Underline color: Auto

| Page 5: [2] Formatted | RChatfield | 4/19/20 4:29:00 PM |
|---|---|---|

Underline color: Auto

| Page 5: [2] Formatted | RChatfield | 4/19/20 4:29:00 PM |
|---|---|---|

Underline color: Auto

| Page 5: [2] Formatted | RChatfield | 4/19/20 4:29:00 PM |
|---|---|---|

Underline color: Auto

| Page 5: [2] Formatted | RChatfield | 4/19/20 4:29:00 PM |
|---|---|---|

Underline color: Auto

| Page 5: [2] Formatted | RChatfield | 4/19/20 4:29:00 PM |
|---|---|---|

Underline color: Auto

| Page 5: [2] Formatted | RChatfield | 4/19/20 4:29:00 PM |
|---|---|---|

Underline color: Auto

| Page 5: [2] Formatted | RChatfield | 4/19/20 4:29:00 PM |
|---|---|---|

Underline color: Auto

| Page 5: [2] Formatted | RChatfield | 4/19/20 4:29:00 PM |
|---|---|---|

Underline color: Auto

| Page 5: [2] Formatted | RChatfield | 4/19/20 4:29:00 PM |
|---|---|---|

Underline color: Auto

| Page 5: [1] Deleted | RChatfield | 4/19/20 4:31:00 PM |
|---|---|---|

| Page 5: [2] Formatted | RChatfield | 4/19/20 4:29:00 PM |

Underline color: Auto

| Page 5: [2] Formatted | RChatfield | 4/19/20 4:29:00 PM |

Underline color: Auto

| Page 5: [2] Formatted | RChatfield | 4/19/20 4:29:00 PM |

Underline color: Auto

| Page 5: [2] Formatted | RChatfield | 4/19/20 4:29:00 PM |

Underline color: Auto

| Page 5: [2] Formatted | RChatfield | 4/19/20 4:29:00 PM |

Underline color: Auto

| Page 5: [2] Formatted | RChatfield | 4/19/20 4:29:00 PM |

Underline color: Auto

| Page 5: [2] Formatted | RChatfield | 4/19/20 4:29:00 PM |

Underline color: Auto

| Page 5: [2] Formatted | RChatfield | 4/19/20 4:29:00 PM |

Underline color: Auto

| Page 5: [2] Formatted | RChatfield | 4/19/20 4:29:00 PM |

Underline color: Auto

| Page 5: [2] Formatted | RChatfield | 4/19/20 4:29:00 PM |

Underline color: Auto

| Page 5: [2] Formatted | RChatfield | 4/19/20 4:29:00 PM |

Underline color: Auto

| Page 5: [2] Formatted | RChatfield | 4/19/20 4:29:00 PM |

Underline color: Auto

| Page 5: [2] Formatted | RChatfield | 4/19/20 4:29:00 PM |

Underline color: Auto

| Page 5: [3] Deleted | RChatfield | 3/25/20 5:22:00 PM |

| Page 5: [4] Formatted | RChatfield | 4/19/20 4:29:00 PM |

Font: Not Italic, Underline color: Auto

| Page 5: [4] Formatted | RChatfield | 4/19/20 4:29:00 PM |

Font: Not Italic, Underline color: Auto

| Page 5: [4] Formatted | RChatfield | 4/19/20 4:29:00 PM |

Font: Not Italic, Underline color: Auto

| Page 5: [5] Deleted | RChatfield | 4/19/20 4:35:00 PM |

| Page 5: [5] Deleted | RChatfield | 4/19/20 4:35:00 PM |

1.2.

| Page 5: [6] Deleted | RChatfield | 4/19/20 7:10:00 PM |
|---|---|---|

| Page 5: [6] Deleted | RChatfield | 4/19/20 7:10:00 PM |
|---|---|---|

| Page 5: [7] Formatted | RChatfield | 4/19/20 4:39:00 PM |
|---|---|---|

Font: Not Italic

| Page 5: [7] Formatted | RChatfield | 4/19/20 4:39:00 PM |
|---|---|---|

Font: Not Italic

| Page 5: [8] Deleted | RChatfield | 4/6/20 2:18:00 PM |
|---|---|---|

| Page 5: [8] Deleted | RChatfield | 4/6/20 2:18:00 PM |
|---|---|---|

| Page 8: [9] Deleted | RChatfield | 4/13/20 10:07:00 PM |
|---|---|---|
| Page 8: [10] Deleted | RChatfield | 4/13/20 11:59:00 PM |
| Page 8: [11] Deleted | RChatfield | 4/2/20 3:31:00 PM |
| Page 8: [12] Deleted | RChatfield | 4/2/20 3:31:00 PM |
| Page 10: [13] Deleted | RChatfield | 4/2/20 3:37:00 PM |

| Page 10: [13] Deleted | RChatfield | 4/2/20 3:37:00 PM |
|---|---|---|

| Page 10: [13] Deleted | RChatfield | 4/2/20 3:37:00 PM |
|---|---|---|

| Page 10: [14] Deleted | RChatfield | 4/12/20 10:38:00 PM |
|---|---|---|

| Page 10: [14] Deleted | RChatfield | 4/12/20 10:38:00 PM |
|---|---|---|

| Page 10: [14] Deleted | RChatfield | 4/12/20 10:38:00 PM |
|---|---|---|

| Page 10: [14] Deleted | RChatfield | 4/12/20 10:38:00 PM |
|---|---|---|

| Page 10: [14] Deleted | RChatfield | 4/12/20 10:38:00 PM |
|---|---|---|

| Page 10: [14] Deleted | RChatfield | 4/12/20 10:38:00 PM |
|---|---|---|

| Page 10: [15] Change | Unknown | |
|---|---|---|

Field Code Changed

| Page 10: [15] Change | Unknown | |
|---|---|---|

Field Code Changed

| Page 10: [16] Deleted | RChatfield | 4/12/20 10:45:00 PM |
|---|---|---|

| Page 10: [16] Deleted | RChatfield | 4/12/20 10:45:00 PM |
|---|---|---|

| Page 10: [17] Change | Unknown | |
|---|---|---|

Field Code Changed

| Page 10: [17] Change | Unknown | |
|---|---|---|

Field Code Changed

| Page 10: [18] Deleted | RChatfield | 4/12/20 11:26:00 PM |
|---|---|---|

| Page 10: [18] Deleted | RChatfield | 4/12/20 11:26:00 PM |
|---|---|---|

| Page 10: [18] Deleted | RChatfield | 4/12/20 11:26:00 PM |
|---|---|---|

| Page 10: [18] Deleted | RChatfield | 4/12/20 11:26:00 PM |
|---|---|---|

| Page 10: [18] Deleted | RChatfield | 4/12/20 11:26:00 PM |
|---|---|---|

| Page 10: [19] Deleted | RChatfield | 4/14/20 12:18:00 AM |
|---|---|---|

| Page 10: [20] Formatted | RChatfield | 4/19/20 10:33:00 PM |
|---|---|---|

Font: Not Bold

| Page 10: [20] Formatted | RChatfield | 4/19/20 10:33:00 PM |
|---|---|---|

Font: Not Bold

| Page 11: [21] Deleted | RChatfield | 4/12/20 12:44:00 PM |
|---|---|---|

| Page 11: [22] Formatted | RChatfield | 4/12/20 12:52:00 PM |
|---|---|---|

Font: Helvetica

| Page 11: [22] Formatted | RChatfield | 4/12/20 12:52:00 PM |
|---|---|---|

Font: Helvetica

| Page 11: [23] Formatted | RChatfield | 4/12/20 12:52:00 PM |
|---|---|---|

Font: Helvetica, Not Italic

| Page 11: [23] Formatted | RChatfield | 4/12/20 12:52:00 PM |
|---|---|---|

Font: Helvetica, Not Italic

| Page 11: [24] Formatted | RChatfield | 4/12/20 12:52:00 PM |
|---|---|---|

Font: Helvetica

| Page 11: [24] Formatted | RChatfield | 4/12/20 12:52:00 PM |
|---|---|---|

Font: Helvetica

| Page 11: [25] Formatted | RChatfield | 4/12/20 12:52:00 PM |
|---|---|---|

Font: Helvetica, Not Italic

| Page 11: [25] Formatted | RChatfield | 4/12/20 12:52:00 PM |
|---|---|---|

Font: Helvetica, Not Italic

| Page 11: [26] Formatted | RChatfield | 4/12/20 12:52:00 PM |
|---|---|---|

Font: Helvetica

| Page 11: [26] Formatted | RChatfield | 4/12/20 12:52:00 PM |
|---|---|---|

Font: Helvetica

| Page 11: [27] Formatted | RChatfield | 4/12/20 12:52:00 PM |
|---|---|---|

Font: Helvetica, Not Italic

| Page 11: [27] Formatted | RChatfield | 4/12/20 12:52:00 PM |
|---|---|---|

Font: Helvetica, Not Italic

| Page 11: [28] Formatted | RChatfield | 4/12/20 12:52:00 PM |
|---|---|---|

Font: Helvetica, Not Italic

| Page 11: [28] Formatted | RChatfield | 4/12/20 12:52:00 PM |
|---|---|---|

Font: Helvetica, Not Italic

| Page 11: [29] Formatted | RChatfield | 4/12/20 12:52:00 PM |
|---|---|---|

Font: Helvetica

| Page 11: [29] Formatted | RChatfield | 4/12/20 12:52:00 PM |
|---|---|---|

Font: Helvetica

| Page 11: [29] Formatted | RChatfield | 4/12/20 12:52:00 PM |
|---|---|---|

Font: Helvetica

| Page 11: [30] Formatted | RChatfield | 4/12/20 12:52:00 PM |
|---|---|---|

Font: Helvetica

| Page 11: [30] Formatted | RChatfield | 4/12/20 12:52:00 PM |
|---|---|---|

Font: Helvetica

| Page 14: [31] Deleted | RChatfield | 3/30/20 3:45:00 PM |
|---|---|---|

| Page 14: [32] Formatted | RChatfield | 3/28/20 3:23:00 PM |
|---|---|---|

Font: 11 pt

| Page 14: [33] Formatted | RChatfield | 3/28/20 3:23:00 PM |
|---|---|---|

Font: 11 pt

| Page 14: [34] Formatted | RChatfield | 3/28/20 3:23:00 PM |
|---|---|---|

Font: 11 pt

| Page 14: [35] Formatted | RChatfield | 3/28/20 3:23:00 PM |
|---|---|---|

Font: 11 pt

| Page 14: [36] Formatted | RChatfield | 3/28/20 3:23:00 PM |
|---|---|---|

Font: 11 pt

| Page 14: [37] Formatted | RChatfield | 3/28/20 3:23:00 PM |
|---|---|---|

Font: 11 pt

| Page 14: [38] Formatted | RChatfield | 3/28/20 3:23:00 PM |
|---|---|---|

Font: 11 pt

| Page 14: [39] Formatted | RChatfield | 3/28/20 3:23:00 PM |
|---|---|---|

Font: 11 pt

| Page 14: [40] Formatted | RChatfield | 3/28/20 3:23:00 PM |
|---|---|---|

Font: 11 pt

| Page 14: [41] Formatted | RChatfield | 3/28/20 3:23:00 PM |
|---|---|---|

Font: 11 pt

| Page 14: [42] Formatted | RChatfield | 3/28/20 3:23:00 PM |
|---|---|---|

Font: 11 pt

| Page 14: [43] Formatted | RChatfield | 3/28/20 3:23:00 PM |
|---|---|---|

Font: 11 pt

| Page 14: [44] Formatted | RChatfield | 3/28/20 3:23:00 PM |
|---|---|---|

Font: 11 pt

| Page 14: [45] Formatted | RChatfield | 3/28/20 3:23:00 PM |
|---|---|---|

Font: 11 pt

| Page 14: [46] Formatted | RChatfield | 3/28/20 3:23:00 PM |
|---|---|---|

Font: 11 pt

| Page 14: [47] Formatted | RChatfield | 3/28/20 3:23:00 PM |
|---|---|---|

Font: 11 pt

| Page 14: [48] Formatted | RChatfield | 3/28/20 3:23:00 PM |
|---|---|---|

Font: 11 pt

| Page 14: [49] Formatted | RChatfield | 3/28/20 3:23:00 PM |
|---|---|---|

Font: 11 pt

| Page 14: [50] Formatted | RChatfield | 3/28/20 3:23:00 PM |
|---|---|---|

Font: 11 pt

| Page 14: [51] Formatted | RChatfield | 3/28/20 3:23:00 PM |
|---|---|---|

Font: 11 pt

| Page 14: [52] Formatted | RChatfield | 3/28/20 3:23:00 PM |
|---|---|---|

Font: 11 pt

| Page 14: [53] Formatted | RChatfield | 3/28/20 3:23:00 PM |
|---|---|---|

Font: 11 pt

| Page 14: [54] Formatted | RChatfield | 3/28/20 3:23:00 PM |
|---|---|---|

Font: 11 pt

| Page 14: [55] Formatted | RChatfield | 3/28/20 3:23:00 PM |
|---|---|---|

Font: 11 pt

| Page 14: [56] Formatted | RChatfield | 3/28/20 3:23:00 PM |
|---|---|---|

Font: 11 pt

| Page 14: [57] Deleted | RChatfield | 4/19/20 10:34:00 PM |
|---|---|---|

| Page 27: [58] Deleted | RChatfield | 4/19/20 8:59:00 PM |
|---|---|---|

| Page 28: [59] Deleted | RChatfield | 4/19/20 10:37:00 PM |
|---|---|---|

| Page 28: [60] Deleted | RChatfield | 4/13/20 12:48:00 AM |
|---|---|---|

| Page 28: [60] Deleted | RChatfield | 4/13/20 12:48:00 AM |
|---|---|---|

| Page 29: [61] Deleted | RChatfield | 4/14/20 12:21:00 AM |
|---|---|---|

---

## Referee Report (RR1)

AMT Review Round 2

While this manuscript is much improved, and many things that were unclear in the first manuscript are addressed, I still find many sections to be confusing and unclear. While I feel that the scientific value of this work is important, I also feel that a reader unfamiliar with the material and notation would not be able to follow this manuscript as written. The style of the presentation frequently makes statements that may be clear to the authors, but were not clear to me, and I believe will not be clear to many readers. As such, I do not believe this revised manuscript is suitable for publication until additional revisions are made that clarify the confusing aspect related to notation and equations (some of which are included in the Detailed Notes below).

In particular, while the notation and text has been improved, there are still inconsistencies in the notation that inhibit a reader's ability to follow the text and equations as well as many typos (some examples of these are noted in the Detailed Notes below). There are also many asides and tangents (such as the parenthetical in Section 9.1, L895-897 and the 8th point in Section 6.3) that, while interesting, detract from the overall message of this manuscript.

Additionally, some of the language in this manuscript lacks specificity. The word "appropriate" is used many times without much indication of what appropriate means (L19, L87, L493, L527, L537, L705, L871, L874). Similarly, the word "reasonable" is used (L190, L424, L476), but there is no discussion of what "reasonable" means in the context. On L178, the phrase "attempted very careful estimate" is confusing, and I'm not sure what was attempted and what criteria was used. There are other examples throughout the manuscript that I found confusing rather than clarifying.

**Detailed Notes**

L13: "US." should be "U.S."

L55: What do you mean by "Illustrations show that the theory is robust."? Why is this text a different color?

L89-101: The notation is inconsistent and confusing. Both $C_{tot}$ and $\Delta C_{tot}$ are used, and $C_{burn}$ has different definitions (L89: "$C_{burn} = \Delta CO_2 + \Delta CO + ...$" ; L93: "$C_{burn} = CO_2 + CO + ...$"). The terminology in this manuscript is already complex, so inconsistencies like this make it very hard to follow.

L164: The shift to x and y notation is also confusing. Here the authors define x as $C_{tot}$, while earlier in L89 $\Delta x$ is defined as $C_{burn}$. Is this intentional?

L175-182: I still don't understand the $a_{j<--CO}$, $a_{CO<--(fire-added\ Cburn)}$, and $a_{CO<--(fire-added\ CO2+CO)}$ notation, and feel that this section is still not clear. It seems to rely heavily on previous knowledge of the notation and material and does not present it clearly enough for someone new to the material to follow or understand.

Figure 2: The grey line should be labeled or mentioned in the caption. Otherwise readers are left to infer its meaning.

L313: "and the much greater similarities of the such changes of tracers" – A typo?

Figure 6: The x-axis is labelled as $C_{tot}$, but in L474, it is referred to as $x = C_{burn}$. Which is it?

L516: There is no Equation 13.

L516: The term $y_a$ is used here, but is not used anywhere else, nor is in the Table of Symbols, so even one following the notation and equations closely is likely to get lost here (or at least confused and left to their own interpretation).

L543-545: The sentence that starts with "A term $c_j$ is included..." doesn't provide enough information as to why $c_j$ should be zero, and mentions "inadequacies in our assumptions" but doesn't explore these. This is confusing and needs further explanation.

Figure 7: This figure still confuses me. I'm still not clear what the dashed lines mean. From an initial glance, it looks like the points on the $CO_2$, $CH_3CN$, Toluene = 0 line were established, and then lines to each of the CO, $CH_3CN$, and Toluene points drawn. I understand this is not the intention, but it's difficult to understand. Additionally, $\mathbf{x^0}$ is mentioned in the caption, but is not located in the figure. The caption mentions an "idealized example" but gives no explanation of what this example is, leaving readers to struggle to understand on their own. Also, what's the "$x=C_{tot}$, ppm" label at the top?

L676: The parenthetical that starts on this line does not close with a ")" anywhere.

Section 6.3: This 8 step list summarizing MERET is not clear and I'm not sure someone trying to duplicate your method would be able to follow it. Specific notes are below:
- Step 3: the notation "I" in "for each plume I" is never used again. How many are "enough samples"? Why do you select a plume "of > 4 points"? You say "subtract this from the $x_i$ values," but in this summary you have not stated $x_i$ yet. This is confusing and unclear.
- Step 4: How do you normalize the tracers? Not enough detail here.
- Step 6: What's a_hat$_{jj}$? is that a typo? There are multiple typos in this step. And this step includes unnecessary asides (indeed, it's longer than the previous 5 steps). I don't think this is necessary and hinders clarity.
- Step 7: The final sentence (with the explanation mark) has a tone that feels out of place.
- Step 8: This isn't really a step, it's more of a note.
- I think a flow chart or some sort of process diagram would be more helpful here.

L702: What are the impacts of the regression being "very over-determined"? This is concerning, as over-determined models are useful for specific situations and not at all for others. This needs to be addressed.

L723-725: $\xi$ is not defined (nor used outside of this section), again leaving the reader with the responsibility on inferring meaning. In general, I found this discussion of the "effective time between independent observations" unclear.

Figure 8: Much of the text of this figure is still too small to read (e.g., what's in the lower right corner?), and the figure overall I don't feel is of publication quality. There are typos e.g. the upper panel legend includes "Consensus (initial est". Also, there unexplained details: What are the numbers at the top and how do they correspond to the vertical lines? Why are there thick and thin vertical lines? How to they correspond to the horizontal lines separating the upper and lower panels, and what do these horizontal lines indicate? What is "Abs_5"? What is "Scat_5?" Why does the left-most axis extend all the way to the top of the figure?

---

## Editor Decision (ED1)

Line 13 U.S.. –eliminate second period

Line 78: Results: Estimation of xi0 and Cburn (use proper super and subscripts)

Line 139: In a later part of the campaign, the DC-8 sampled in Northern Canada (Simpson et al., 2011); we excluded these plumes as representing different , more boreal, forest burn conditions. (space before comma)

Table 1: Prefer to not use " in table

Line 364: needs a period. They and the non-plume points suggest air-mass changes in CO2+CO

Table 2: Periods sometimes in "signifies" descriptor, sometimes not. Make consistent.

Table 2:

expansion and rise.  may be assumed, to be  of the estimated as $\hat{x}_i^{U}$ or casually as $x_i^0$. This is not necessarily air surrounding the plume sample.!

Caps on "May". Not sure I understand the sentence that begins with "May". Eliminate exclamation point at end.

Table 2 continued:

or          Background concentration of tracer $j$ . Typically estimated as a minimum value from observed probability density function for samples in a particular flight intensive, especially non-plume samples without signals of stratospheric air.

or? (must be missing something here)

What does this mean? (Table 2):

An early approximation to implied from normalized and scaling, not
required by algorithm but a convenient check

Table 2: - font change mid sentence

$y_{ij}^{U}$          $y$ intercept implied by $x_i$ , $y_{ij}$ and the estimated slopes $a_j$ for $j$

Line 736: 6.3. Summary of the MERET algorithm and notes – not consistent with table of contents. (6.3. Summary of the MERET method)

Line 923: 8.1 Table of several significant emissions (repeat of 8.1 should be labeled 8.2)

---

## Author Response (AR2)

| | |
|---|---|
| ... additional revisions are made that clarify the confusing aspects related to notation and equations. In particular, he felt that while the notation and text has been improved, there are still inconsistencies in the notation that inhibit a reader's ability to follow the text and equations as well as many typos (some examples of these are noted in the Detailed Notes below). There are also many asides and tangents (such as the parenthetical in Section 9.1, L895-897 and the 8th point in Section 6.3) that, while interesting, detract from the overall message of this manuscript. He feels that the style of the presentation frequently makes statements that may be clear to the authors, but were not clear to him, and believes will not be clear to many readers. | √ |
| Additionally, the reviewer felt that some of the language in this manuscript lacks specificity. The word "appropriate" is used many times without much indication of what appropriate means (L19, L87, L493, L527, L537, L705, L871, L874). Similarly, the word "reasonable" is used (L190, L424, L476), but there is no discussion of what "reasonable" means in the context. On L178, the phrase "attempted very careful estimate" is confusing, and I'm not sure what was attempted and what criteria was used. There are other examples throughout the manuscript that he found confusing rather than clarifying. | √ rappropriate, reasonable: emoved, other wording Made similar changes, |
| **Detailed Notes**
L13: "US." should be "U.S." | √ |
| L55: What do you mean by "Illustrations show that the theory is robust."? Why is this text a different color? | √ Accident, restated |
| L89-101: The notation is inconsistent and confusing. Both $C_{tot}$ and $\Delta C_{tot}$ are used, and $C_{burn}$ hasdifferent definitions (L89: "$C_{burn} = \Delta CO_2 + \Delta CO + ...$" ; L93: "$C_{burn} = CO_2 + CO + ...$"). The terminology in this manuscript is already complex, so inconsistencies like this make it very hard to follow.
L164: The shift to x and y notation is also confusing. Here the authors define x as $C_{tot}$, whileearlier in L89 $\Delta x$ is defined as $C_{burn}$. Is this intentional? | √ completely reworded, x and Cburn notation explained at first use |
| L175-182: I still don't understand the $a_{j \leftarrow CO}$, $a_{CO \leftarrow (fire\text{-}added\ Cburn)}$, and $a_{CO \leftarrow (fire\text{-}added\ CO2+CO)}$ notation, and feel that this section is still not clear. It seems to rely heavily on previous knowledge of the notation and material and does not present it clearly enough for someone new to the material to follow or understand. | √ Redone |
| Figure 2: The grey line should be labeled or mentioned in the caption. Otherwise readers are left to infer its meaning. | √ explained |
| L313: "and the much greater similarities of the such changes of tracers" – A typo? | √ |

| | |
|---|---|
| Figure 6: The x-axis is labelled as $C_{tot}$, but in L474, it is referred to as $x = C_{burn}$. Which is it? | √ Explained inf figure and caption |
| L516: There is no Equation 13. | √ |
| L516: The term $y_a$ is used here, but is not used anywhere else, nor is in the Table of Symbols, so even one following the notation and equations closely is likely to get lost here (or at least confused and left to their own interpretation). | √ chanted, vE and y0 relation describeds |
| L543-545: The sentence that starts with "A term $c_j$ is included..." doesn't provide enough information as to why $c_j$ should be zero, and mentions "inadequacies in our assumptions" but doesn't explore these. This is confusing and needs further explanation. | √ explained |
| Figure 7: This figure still confuses me. I'm still not clear what the dashed lines mean. From an initial glance, it looks like the points on the $CO_2$, $CH_3CN$, Toluene = 0 line were established, and then lines to each of the CO, $CH_3CN$, and Toluene points drawn. I understand this is not the intention, but it's difficult to understand. Additionally, $x_0$ is mentioned in the caption, but is not located in the figure. The caption mentions an "idealized example" but gives no explanation of what this example is, leaving readers to struggle to understand on their own. Also, what's the "$x=C_{tot}$, ppm" label at the top? | √ Figure has confusing extended lines removed, notation added, triangles added, refer to Fig.6 describecd |
| L676: The parenthetical that starts on this line does not close with a ")" anywhere. Section 6.3: This 8 step list summarizing MERET is not clear and I'm not sure someone trying to duplicate your method would be able to follow it. Specific notes are below: | √ 8 steps revised and keyed to NEW Fig. 8 |
| • Step 3: the notation "I" in "for each plume I" is never used again. How many are "enough samples"? Why do you select a plume "of > 4 points"? You say "subtract this from the $x_i$ values," but in this summary you have not stated $x_i$ yet. This is confusing and unclear. | √ rewritten, simplified |
| • Step 4: How do you normalize the tracers? Not enough detail here. | √ Equation added |
| • Step 6: What's $a\_hat_{jj}$? is that a typo? There are multiple typos in this step. And this step includes unnecessary asides (indeed, it's longer than the previous 5 steps). I don't think this is necessary and hinders clarity. | √ Fixed |

| | |
|---|---|
| • Step 7: The final sentence (with the explanation mark) has a tone that feels out of place. | √ Rewritten |
| • Step 8: This isn't really a step, it's more of a note. | √ New steps and figure and notes |
| • I think a flow chart or some sort of process diagram would be more helpful here. | √ Added, new Fig. 8 |
| L702: What are the impacts of the regression being "very over-determined"? This is concerning, as over-determined models are useful for specific situations and not at all for others. This needs to be addressed. | √ rewritten: Over-determined is GOOD. |
| L723-725: $\xi$ is not defined (nor used outside of this section), again leaving the reader with the responsibility on inferring meaning. In general, I found this discussion of the "effective time between independent observations" unclear. | √ defined |
| Figure 8: Much of the text of this figure is still too small to read (e.g., what's in the lower right corner?), and the figure overall I don't feel is of publication quality. There are typos e.g. the upper panel legend includes "Consensus (initial est". Also, there unexplained details: What are the numbers at the top and how do they correspond to the vertical lines? Why are there thick and thin vertical lines? How to they correspond to the horizontal lines separating the upper and lower panels, and what do these horizontal lines indicate? What is "Abs_5"? What is "Scat_5?" Why does the left-most axis extend all the way to the top of the figure? | √ redrawn, larger text. |
| In abstract: In summary, MERET allows fine spatial resolution (EnRs for individual observations) and comparison of similar plumes distant in time and space. What does this sentence mean? Can it be reworded? | √ rworded |
| Line 43: Recommend change to: In the first approach, measurements on the ground close to an open fire or on laboratory fires that are controlled to approximate natural conditions, can provide the most detailed information on sources. | √ |
| Line 48: Recommend change to: In the second, measurements made from aircraft, provide a much wider sampling of different fires and emissions from different regions of a single fire. | √ |
| Line 56: This theory implies (? Is this really the word you want here?) a statistical regression technique; a second methodology section then gives details of implementation | √ reworded for more mathematically thorough attention in estimation |
| Line 58: the atmospheric signal of fuel burned, $CO_2$+CO; (this is only an approximation) | √ All rewritten, duplication removed |

| | |
|---|---|
| Line 61: Since this work takes a complex path winding through observations, simple theory, examples of regression methodology, | √ |
| Line 126: This important task is beyond reasonable treatment in this publication, which focuses on the improving the understanding of airborne samples. (first eliminate the "the" in red – second restate – understanding what exactly?) | √ |
| 133: These also should have a direct relation to the fuel carbon burned and other properties such as burning conditions, fuel moisture, and fuel $N$ content. | √ |
| 139: In a later part of the campaign, the DC-8 sampled in Northern Canada (Simpson et al., 2011); we excluded these plumes as representing different forest burn conditions. (what does this mean?) | √ boreal/eemerate |
| 238: Section 7 provides a limited number or EnR estimates and describes graphically how flight segments describing similar emissions conditions can be identi240 | √ |
| 265: In examining EnRs for various species, we also use the organic aerosol (OA) measurements of Jose Jimenez and ionic composition information from the Jack Dibb group. (Not a standard way to present this – use references, Jimenez et al., …) | √ Jourrnal ref instesd |
| 274: frequent) can samples for the first flights prior to the Rim Fire of 26August, and | √ |
| From Table 1: Wisthaler, et all. 2013. (appears 4 times) | √ |
| 318: That theory will suggest a method worked out in Sections 5 and 6 to find the key variable, $C_{bkgd}$, that then provides $C_{burn}$ and thus EnRs. (Clumsy language) | √ |
| Line 538: Methane in particular have long atmospheric lifetime and several sources; consequently, it can exhibit variations that are more than 10% of the fire emis (change –e.g., "has a long") | √ |
| 701: Plumes were characterized by levels of $CH_3CN$ above 0.225 ppb, over four times the assumed background of 0.054 ppb. (Where did you get this number?) Seems significantly low | ¥ lower value Need to be clearly above locally sampled background |

[revised manuscript text omitted]

Highlight

| Page 15: [2] Formatted | AugRev | 8/10/20 4:15:00 PM |
| --- | --- | --- |

Highlight

| Page 15: [2] Formatted | AugRev | 8/10/20 4:15:00 PM |
| --- | --- | --- |

Highlight

| Page 15: [2] Formatted | AugRev | 8/10/20 4:15:00 PM |
| --- | --- | --- |

Highlight

| Page 15: [3] Deleted | AugRev | 8/10/20 4:15:00 PM |
| --- | --- | --- |

| Page 15: [3] Deleted | AugRev | 8/10/20 4:15:00 PM |
| --- | --- | --- |

| Page 15: [3] Deleted | AugRev | 8/10/20 4:15:00 PM |
| --- | --- | --- |

| Page 15: [4] Formatted | AugRev | 8/10/20 4:15:00 PM |
| --- | --- | --- |

Font: 12 pt

| Page 15: [4] Formatted | AugRev | 8/10/20 4:15:00 PM |
| --- | --- | --- |

Font: 12 pt

| Page 15: [4] Formatted | AugRev | 8/10/20 4:15:00 PM |
| --- | --- | --- |

Font: 12 pt

| Page 15: [4] Formatted | AugRev | 8/10/20 4:15:00 PM |
| --- | --- | --- |

Font: 12 pt

| Page 15: [4] Formatted | AugRev | 8/10/20 4:15:00 PM |
| --- | --- | --- |

Font: 12 pt

| Page 15: [4] Formatted | AugRev | 8/10/20 4:15:00 PM |
| --- | --- | --- |

Font: 12 pt

| Page 15: [4] Formatted | AugRev | 8/10/20 4:15:00 PM |
| --- | --- | --- |

Font: 12 pt

| Page 15: [4] Formatted | AugRev | 8/10/20 4:15:00 PM |
| --- | --- | --- |

Font: 12 pt

| Page 15: [4] Formatted | AugRev | 8/10/20 4:15:00 PM |
| --- | --- | --- |

Font: 12 pt

| Page 15: [4] Formatted | AugRev | 8/10/20 4:15:00 PM |
| --- | --- | --- |

Font: 12 pt

| Page 15: [4] Formatted | AugRev | 8/10/20 4:15:00 PM |
|---|---|---|

Font: 12 pt

| Page 15: [4] Formatted | AugRev | 8/10/20 4:15:00 PM |
|---|---|---|

Font: 12 pt

| Page 15: [4] Formatted | AugRev | 8/10/20 4:15:00 PM |
|---|---|---|

Font: 12 pt

| Page 15: [4] Formatted | AugRev | 8/10/20 4:15:00 PM |
|---|---|---|

Font: 12 pt

| Page 15: [4] Formatted | AugRev | 8/10/20 4:15:00 PM |
|---|---|---|

Font: 12 pt

| Page 15: [4] Formatted | AugRev | 8/10/20 4:15:00 PM |
|---|---|---|

Font: 12 pt

| Page 15: [4] Formatted | AugRev | 8/10/20 4:15:00 PM |
|---|---|---|

Font: 12 pt

| Page 15: [4] Formatted | AugRev | 8/10/20 4:15:00 PM |
|---|---|---|

Font: 12 pt

| Page 15: [4] Formatted | AugRev | 8/10/20 4:15:00 PM |
|---|---|---|

Font: 12 pt

| Page 15: [4] Formatted | AugRev | 8/10/20 4:15:00 PM |
|---|---|---|

Font: 12 pt

| Page 15: [4] Formatted | AugRev | 8/10/20 4:15:00 PM |
|---|---|---|

Font: 12 pt

| Page 15: [4] Formatted | AugRev | 8/10/20 4:15:00 PM |
|---|---|---|

Font: 12 pt

| Page 15: [4] Formatted | AugRev | 8/10/20 4:15:00 PM |
|---|---|---|

Font: 12 pt

| Page 15: [4] Formatted | AugRev | 8/10/20 4:15:00 PM |
|---|---|---|

Font: 12 pt

| Page 15: [4] Formatted | AugRev | 8/10/20 4:15:00 PM |
|---|---|---|

Font: 12 pt

| Page 15: [4] Formatted | AugRev | 8/10/20 4:15:00 PM |
|---|---|---|

Font: 12 pt

| Page 15: [4] Formatted | AugRev | 8/10/20 4:15:00 PM |
|---|---|---|

Font: 12 pt

| Page 15: [4] Formatted | AugRev | 8/10/20 4:15:00 PM |
|---|---|---|

Font: 12 pt

| Page 16: [5] Formatted Table | RBChatfield | 8/12/20 11:15:00 AM |
|---|---|---|

Formatted Table

| Page 16: [6] Formatted Table | RBChatfield | 8/12/20 11:15:00 AM |
|---|---|---|

Formatted Table

| Page 16: [7] Formatted | AugRev | 8/10/20 4:15:00 PM |
|---|---|---|

Font: 10 pt, Not Bold, Italic

| Page 16: [8] Formatted | AugRev | 8/10/20 4:15:00 PM |
|---|---|---|

Left

| Page 16: [9] Formatted | AugRev | 8/10/20 4:15:00 PM |
|---|---|---|

Left

| Page 16: [10] Formatted | AugRev | 8/10/20 4:15:00 PM |
|---|---|---|

Font: 12 pt, Not Bold

| Page 16: [11] Formatted | AugRev | 8/10/20 4:15:00 PM |
|---|---|---|

Font: Not Bold

| Page 16: [12] Deleted | AugRev | 8/10/20 4:15:00 PM |
|---|---|---|
| Page 16: [13] Inserted Cells | AugRev | 8/10/20 4:15:00 PM |

Inserted Cells

| Page 16: [14] Inserted Cells | AugRev | 8/10/20 4:15:00 PM |
|---|---|---|

Inserted Cells

| Page 16: [15] Formatted | AugRev | 8/10/20 4:15:00 PM |
|---|---|---|

Formatted

| Page 16: [16] Formatted | AugRev | 8/10/20 4:15:00 PM |
|---|---|---|

Font: Times New Roman

| Page 16: [17] Formatted | AugRev | 8/10/20 4:15:00 PM |
|---|---|---|

Formatted

| Page 16: [18] Deleted | AugRev | 8/10/20 4:15:00 PM |
|---|---|---|

| Page 16: [18] Deleted | AugRev | 8/10/20 4:15:00 PM |
|---|---|---|
| Page 16: [19] Formatted | AugRev | 8/10/20 4:15:00 PM |

Formatted

| Page 16: [20] Formatted | AugRev | 8/10/20 4:15:00 PM |
|---|---|---|

Font: Times New Roman

| Page 16: [21] Deleted | AugRev | 8/10/20 4:15:00 PM |
|---|---|---|
| Page 16: [22] Formatted | AugRev | 8/10/20 4:15:00 PM |

Border: : (No border)

| Page 16: [23] Formatted | AugRev | 8/10/20 4:15:00 PM |
|---|---|---|

Formatted

| Page 16: [24] Formatted | AugRev | 8/10/20 4:15:00 PM |
|---|---|---|

Formatted

| Page 16: [25] Formatted | AugRev | 8/10/20 4:15:00 PM |
|---|---|---|

Formatted

| Page 16: [26] Formatted | AugRev | 8/10/20 4:15:00 PM |
|---|---|---|

Formatted

| Page 16: [27] Formatted | AugRev | 8/10/20 4:15:00 PM |
|---|---|---|

Formatted

| Page 16: [28] Formatted | AugRev | 8/10/20 4:15:00 PM |
|---|---|---|

Formatted

| Page 16: [29] Formatted | AugRev | 8/10/20 4:15:00 PM |
|---|---|---|

Formatted

| _or_ | Background concentration of tracer $j$ . Typically estimated as a mini-mum value from observed probability density function for samples in a particular flight intensive, especially non-plume samples without signals of stratospheric air. | $E$ |
|---|---|---|

| Page 16: [30] Deleted | AugRev | 8/10/20 4:15:00 PM |
|---|---|---|
| Page 16: [31] Formatted | AugRev | 8/10/20 4:15:00 PM |

Centered

| Page 16: [32] Formatted | AugRev | 8/10/20 4:15:00 PM |
|---|---|---|

Font: 10 pt

| Page 16: [32] Formatted | AugRev | 8/10/20 4:15:00 PM |
|---|---|---|

Font: 10 pt

| Page 16: [32] Formatted | AugRev | 8/10/20 4:15:00 PM |
|---|---|---|

Font: 10 pt

| Page 16: [33] Formatted | AugRev | 8/10/20 4:15:00 PM |
|---|---|---|

Font: 10 pt

| Page 16: [33] Formatted | AugRev | 8/10/20 4:15:00 PM |
|---|---|---|

Font: 10 pt

_or_         Background concentration of tracer $j$ . Typically estimated as a mini-
mum value from observed probability density function for samples in
a particular flight intensive, especially non-plume samples without
signals of stratospheric air.         $E$

| Page 16: [34] Deleted | AugRev | 8/10/20 4:15:00 PM |
|---|---|---|

| Page 16: [35] Formatted | AugRev | 8/10/20 4:15:00 PM |
|---|---|---|

Centered

| Page 19: [36] Formatted | AugRev | 8/10/20 4:15:00 PM |
|---|---|---|

Font: 12 pt

| Page 19: [36] Formatted | AugRev | 8/10/20 4:15:00 PM |
|---|---|---|

Font: 12 pt

| Page 19: [36] Formatted | AugRev | 8/10/20 4:15:00 PM |
|---|---|---|

Font: 12 pt

| Page 19: [36] Formatted | AugRev | 8/10/20 4:15:00 PM |
|---|---|---|

Font: 12 pt

| Page 19: [36] Formatted | AugRev | 8/10/20 4:15:00 PM |
|---|---|---|

Font: 12 pt

| Page 19: [36] Formatted | AugRev | 8/10/20 4:15:00 PM |
|---|---|---|

Font: 12 pt

| Page 19: [36] Formatted | AugRev | 8/10/20 4:15:00 PM |
|---|---|---|

Font: 12 pt

| Page 19: [36] Formatted | AugRev | 8/10/20 4:15:00 PM |
|---|---|---|

Font: 12 pt

| Page 19: [36] Formatted | AugRev | 8/10/20 4:15:00 PM |
|---|---|---|

Font: 12 pt

| Page 19: [36] Formatted | AugRev | 8/10/20 4:15:00 PM |
|---|---|---|

Font: 12 pt

| Page 19: [36] Formatted | AugRev | 8/10/20 4:15:00 PM |
|---|---|---|

Font: 12 pt

| Page 19: [36] Formatted | AugRev | 8/10/20 4:15:00 PM |
|---|---|---|

Font: 12 pt

| Page 19: [36] Formatted | AugRev | 8/10/20 4:15:00 PM |
|---|---|---|

Font: 12 pt

| Page 19: [36] Formatted | AugRev | 8/10/20 4:15:00 PM |
|---|---|---|

Font: 12 pt

| Page 19: [36] Formatted | AugRev | 8/10/20 4:15:00 PM |

Font: 12 pt

| Page 19: [36] Formatted | AugRev | 8/10/20 4:15:00 PM |

Font: 12 pt

| Page 19: [36] Formatted | AugRev | 8/10/20 4:15:00 PM |

Font: 12 pt

| Page 19: [36] Formatted | AugRev | 8/10/20 4:15:00 PM |

Font: 12 pt

| Page 19: [37] Formatted | AugRev | 8/10/20 4:15:00 PM |

Font: 12 pt

| Page 19: [37] Formatted | AugRev | 8/10/20 4:15:00 PM |

Font: 12 pt

| Page 19: [37] Formatted | AugRev | 8/10/20 4:15:00 PM |

Font: 12 pt

| Page 19: [37] Formatted | AugRev | 8/10/20 4:15:00 PM |

Font: 12 pt

| Page 19: [37] Formatted | AugRev | 8/10/20 4:15:00 PM |

Font: 12 pt

| Page 19: [37] Formatted | AugRev | 8/10/20 4:15:00 PM |

Font: 12 pt

| Page 19: [37] Formatted | AugRev | 8/10/20 4:15:00 PM |

Font: 12 pt

| Page 19: [37] Formatted | AugRev | 8/10/20 4:15:00 PM |

Font: 12 pt

| Page 19: [38] Formatted | AugRev | 8/10/20 4:15:00 PM |

Font: 12 pt

| Page 19: [38] Formatted | AugRev | 8/10/20 4:15:00 PM |

Font: 12 pt

| Page 19: [38] Formatted | AugRev | 8/10/20 4:15:00 PM |

Font: 12 pt

| Page 19: [38] Formatted | AugRev | 8/10/20 4:15:00 PM |

Font: 12 pt

| Page 19: [38] Formatted | AugRev | 8/10/20 4:15:00 PM |

Font: 12 pt

| Page 19: [38] Formatted | AugRev | 8/10/20 4:15:00 PM |

Font: 12 pt

| Page 19: [38] Formatted | AugRev | 8/10/20 4:15:00 PM |

Font: 12 pt

| Page 19: [38] Formatted | AugRev | 8/10/20 4:15:00 PM |
|---|---|---|

Font: 12 pt

| Page 19: [38] Formatted | AugRev | 8/10/20 4:15:00 PM |
|---|---|---|

Font: 12 pt

| Page 19: [38] Formatted | AugRev | 8/10/20 4:15:00 PM |
|---|---|---|

Font: 12 pt

| Page 19: [38] Formatted | AugRev | 8/10/20 4:15:00 PM |
|---|---|---|

Font: 12 pt

| Page 19: [38] Formatted | AugRev | 8/10/20 4:15:00 PM |
|---|---|---|

Font: 12 pt

| Page 19: [38] Formatted | AugRev | 8/10/20 4:15:00 PM |
|---|---|---|

Font: 12 pt

| Page 19: [38] Formatted | AugRev | 8/10/20 4:15:00 PM |
|---|---|---|

Font: 12 pt

| Page 19: [38] Formatted | AugRev | 8/10/20 4:15:00 PM |
|---|---|---|

Font: 12 pt

| Page 19: [38] Formatted | AugRev | 8/10/20 4:15:00 PM |
|---|---|---|

Font: 12 pt

| Page 19: [38] Formatted | AugRev | 8/10/20 4:15:00 PM |
|---|---|---|

Font: 12 pt

| Page 19: [38] Formatted | AugRev | 8/10/20 4:15:00 PM |
|---|---|---|

Font: 12 pt

| Page 19: [38] Formatted | AugRev | 8/10/20 4:15:00 PM |
|---|---|---|

Font: 12 pt

| Page 19: [39] Formatted | AugRev | 8/10/20 4:15:00 PM |
|---|---|---|

Font: 12 pt

| Page 19: [39] Formatted | AugRev | 8/10/20 4:15:00 PM |
|---|---|---|

Font: 12 pt

| Page 19: [39] Formatted | AugRev | 8/10/20 4:15:00 PM |
|---|---|---|

Font: 12 pt

| Page 19: [39] Formatted | AugRev | 8/10/20 4:15:00 PM |
|---|---|---|

Font: 12 pt

| Page 19: [39] Formatted | AugRev | 8/10/20 4:15:00 PM |
|---|---|---|

Font: 12 pt

| Page 19: [39] Formatted | AugRev | 8/10/20 4:15:00 PM |
|---|---|---|

Font: 12 pt

| Page 19: [39] Formatted | AugRev | 8/10/20 4:15:00 PM |
|---|---|---|

Font: 12 pt

| Page 19: [39] Formatted | AugRev | 8/10/20 4:15:00 PM |
|---|---|---|

Font: 12 pt

| Page 19: [39] Formatted | AugRev | 8/10/20 4:15:00 PM |
|---|---|---|

Font: 12 pt

| Page 19: [39] Formatted | AugRev | 8/10/20 4:15:00 PM |
|---|---|---|

Font: 12 pt

| Page 19: [39] Formatted | AugRev | 8/10/20 4:15:00 PM |
|---|---|---|

Font: 12 pt

| Page 19: [39] Formatted | AugRev | 8/10/20 4:15:00 PM |
|---|---|---|

Font: 12 pt

| Page 19: [39] Formatted | AugRev | 8/10/20 4:15:00 PM |
|---|---|---|

Font: 12 pt

| Page 19: [39] Formatted | AugRev | 8/10/20 4:15:00 PM |
|---|---|---|

Font: 12 pt

| Page 19: [40] Deleted | AugRev | 8/10/20 4:15:00 PM |
|---|---|---|

| Page 19: [40] Deleted | AugRev | 8/10/20 4:15:00 PM |
|---|---|---|

| Page 19: [41] Deleted | AugRev | 8/10/20 4:15:00 PM |
|---|---|---|

| Page 19: [41] Deleted | AugRev | 8/10/20 4:15:00 PM |
|---|---|---|

| Page 22: [42] Deleted | AugRev | 8/10/20 4:15:00 PM |
|---|---|---|

| Page 22: [43] Deleted | AugRev | 8/10/20 4:15:00 PM |
|---|---|---|

| Page 22: [44] Formatted | AugRev | 8/10/20 4:15:00 PM |
|---|---|---|

Font: Not Italic, Not Superscript/ Subscript

| Page 22: [45] Deleted | AugRev | 8/10/20 4:15:00 PM |
|---|---|---|

| Page 25: [46] Deleted | AugRev | 8/10/20 4:15:00 PM |
|---|---|---|

**Figure 9.** Summary of the MERET algorithm. See text for further detail.

| Page 25: [47] Deleted | AugRev | 8/10/20 4:15:00 PM |
|---|---|---|

| Page 25: [48] Deleted | AugRev | 8/10/20 4:15:00 PM |
|---|---|---|

| Page 28: [49] Formatted | AugRev | 8/10/20 4:15:00 PM |
|---|---|---|

Font: 10.5 pt

| Page 28: [49] Formatted | AugRev | 8/10/20 4:15:00 PM |
|---|---|---|

Font: 10.5 pt

| Page 28: [49] Formatted | AugRev | 8/10/20 4:15:00 PM |
|---|---|---|

Font: 10.5 pt

| Page 28: [50] Formatted | AugRev | 8/10/20 4:15:00 PM |
|---|---|---|

Font: 10.5 pt

| Page 28: [50] Formatted | AugRev | 8/10/20 4:15:00 PM |
|---|---|---|

Font: 10.5 pt

| Page 28: [50] Formatted | AugRev | 8/10/20 4:15:00 PM |
|---|---|---|

Font: 10.5 pt

| Page 28: [50] Formatted | AugRev | 8/10/20 4:15:00 PM |
|---|---|---|

Font: 10.5 pt

| Page 28: [51] Formatted | AugRev | 8/10/20 4:15:00 PM |
|---|---|---|

Font: 10.5 pt

| Page 28: [51] Formatted | AugRev | 8/10/20 4:15:00 PM |
|---|---|---|

Font: 10.5 pt

| Page 28: [51] Formatted | AugRev | 8/10/20 4:15:00 PM |
|---|---|---|

Font: 10.5 pt

| Page 28: [51] Formatted | AugRev | 8/10/20 4:15:00 PM |
|---|---|---|

Font: 10.5 pt

| Page 28: [52] Formatted | AugRev | 8/10/20 4:15:00 PM |
|---|---|---|

Font: 10.5 pt

| Page 28: [52] Formatted | AugRev | 8/10/20 4:15:00 PM |
|---|---|---|

Font: 10.5 pt

| Page 28: [52] Formatted | AugRev | 8/10/20 4:15:00 PM |
|---|---|---|

Font: 10.5 pt

| Page 28: [52] Formatted | AugRev | 8/10/20 4:15:00 PM |
|---|---|---|

Font: 10.5 pt

| Page 28: [52] Formatted | AugRev | 8/10/20 4:15:00 PM |
|---|---|---|

Font: 10.5 pt

| Page 28: [52] Formatted | AugRev | 8/10/20 4:15:00 PM |
|---|---|---|

Font: 10.5 pt

| Page 28: [53] Formatted | AugRev | 8/10/20 4:15:00 PM |
|---|---|---|

Font: 10.5 pt

| Page 28: [53] Formatted | AugRev | 8/10/20 4:15:00 PM |
|---|---|---|

Font: 10.5 pt

---

## Author Response (AR3)

| Line 13 U.S –eliminate second period                                                                            |                                                                                                                                                                                                                                                        | V                                       |
|-----------------------------------------------------------------------------------------------------------------|--------------------------------------------------------------------------------------------------------------------------------------------------------------------------------------------------------------------------------------------------------|-----------------------------------------|
| Line 78: Results: Estimation of xi0 and Cburn (use proper super and subscripts)                                 |                                                                                                                                                                                                                                                        |                                         |
| Line 139: In a later pa
we excluded these pl
comma)                                                       | rt of the campaign, the DC-8 sampled in Northern Canada (Simpson et al., 2011);
umes as representing different , more boreal, forest burn conditions. (space before                                                                                 | ٧                                       |
| Table 1: Prefer to not                                                                                          | use " in table                                                                                                                                                                                                                                         |                                         |
| Line 364: needs a per                                                                                           | iod. They and the non-plume points suggest air-mass changes in CO2+CO                                                                                                                                                                                  | V                                       |
| Table 2: Periods some                                                                                           | etimes in "signifies" descriptor, sometimes not. Make consistent.                                                                                                                                                                                      | (understand)
√                       |
| Table 2:                                                                                                        |                                                                                                                                                                                                                                                        |                                         |
| expansion an
or casually a                                                                                   | Ind rise. may be assumed, to be of the estimated as $\hat{x}_i^0$ is $x_i^0$ . This is not necessarily air surrounding the plume                                                                                                                       | V wording
improved
(see
below) |
| Caps on "May". Not s
at end.                                                                                 | ure I understand the sentence that begins with "May". Eliminate exclamation point                                                                                                                                                                      | v
v                                  |
| Table 2 continued:                                                                                              |                                                                                                                                                                                                                                                        |                                         |
| or                                                                                                              | Background concentration of tracer $j$ . Typically estimated as a mini-
mum value from observed probability density function for samples in
a particular flight intensive, especially non-plume samples without
signals of stratospheric air. | V wording
improved                   |
| or? (must be missing something here)                                                                            |                                                                                                                                                                                                                                                        | V                                       |
| What does this mean                                                                                             | ? (Table 2):                                                                                                                                                                                                                                           | √ wording                               |
| An early approximation to implied from normalized and scaling, not required by algorithm but a convenient check |                                                                                                                                                                                                                                                        | rewritten
(see
holow)             |
| Table 2: - font change mid sentence                                                                             |                                                                                                                                                                                                                                                        | below)                                  |
| $\mathcal{Y}_{ij}^0$                                                                                            | y intercept implied by $x_i$ , $y_{ij}$ and the estimated slopes $a_j$ for $j$                                                                                                                                                                         | v                                       |
| Line 736: 6.3. Summa
Summary of the MER                                                                      | ry of the MERET algorithm and notes – not consistent with table of contents. (6.3.
ET method)                                                                                                                                                       | V                                       |
| Line 923: 8.1 Table of                                                                                          | several significant emissions (repeat of 8.1 should be labeled 8.2)                                                                                                                                                                                    | ٧                                       |
|                                                                                                                 |                                                                                                                                                                                                                                                        |                                         |
|                                                                                                                 |                                                                                                                                                                                                                                                        |                                         |

| Symbol                                      | Signifies                                                                                                                                                                                                                                                                                             | Obse
Estime
Hypor | erve
ated,
theti | d,
, or
cal |
|---------------------------------------------|-------------------------------------------------------------------------------------------------------------------------------------------------------------------------------------------------------------------------------------------------------------------------------------------------------|-------------------------|------------------------|-------------------|
| $C_{\rm tot}$                               | $CO_2 + CO$ (+ other carbon, ignored here), ppm.                                                                                                                                                                                                                                                      | 0                       |                        |                   |
| C burn                           | $CO_2 + CO$ (+ other carbon, ignored) emitted from fire, present downwind, in plume sample, to be estimated as $(x_i - x_i^0)$ ).                                                                                                                                                                     |                         | Ε                      |                   |
| $C_{\rm bkgd}$                              | $CO_2 + CO$ not emitted from fire, present downwind in plume sample,
thought of as a mixture of $C_{tot}$ entrained at various stages in plume
expansion and rise. The background may be assumed for illustration,
or computed from the estimated $x_i^0$ . This is not necessarily air sur- |                         | Ε                      | H *        |
| $C_{\rm bkgd}^{\rm Approx}, v_i$            | rounding the plume sample.

[revised manuscript text omitted]

1. Introduction

70

75

80

1.1. Importance of previous work

1.2. Development of EF estimation to date

- 2. Methodology: defining an indicator dataset
- 3. Observed behavior of  $C_{tot}$  in fire plumes Properties of tracers
- Theory: expanding plume for several species

   A general relationship
   Examples showing robustness of computations of idealized Ctot
   Theory: A regression relationship for EnRs
   Methodology:

   Finding the CO2 + CO background
- 6.2. Practicalities: variable EnRs
- 6.3. Summary of the MERET algorithm and notes
- 6.4. Number of independent samples
- 7. Results: Estimation of  $x_{i}^{0}$  and  $C_{burn}$

 of

 to
 e

 sa

 ta difference

 ples
 on

 •

space between paragraphs of the same style

- 8. Estimates of emissions ratios: Two MERET examples
  - 8.1. MERET results for our two examples
  - 8.2. Table of several significant emissions
- Conclusions
   9.1. Questions for future research

End material and References.

A further guide is given at the end of Section 1.

Let us introduce our view of enhancement ratios, emission ratios, and emission factors. Under appropriately defined circumstances, the amount of fuel carbon burned that is liberated to the atmosphere is the sum of carbon added to the ambient air in the form of all fire-originated gases and particles as a result of combustion: In deriving emissions factors, i.e., how much of a species is emitted per kg of biomass burned, it is usual to obtain the amount of carbon burned by taking the difference of the sum of excess mixing ratios,  $CO_2 + CO +$  other carbon-containing emissions, including aerosol particles. To an accuracy within  $\geq 1.5$  % (totals from the datasets we analyzed) to 3 % (Andreae and Merlet, 2001), carbon burned or  $C_{burn}$  is approximated by the excess ( $CO_2 + CO$ ), as measured above a background concentration,  $C_{bkgd}$  (Andreae et al., 1988)

105 where the  $\Delta$ 's refer to the enhancement relative to pre-burn air, and (O)VOCs refers to the carbon content of volatile organic species, possibly oxygenated (O). In measurement situations, where frequent, accurate measurements of CH4 and particulate *C* are also available, their inclusion could add <1% precision to the estimates. Analysis proceeds similarly including these terms. This work uses some algebra and graphics, so we introduce  $x = C_{Tot}$  and  $x^0 C_{Bkgd}$ 110 An Enhancement Ratio (EnR) for a species or property *j* with mixing ratio  $y_i$  is then

An Enhancement Ratio (EnR) for a species or property *j* with mixing ratio  $y_j$  is then EnRj =  $\Delta y_j / \Delta C_{\text{Burn}}$ . We will use this term "enhancement ratio," EnR, in this paper. When EnRs are sampled prior to substantial atmospheric transformation (e.g., chemistry or particulate processes), they describe ERs. More on the relationships of EnRs, ERs, and EFs is found in the Supplementary Material (SM), "Note on EnRs and ERs". ER estimation constitutes the analysis of atmospheric samples that contribute to EF. Emission factors are defined relative to the amount

of fuel burned and are derived from emission ratios by accounting for the concentration of carbon in the biomass burned and adjustment of units (Andreae and Merlet, 2001). Separate methods of land analysis are employed. EFs can be derived from ERs by

$$\mathrm{EF}_{j} = \mathrm{ER}_{j} \times \frac{MW_{j}}{MW_{c}} \times C_{\mathrm{BM}} \tag{2}$$

where  $ER_j$  is the emission ratio of species *j*,  $MW_j$  and  $MW_c$  are the molecular weight of species *j* and the atomic weight of carbon, respectively, and  $C_{BM}$  is the carbon content of the dry biomass. We focus on improving methods of finding EnRs and ERs, which enable EF estimation.

One part of EF estimation concerns the amount of fuel consumed in fires, its carbon content, and the fraction liberated to the atmosphere (i.e., excluding char remaining on the ground); here we will focus on the other part of the question, which concerns the relationship of emitted compounds to the C liberated to the atmosphere. Many of the EnRs we calculate appear good candidates for EF estimates. One remaining task, making specific links of particular EFs to appropri(Moved (insertion) [2]

[revised manuscript text omitted]

**215 1.2. Development of EF estimation to date**

EnRs and EFs for biomass burning plumes have largely been based on measurements of the CO2 or CO concentrations in the plumes. Typical analyses begin with measurements of  $C_{tot}$  and the concentrations of several tracers we may call  $y_i$ ,  $j = 1, ..., N_{\text{Tracers.}}$  Multiple instances i = 1, ... $N_{\text{Instances}}$  are observed, e.g., every few seconds or few minutes within a plume. An affine depend-220 ence (linear polynomial relationship including an intercept) is observed between each of the tracers and  $C_{tot}$  with a y-intercept that depends most significantly on the local out-of-plume background values of CO2, CO, and each tracer individually.  $C_{\rm tot} = C_{\rm bkgd} + C_{\rm burn}$ (3) The following analysis suggests several complexities that must be addressed in order to under-225 stand these affine relationships. Several aspects of slopes, intercepts, and deviations from linearity of the relationship of tracer  $y_i$  to  $C_{tot}$  plots must be examined, and so we transition to graphic terminology with x representing  $C_{tot}$ . Later we will describe measurements of  $C_{tot}$  and tracers j at a given instance i,  $x_i$  and  $y_{ij}$ . For a simple plume within a homogeneous mixed layer characterized by an x concentration  $x^0$  and y concentration  $y^E$ , we write  $x = x^0 + (x - x^0)$

| Formatted: Font: 12 pt  |  |
|-------------------------|--|
| Formatted: Font: 12 pt  |  |
| Deleted: x E |  |
| Formatted: Font: 12 pt  |  |
| Formatted: Font: 12 pt  |  |
| Deleted: E              |  |
| Formatted: Font: 12 pt  |  |
| Formatted: Font: 12 pt  |  |
| Formatted: Font: 12 pt  |  |
| Deleted: E              |  |
| Formatted: Font: 12 pt  |  |
| Formatted: Font: 12 pt  |  |

(4)

[revised manuscript text omitted]

| Formatted [2]                                                                                                                                                                                                                                                                                                                                                                                    |
|--------------------------------------------------------------------------------------------------------------------------------------------------------------------------------------------------------------------------------------------------------------------------------------------------------------------------------------------------------------------------------------------------|
| Deleted: E                                                                                                                                                                                                                                                                                                                                                                                       |
| Formatted: Font: 12 pt                                                                                                                                                                                                                                                                                                                                                                           |
| Deleted: EFsNRs directly. Early estimations (e.g., Greenberg et al., 1984; Andreae et al. 1988) used plots and regressions against CO 2 to estimate EnRs and EFs. These earliest techniques assumed fire was the main origin of CO 2 . Very early it was recognized that other effects, e.g., variation of photosynthesis, respiration, and mixing required a more careful |
| Formatted: Font: (Default) Times New Roman                                                                                                                                                                                                                                                                                                                                                       |
| Formatted: Font: 12 pt                                                                                                                                                                                                                                                                                                                                                                           |
| Formatted [4]                                                                                                                                                                                                                                                                                                                                                                                    |
| Formatted: Font: 12 pt                                                                                                                                                                                                                                                                                                                                                                           |
| Deleted:                                                                                                                                                                                                                                                                                                                                                                                         |
| Deleted: times                                                                                                                                                                                                                                                                                                                                                                                   |
| Formatted: Font: 12 pt                                                                                                                                                                                                                                                                                                                                                                           |
| Formatted: Font: 12 pt                                                                                                                                                                                                                                                                                                                                                                           |
| Formatted [5]                                                                                                                                                                                                                                                                                                                                                                                    |
| Deleted: careful, best                                                                                                                                                                                                                                                                                                                                                                           |
|                                                                                                                                                                                                                                                                                                                                                                                                  |
| Deleted: E                                                                                                                                                                                                                                                                                                                                                                                       |
| Deleted: careful                                                                                                                                                                                                                                                                                                                                                                                 |
| Deleted: E                                                                                                                                                                                                                                                                                                                                                                                       |

[revised manuscript text omitted]

Moved (insertion) [3]

in Section 8.2 and Chatfield and Andreae (2019). We noted some localized observations of perplexing, consistently negative  $\Delta CH_{4/2}C_{tot}$  relationships in the ARCTAS data (but not other species) and removed these observation instances. Such relationships were found close to seaports or oil-producing regions.

405

400

| Table 1. Indicator Variables                                    |                                     |                            |                                                |                                                               | Formatted: Line spacing: single, Keep lines |
|-----------------------------------------------------------------|-------------------------------------|----------------------------|------------------------------------------------|---------------------------------------------------------------|---------------------------------------------|
| Concentration / property Abbreviation Technique Group Reference |                                     | Reference                  | together                                       |                                                               |                                             |
|                                                                 |                                     | _                          |                                                |                                                               | Formatted: Keep lines together              |
| Extensive quantities                                            | Proportional to to                  | otal burned materia        | l, as measured b                               | y C burn :                                         |                                             |
| Toluene                                                         | $C_6H_5CH_3$                        | PTRMS                      | Wisthaler                                      | Wisthaler, et al. 2013.                                       | Deleted: 1                                  |
| Benzene                                                         | C 6 H 6       | PTRMS                      | Wisthaler                                      | Wisthaler, et al. 2013.                                       | Deleted: 1                                  |
| Formaldehyde                                                    | НСНО                                | LAS                        | Fried                                          | Fried et al., 2008.                                           |                                             |
| Acetonitrile                                                    | CH 3 CN                  | PTRMS                      | Wisthaler                                      | Wisthaler, et al. 2013.                                       | Deleted: 1                                  |
| Absorption Coefficient Dry,
Total, 532 nm                    | babs, Abs_5                         | Nephelometry               | Anderson                                       | Wagner et al., 2015, Anderson
Langley Aerosol Group, LARGE |                                             |
| Scattering Coefficient, Dry,
Submicron 550 nm                | $b_{\text{acat}}$ , Scat_5          | Nephelometry               | Anderson                                       | Wagner et al., 2015, Anderson
Langley Aerosol Group, LARGE | Deleted:                                    |
| Carbon monoxide                                                 | СО                                  | LAS, GC                    | Diskin, Blake                                  | Pfister et al., 2011.                                         | Deleted:                                    |
| Acetaldehyde                                                    | CH 3 CHO                 | PTRMS                      | Wisthaler                                      | Wisthaler, et all. 2013.                                      |                                             |
| Intensive quantities                                            | Not proportional to                 | carbon burned              |                                                |                                                               |                                             |
| Single Scattering Albedo                                        | SSA                                 | Nephelometry               | Anderson                                       | Wagner et al., 2015, Anderson                                 |                                             |
| Ångström Exponent, scattering                                   | ÅE                                  | Nephelometry               | Anderson                                       | Wagner et al., 2015, Anderson                                 | Deleted: "                                  |
| Other variables used                                            | $O_3$ , $NO_x = NO + NO_2$ , $NO_y$ | Chemilumines-
cence, UV | Weinheimer
(ARCTAS)
Ryerson
(SEAC4RS) | Weinheimer, et al. et al. 1994.
Ryerson et al., 2000       |                                             |
| Methane                                                         | CH 4                     | LAS, GC                    | Diskin, Blake                                  | Pfister et al., 2011.                                         |                                             |
| Methanol                                                        | CH 3 OH                  | PTRMS                      | Wisthaler                                      | Wisthaler, et al. 2013.                                       | Deleted: 1                                  |

[revised manuscript text omitted]

---

## Author Response (AR4)

Line 13 U.S.. –eliminate second period

Line 78: Results: Estimation of xi0 and Cburn (use proper super and subscripts)

Line 139: In a later part of the campaign, the DC-8 sampled in Northern Canada (Simpson et al., 2011); we excluded these plumes as representing different , more boreal, forest burn conditions. (space before comma)

Table 1: Prefer to not use " in table

Line 364: needs a period. They and the non-plume points suggest air-mass changes in CO2+CO

Table 2: Periods sometimes in "signifies" descriptor, sometimes not. Make consistent.

Table 2:

expansion and rise.    may be assumed, to be  of the estimated as $\hat{x}_i^{\upsilon}$ or casually as $x_i^0$. This is not necessarily air surrounding the plume sample.!

Caps on "May". Not sure I understand the sentence that begins with "May". Eliminate exclamation point at end.

Table 2 continued:

or                Background concentration of tracer $j$ . Typically estimated as a minimum value from observed probability density function for samples in a particular flight intensive, especially non-plume samples without signals of stratospheric air.

or? (must be missing something here)

What does this mean? (Table 2):

An early approximation to implied from normalized and scaling, not
required by algorithm but a convenient check

Table 2: - font change mid sentence

$y_{ij}^{\upsilon}$                $y$ intercept implied by $x_i$ , $y_{ij}$ and the estimated slopes $a_j$ for $j$

Line 736: 6.3. Summary of the MERET algorithm and notes – not consistent with table of contents. (6.3. Summary of the MERET method)

Line 923: 8.1 Table of several significant emissions (repeat of 8.1 should be labeled 8.2)

√
√

√

√
(understand)
√

√ wording
improved
(see
below)
√

√ wording
improved

√

√ wording
rewritten
(see
below)

√

√

√

[revised manuscript text omitted]